# Mitigating Propagation Failures in PINNs using Evolutionary Sampling

## Abstract

Despite the success of physics-informed neural networks (PINNs) in approximating partial differential equations (PDEs), it is known that PINNs can sometimes fail to converge to the correct solution in problems involving complicated PDEs. This is reflected in several recent studies on characterizing and mitigating the "failure modes" of PINNs. While most of these studies have focused on balancing loss functions or adaptively tuning PDE coefficients, what is missing is a thorough understanding of the connection between failure modes of PINNs and sampling strategies used for training PINNs. In this paper, we provide a novel perspective of failure modes of PINNs by hypothesizing that the training of PINNs rely on successful "propagation" of solution from initial and/or boundary condition points to interior points. We show that PINNs with poor sampling strategies can get stuck at trivial solutions if there are *propagation failures*. We additionally demonstrate that propagation failures are characterized by highly imbalanced PDE residual fields where very high residuals are observed over very narrow regions. To mitigate propagation failures, we propose a novel *evolutionary sampling* (Evo) method that can incrementally accumulate collocation points in regions of high PDE residuals with little to no computational overhead. We provide an extension of Evo to respect the principle of causality while solving time-dependent PDEs. We theoretically analyze the behavior of Evo and empirically demonstrate its efficacy and efficiency in comparison with baselines on a variety of PDE problems.

## 1 Introduction

Physics-informed neural networks (PINNs) (Raissi et al., 2019) represent a seminal line of work in deep learning for solving partial differential equations (PDEs), which appear naturally in a number of domains. The basic idea of PINNs for solving a PDE is to train a neural network to minimize errors w.r.t. the solution provided at initial/boundary points of a spatio-temporal domain, as well as the PDE residuals observed over a sample of interior points, referred to as collocation points. Despite the success of PINNs, it is known that PINNs can sometimes fail to converge to the correct solution in problems involving complicated PDEs, as reflected in several recent studies on characterizing the "failure modes" of PINNs (Wang et al., 2020; 2022c; Krishnapriyan et al., 2021). Many of these failure modes are related to the susceptibility of PINNs in getting stuck at trivial solutions acting as poor local minima, due to the unique optimization challenges of PINNs. In particular, note that training PINNs is different from conventional deep learning problems as we only have access to the correct solution on the initial and/or boundary points, while for all interior points in the domain, we can only compute PDE residuals. Also note that minimizing PDE residuals does not guarantee convergence to a correct solution since there are many trivial solutions of commonly observed PDEs that show 0 residuals. While previous studies on understanding and preventing failure modes of PINNs have mainly focused on modifying network architectures or balancing loss functions during PINN training, the effect of sampling collocation points on avoiding failure modes of PINNs has been largely overlooked. Although some previous approaches have explored the effect of sampling strategies on PINN training (Wang et al., 2022a; Lu et al., 2021), they either suffer from large computation costs or fail to converge to correct solutions, empirically demonstrated in our results.

In this work, we present a novel perspective of failure modes of PINNs by postulating the propagation hypothesis: "in order for PINNs to avoid converging to trivial solutions at interior points, the correct solution must be *propagated* from the initial/boundary points to the interior points." When this propagation is hindered, PINNs can get stuck at trivial solutions that are difficult to escape, referred to as the *propagation failure* mode. This hypothesis is motivated from a similar behavior observed in

numerical methods where the solution of the PDE at initial/boundary points are iteratively propagated to interior points using finite differencing schemes (LeVeque, 2007).

We show that propagation failures in PINNs are characterized by highly imbalanced PDE residual fields, where very high residuals are observed in narrow regions of the domain. Such high residual regions are not adequately sampled in the set of collocation points (which generally is kept fixed across all training iterations), making it difficult to overcome the propagation failure mode. This motivates us to develop sampling strategies that focus on selecting more collocation points from high residual regions. This is related to the idea of local-adaptive mesh refinement used in FEM (Zienkiewicz et al., 2005) to selectively refine the computational mesh in regions with higher errors.

We propose a novel *evolutionary sampling* (Evo) strategy that can accumulate collocation points in high PDE residual regions, thereby dynamically emphasizing on these skewed regions as we progress in training iterations. We also provide a causal extension of our proposed Evo algorithm (Causal Evo) that can explicitly encode the strong inductive bias of causality in propagating the solution from initial points to interior points over training iterations, when solving time-dependent PDEs. We theoretically prove the adaptive quality of Evo to accumulate points from high residual regions that persist over iterations. We empirically demonstrate the efficacy and efficiency of our proposed sampling methods in a variety of benchmark PDE problems previously studied in the PINN literature. We show the Evo and Causal Evo are able to mitigate propagation failure modes and converge to the correct solution with significantly smaller sample sizes as compared to baseline methods, while incurring negligible computational overhead. We also demonstrate the ability of Evo to solve a particularly hard PDE problem—solving 2D Eikonal equations for complex arbitrary surface geometries.

The novel contributions of our work are as follows: (1) We provide a novel perspective for characterizing failure modes in PINNs by postulating the "Propagation Hypothesis." (2) We propose a novel evolutionary algorithm Evo to adaptively sample collocation points in PINNs that shows superior prediction performance empirically with little to no computational overhead compared to existing methods for adaptive sampling. (3) We theoretically show that Evo can accumulate points from high residual regions if they persist over iterations and release points if they have been resolved by PINN training, while maintaining non-zero representation of points from other regions.

## 2 BACKGROUND AND RELATED WORK

**Physics-Informed Neural Networks (PINNs).** The basic formulation of PINN (Raissi et al., 2017) is to use a neural network $f_\theta(x, t)$ to infer the forward solution $u$ of a non-linear PDE:

$$u_t + \mathcal{N}_x[u] = 0, \ x \in \mathcal{X}, t \in [0, T]; \ u(x, 0) = h(x), \ x \in \mathcal{X}; \ u(x, t) = g(x, t), \ t \in [0, T], x \in \partial\mathcal{X}$$

where $x$ and $t$ are the space and time coordinates, respectively, $\mathcal{X}$ is the spatial domain, $\partial\mathcal{X}$ is the boundary of spatial domain, and $T$ is the time horizon. The PDE is enforced on the entire spatio-temporal domain ($\Omega = \mathcal{X} \times [0, T]$) on a set of collocation points $\{\mathbf{x_r}^i = (x_r^i, t_r^i)\}_{i=1}^{N_r}$ by computing the PDE residual ($\mathcal{R}(x, t)$) and the corresponding PDE Loss ($\mathcal{L}_r$) as follows:

$$\mathcal{R}_\theta(x, t) = \frac{\partial}{\partial t} f_\theta(x, t) - \mathcal{N}_x[f_\theta(x, t)] \tag{1}$$

$$\mathcal{L}_r(\theta) = \mathbb{E}_{\mathbf{x_r} \sim \mathcal{U}(\Omega)}[\mathcal{R}_\theta(\mathbf{x_r})^2] \approx \frac{1}{N_r} \sum_{i=1}^{N_r} [\mathcal{R}_\theta(x_r^i, t_r^i)]^2 \tag{2}$$

where $\mathcal{L}_r$ is the expectation of the squared PDE Residuals over collocation points sampled from a uniform distribution $\mathcal{U}$. PINNs approximate the solution of the PDE by optimizing the following overall loss function $\mathcal{L} = \lambda_r \mathcal{L}_r(\theta) + \lambda_{bc} \mathcal{L}_{bc}(\theta) + \lambda_{ic} \mathcal{L}_{ic}(\theta)$, where $\mathcal{L}_{ic}$ and $\mathcal{L}_{bc}$ are the mean squared loss on the initial and boundary data respectively, and $\lambda_r, \lambda_{ic}, \lambda_{bc}$ are hyperparameters that control the interplay between the different loss terms. Although PINNs can be applied to inverse problems, i.e., to estimate PDE parameters from observations, we only focus on forward problems in this paper.

**Prior Work on Characterizing Failure Modes of PINNs.** Despite the popularity of PINNs in approximating PDEs, several works have emphasized the presence of failure modes while training PINNs. One early work (Wang et al., 2020) demonstrated that imbalance in the gradients of multiple loss terms could lead to poor convergence of PINNs, motivating the development of Adaptive PINNs. Another recent development (Wang et al., 2022c) made use of the Neural Tangent Kernel (NTK) theory to indicate that the different convergence rates of the loss terms can lead to training instabilities.

Large values of PDE coefficients have also been connected to possible failure modes in PINNs (Krishnapriyan et al., 2021). In another line of work, the tendency of PINNs to get stuck at trivial solutions due to poor initializations has been demonstrated theoretically in Wong et al. (2022) and empirically in Rohrhofer et al. (2022). In all these works, the effect of sampling collocation points on PINN failure modes has largely been overlooked. Although some recent works have explored strategies to grow the representation of collocation points with high residuals, either by modifying the sampling procedure Wu et al. (2022); Lu et al. (2021); Nabian et al. (2021) or choosing higher-order $L^p$ norms of PDE loss Wang et al. (2022a), they either require a prohibitively dense set of collocation points and hence are computationally expensive, or suffer from poor convergence to the correct solution as empirically demonstrated in Section 5. In another recent line of work on Causal PINNs (Wang et al., 2022b), it was shown that traditional approaches for training PINNs can violate the principle of causality for time-dependent PDEs. Hence, they proposed an explicit way of incorporating the causal structure in the training procedure. However, this solution can only be applied to time-dependent PDEs, and as we demonstrate empirically in Section 5, also requires large sample sizes.

## 3 PROPAGATION HYPOTHESIS: A NEW PERSPECTIVE OF FAILURE MODES IN PINNS

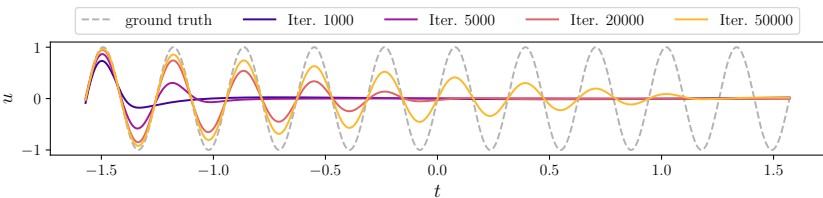

Figure 1: PINN solutions for a simple ODE: $u_{xx} + k^2 u = 0$ ($k = 20$) with the analytical solution, $u = A\sin(kx) + B\cos(kx)$. We can see smooth *propagation* of the correct solution from the boundary point at $x = 0$ to interior points ($x > 0$) as we increase training iterations.

**What is Unique About Training PINNs?** Training PINNs presents fundamentally different optimization challenges than those encountered in conventional deep learning problems. In a conventional supervised learning problem, the correct solution for every training sample is known and the training samples are considered representative of the test samples such that the trained model can easily be extrapolated on closely situated test samples. However, in the context of PINNs, we only have access to the "correct" solution of the PDE on the initial and/or boundary points, while not having any labels for the interior points in the spatio-temporal domain $\Omega$. Note that the interior points in $\Omega$ can be quite far away from the initial/boundary points, making extrapolation difficult. Further, training PINNs involves minimizing the PDE residuals over a set of collocation points sampled from $\Omega$. However, minimizing PDE residuals alone is not sufficient to ensure convergence to the correct solution, since there may exist many trivial solutions of a PDE showing very small residuals. For example, $u(x, t) = 0$ is a trivial solution for any homogeneous PDE, which a neural network is likely to get stuck at in the absence of correct solution at initial/boundary points. Another unique aspect of training PINNs is that minimizing PDE residuals requires computing the gradients of the output w.r.t. $(x, t)$ (e.g., $u_x$ and $u_t$). Hence, the solution at a collocation point is affected by the solutions at nearby points leading to local propagation of solutions.

**Propagation Hypothesis.** In light of the unique properties of PINNs, we postulate that in order for PINNs to converge to the "correct" solution, the correct solution must propagate from the initial and/or boundary points to the interior points as we progress in training iterations. We draw inspiration for this hypothesis from a similar behavior observed in numerical methods for solving PDEs, where the solution of the PDE at initial/boundary points are iteratively propagated to interior points using finite differencing schemes (LeVeque, 2007). Figure 1 demonstrates the propagation hypothesis of PINNs for a simple ordinary differential equation (ODE).

**Propagation *Failure*: Why It Happens and How to Diagnose?** As a corollary of the propagation hypothesis, PINNs can fail to converge to the correct solution if the solution at initial/boundary points is *unable to propagate* to interior points during the training process. We call this phenomenon the "*propagation failure*" mode of PINNs. This is likely to happen if some collocation points start converging to trivial solutions before the correct solution from initial/boundary points is able to reach

them. Such collocation points would also propagate their trivial solutions to nearby interior points, leading to a cascading effect in the learning of trivial solutions over large regions of $\Omega$ and further hindering the propagation of the correct solution from initial/boundary points.

To *diagnose* propagation failures, note that the PDE residuals are expected to be low over both types of regions: regions that have converged to the correct solution and regions that have converged to trivial solutions. However, the boundaries of these two types of regions would show a sharp discontinuity in solutions, leading to very high PDE residuals in very narrow regions. A similar phenomenon is observed in numerical methods where sharp high-error regions disrupt the evolution of the PDE solution at surrounding regions, leading to cascading of errors. We use the imbalance of high PDE residual regions as a diagnosis tool for characterizing propagation failure modes in PINNs.

To demonstrate propagation failure, let us consider an example PDE for the convection equation: $\frac{\partial u}{\partial t} + \beta \frac{\partial u}{\partial x} = 0, u(x, 0) = h(x)$, where $\beta$ is the convection coefficient and $h(x)$ is the initial condition (see Appendix G for details about this PDE). In a previous work (Krishnapriyan et al., 2021), it has been shown that PINNs fail to converge for this PDE for $\beta > 10$. We experiment with two cases, $\beta = 10$ and $\beta = 50$, in Figure 2. We can see that the PDE loss steadily decreases with training iterations for both these cases, but the relative error w.r.t. the ground-truth solution only decreases for $\beta = 10$, while for $\beta = 50$, it remains flat. This suggests that for $\beta = 50$, PINN is likely getting stuck at a trivial solution that shows low PDE residuals but high errors. To diagnose this failure mode, we plot two additional metrics in Figure 2 to measure the imbalance in high PDE residual regions: Fisher-Pearson's coefficient of Skewness (Kokoska & Zwillinger, 2000) and Fisher's Kurtosis (Kokoska & Zwillinger, 2000) (see Appendix H for computation details). High Skewness indicates lack of symmetry in the distribution of PDE residuals while high Kurtosis indicates the presence of a heavy-tail. For $\beta = 10$, we can see that both Skewness and Kurtosis are relatively small across all iterations, indicating absence of imbalance in the residual field. However, for $\beta = 50$, both these metrics shoot up significantly as the training progresses, which indicates the formation of very high residuals in very narrow regions—a characteristic feature of the propagation failure mode. Figure 3 confirms that this indeed is the case by visualizing the PINN solution and PDE residual maps. We see similar trends of propagation failure for other values of $\beta > 10$ (see Appendix J.1).

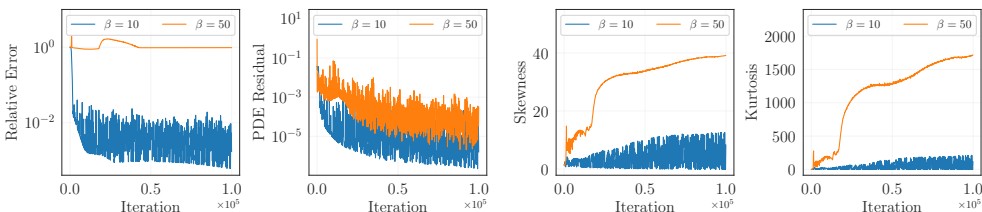

Figure 2: Demonstration of propagation failure while solving the convection equation with $\beta = 50$.

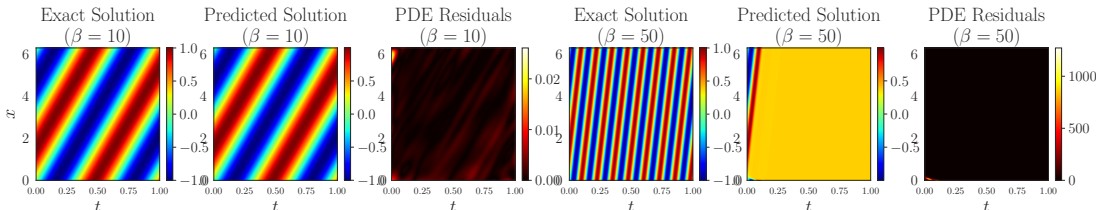

Figure 3: Demonstration of the Exact PDE solution, the predicted PDE solution, and the PDE Residual Field for the convection equation with $\beta = 10$ and $\beta = 50$ respectively.

**Potential Remedies for Mitigating Propagation Failures.** Since *propagation failures* in PINNs are characterized by skewed regions with very high residuals acting as "barriers" in the flow of information, algorithms that can selectively focus on high residual regions during training can potentially break such barriers and thus offer some respite. Recently, a number of algorithms have been proposed to increase the importance of collocation points from high residual regions in PINN training, which can be broadly grouped into two categories. The first category includes methods that alter the sampling strategy such that a point $\mathbf{x_r}$ is picked up as a collocation point with probability proportional to its residual, i.e., $p(\mathbf{x_r}) \propto |\mathcal{R}(\mathbf{x_r})|^k$, where $k \geq 1$. This includes residual-based

adaptive refinement (RAR) methods (Lu et al., 2021) and its variants Wu et al. (2022); Nabian et al. (2021), where a dense set of collocation points $\mathcal{P}_{dense}$ is maintained to approximate the continuous residual field $\mathcal{R}(\mathbf{x})$, and points with high residuals are regularly added from $\mathcal{P}_{dense}$ to the set of collocation points every $K$ iterations according to a sampling function. A second line of work was recently proposed in Wang et al. (2022a), where higher-order $L^p$ norms of the PDE loss (e.g., $L^\infty$) were advocated to be used in the training of PINNs in contrast to the standard practice of using $L^2$ norms, to ensure stability of learned solutions in control problems involving high-dimensional PDEs. Note that by using higher-order $L^p$ norms, we are effectively increasing the importance of collocation points from high residual regions in the PDE loss, thereby having a similar effect as increasing their representation in the sample of collocation points (see Theorem A.1 for more details).

**Challenges with Potential Remedies.** There are two main challenges faced by the potential remedies described above that limit their effectiveness in mitigating propagation failures in PINNs. (1) **High computational complexity:** Sampling methods such as RAR and its variants require using a dense set of collocation points $\mathcal{P}_{dense}$ (typically with 100k $\sim$ 1M points spread uniformly across the entire domain) to locate high residual regions, such that points from high residual regions can added to the training set every $K$ iterations. This increases the computational cost in two ways. First, computing the PDE residuals on the entire dense set is very expensive. Second, the size of the training set keeps growing every $K$ iterations, further increasing the training costs at later iterations. See Appendix F for a detailed analysis of the computational complexity of RAR based methods (2) **Poor Prediction Performance:** As we empirically demonstrate in Section 5, both sampling-based methods such as RAR and its variants as well as $L^\infty$ PDE loss-based methods suffer from poor performance in converging to the correct solution for complex PDE problems. This can be attributed to several reasons. First, while increasing the value of $k$ in sampling-based methods and $p$ in $L^p$ norm-based methods influences a greater skew in the sample set towards points with higher residuals, the optimal values of $k$ or $p$ are generally unknown for an arbitrary PDE. Second, choosing $L^\infty$ loss (or equivalently, only sampling collocation points with highest residuals) may not ideal for the training dynamics of PINNs, as it can lead to oscillatory behavior between different peaks of the PDE residual landscape, while forgetting to retain the solution over other regions. Instead, we need a way to gradually accumulate points from high residual regions in the collocation set while having non-zero representation of points from other regions.

## 4 PROPOSED APPROACH: EVOLUTIONARY SAMPLING (EVO)

Before we present our proposed sampling approach, in the following we first describe the *four motivating properties* that we focus on in this work to mitigate propagation failures: (1) *Accumulation Property*: To break propagation barriers, the set of collocation points should evolve over every iteration such that they start from a uniform distribution and accumulate in regions with high residuals until the PINN training process eventually resolves them in later iterations. This is similar to starting with an $L^2$ loss and increasing the order of $L^p$ norm if high residual regions persist over iterations. (2) *Sample Release Property*: Upon sufficient minimization of a high residual region through PINN training, collocation points that were once accumulated from the region needs to be *released*, such that we can focus on minimizing other high residual regions in later iterations. (3) *Uniform Background Property*: At every iteration, the set of collocation points should contain non-zero support of points sampled from a uniform distribution over the entire domain $\Omega$, such that the collocation points do not collapse solely to high residual regions. (4) *Computational Efficiency*: While satisfying the above properties, we should incur little to no computational overhead in sampling collocation points from high residual regions. Specifically, we should be able to add points from high residual regions without maintaining a dense set of collocation points, $\mathcal{P}_{dense}$, and by only observing the residuals over a small set of $N_r$ points at every iteration.

To satisfy all of the above properties, we present a novel sampling strategy termed *Evolutionary Sampling (Evo)* that is inspired by algorithms used for modeling biological evolution (Eiben et al., 2003). The key idea behind Evo is to gradually *evolve* the population of collocation points at every iteration by retaining points with high residuals from the current population and re-sampling new points from a uniform distribution.

Algorithm 1 shows the pseudo-code of our proposed evolutionary sampling strategy. Analogous to evolutionary algorithms developed in optimization literature (Simon, 2013), we introduce a notion of "fitness" $\mathcal{F}(\mathbf{x_r})$ for every collocation point $\mathbf{x_r}$ such that points with higher fitness are allowed to survive in the next iteration. Specifically, we define $\mathcal{F}(\mathbf{x_r})$ as the absolute value of PDE residual of

---

**Algorithm 1** Proposed Evolutionary Sampling Algorithm For PINN

---

1: Sample the initial population $\mathcal{P}_0$ of $N_r$ collocations point $\mathcal{P}_0 \leftarrow \{\mathbf{x_r}\}_{i=1}^{N_r}$ from a uniform distribution $\mathbf{x_r}^i \sim \mathcal{U}(\Omega)$, where $\Omega$ is the input domain ($\Omega = [0, T] \times \mathcal{X}$).
2: **for** i = 0 to max_iterations - 1 **do**
3:     Compute the fitness of collocation points $\mathbf{x_r} \in \mathcal{P}_i$ as $\mathcal{F}(\mathbf{x_r}) = |\mathcal{R}(\mathbf{x_r})|$.
4:     Compute the threshold $\tau_i = \frac{1}{N_r} \sum_{j=1}^{N_r} \mathcal{F}(\mathbf{x_r}^j)$
5:     Select the retained population $\mathcal{P}_i^r$ such that $\mathcal{P}_i^r \leftarrow \{\mathbf{x_r}^j : \mathcal{F}(\mathbf{x_r}^j) > \tau_{it}\}$
6:     Generate the re-sampled population $\mathcal{P}_i^s \leftarrow \{\mathbf{x_r}^j : \mathbf{x_r}^j \sim \mathcal{U}(\Omega)\}$, s.t. $|\mathcal{P}_i^s| + |\mathcal{P}_i^r| = N_r$
7:     Merge the two populations $\mathcal{P}_{i+1} \leftarrow \mathcal{P}_i^r \cup \mathcal{P}_i^s$
8: **end for**

---

$\mathbf{x_r}$, i.e., $\mathcal{F}(\mathbf{x_r}) = |\mathcal{R}(\mathbf{x_r})|$. At iteration 0, we start with an initial population $\mathcal{P}_0$ of $N_r$ points sampled from a uniform distribution. At iteration $i$, in order to evolve the population to the next iteration, we first construct the "retained population" $\mathcal{P}_i^r$ comprising of points from $\mathcal{P}_i$ with fitness values greater than $\tau_i$, i.e., $\mathcal{P}_i^r \leftarrow \{\mathbf{x}_r^j : \mathcal{F}(\mathbf{x}_r^j) > \tau_i\}$, where $\tau_i$ is equal to the expectation of fitness values over all points in $\mathcal{P}_i$. The remainder of collocation points in $\mathcal{P}_i$ are re-sampled from a uniform distribution, thus constructing the "re-sampled population" $\mathcal{P}_i^s \leftarrow \{\mathbf{x_r}^j : \mathbf{x_r}^j \sim \mathcal{U}(\Omega)\}$. The retained population and the re-sampled population are then merged to generate the population for the next iteration, $\mathcal{P}_{i+1}$. Appendix Figure 8 schematically shows the accumulation of collocation points from high residual regions in Evo over training iterations.

**Analysis of Evo:** Note that at every iteration of PINN training, Evo attempts to retain the set of collocation points in $\mathcal{P}^r$ with the highest fitness (corresponding to high residual regions). At the same time, the PINN optimizer is attempting to minimize the residuals by updating $\theta$ and thus in turn affecting the fitness function. We first show that when $\mathcal{F}(\mathbf{x})$ is fixed (e.g., when $\theta$ is kept constant), Evo maximizes $\mathcal{F}(\mathbf{x})$ in the retained population and thus *accumulates* points from high residual regions. In particular, Theorem 4.1 shows that for a fixed $\mathcal{F}(\mathbf{x})$, the expectation of the retained population in Evo becomes maximum (equal to $L^\infty$) when the number of iterations approaches $\infty$.

**Theorem 4.1** (Accumulation Dynamics Theorem). *Let $\mathcal{F}_\theta(\mathbf{x}) : \mathbb{R}^n \to \mathbb{R}^+$ be a fixed real-valued $k$-Lipschitz continuous objective function optimized using the Evolutionary Sampling algorithm. Then, the expectation of the retained population $\mathbb{E}_{\mathbf{x} \in \mathcal{P}^r}[\mathcal{F}(\mathbf{x})] \geq \max_{\mathbf{x}} \mathcal{F}(\mathbf{x}) - k\epsilon$ as iteration $i \to \infty$, for any arbitrarily small $\epsilon > 0$.*

The proof of Theorem 4.1 can be found in Appendix C.1. This demonstrates the *accumulation property* of Evo as points from high residual regions would keep accumulating in the retained population and make its expectation maximal if the fitness function is kept fixed. However, since the PINN optimizer is also minimizing the residuals at every iteration, we would not expect the fitness function to be fixed unless a high residual region persists over a long number of iterations. In fact, points from a high residual region would keep on accumulating until they are resolved by the PINN optimizer and thus eventually released from $\mathcal{P}^r$. Also note that Evo always maintains some collocation points from a uniform distribution, i.e., the re-sampled population $\mathcal{P}^s$ is always non-empty. Theoretical proofs of the *sample release property* and the *uniform background property* of Evo are provided in Appendix C.2 and Lemma C.1.1, respectively. We also provide details of the computational complexity of Evo in comparison with baseline methods in Appendix F, showing that Evo is *computationally efficient*. Table 2 in the Appendix summarizes our ability to satisfy the motivating properties of our work in comparison with baselines. Note that Evo shares a similar motivation as *local-adaptive mesh refinement* methods developed for Finite Element Methods (FEM) (Zienkiewicz et al., 2005), where the goal is to preferentially refine the computational mesh used in numerical methods based on localization of the errors. It is also related to the idea of *boosting* in ensemble learning where training samples with larger errors are assigned higher weights of being picked in the next epoch, to increasingly focus on high error regions (Schapire, 2003).

### 4.1 CAUSAL EXTENSION OF EVOLUTIONARY SAMPLING (CAUSAL EVO)

In problems with time-dependent PDEs, a strong prior dictating the propagation of solution is the *principle of causality*, where the solution of the PDE needs to be well-approximated at time $t$ before moving to time $t + \Delta t$. To incorporate this prior guidance, we present a Causal Extension of Evo (*Causal Evo*) that includes two modifications: (1) we develop a causal formulation of the PDE loss $\mathcal{L}_r$ that pays attention to the temporal evolution of PDE solutions over iterations, and (2) we develop a

Figure 4: Comparison of Skewness, Kurtosis, Max PDE Residuals, and Mean PDE Residuals over training iterations for PINN-fixed, PINN-dynamic, and Evo for convection equation with $\beta = 50$.

causally biased sampling scheme that respects the causal structure while sampling collocation points. We describe both these modifications in the following.

**Causal Formulation of PDE Loss.** The key idea here is to utilize a simple time-dependent gate function $g(t)$ that can explicitly enforce causality by revealing only a portion of the entire time-domain to PINN training. Specifically, we introduce a continuous gate function $g(t) = (1 - \tanh(\alpha(\tilde{t} - \gamma)))/2$, where $\gamma$ is the scalar shift parameter that controls the fraction of time that is revealed to the model, $\alpha = 5$ is a constant scalar parameter that determines the steepness of the gate, and $\tilde{t}$ is the normalized time, i.e., $\tilde{t} = t/T$. Example causal gates for different settings of $\gamma$ are provided in Appendix D. We use $g(t)$ to obtain a causally-weighted PDE residual loss as $\mathcal{L}_r^g(\theta) = \frac{1}{N_r} \sum_{i=1}^{N_r} [\mathcal{R}(x_r^i, t_r^i)]^2 * g(t_r^i)$. We initially start with a small value of the shift parameter ($\gamma = -0.5$), which essentially only reveals a very small portion of the time domain, and then gradually increase $\gamma$ during training to reveal more portions of the time domain. For $\gamma \geq 1.5$, the entire time domain is revealed.

**Causally Biased Sampling.** We bias the sampling strategy in Evo such that it not only favors the selection of collocation points from high residual regions but also accounts for the causal gate values at every iteration. In particular, we modify the fitness function as $\mathcal{F}(\mathbf{x_r}) = |\mathcal{R}(\mathbf{x_r})| * g(t_r)$. A schematic illustration of causally biased sampling is provided in Appendix D.

**How to Update $\gamma$?** Ideally, at some iteration $i$, we would like increase $\gamma_i$ at the next iteration *only if* the PDE residuals at iteration $i$ are low. Otherwise, $\gamma_i$ should remain in its place until the PDE residuals under the current gate are minimized. To achieve this behaviour, we propose the following update scheme for $\gamma$: $\gamma_{i+1} = \gamma_i + \eta_g e^{-\epsilon \mathcal{L}_r^g(\theta)}$, where $\eta_g$ is the learning rate and $\epsilon$ denotes tolerance that controls how low the PDE loss needs to be before the gate shifts to the right. Since the update in $\gamma$ is inversely proportional to the causally-weighted PDE loss $\mathcal{L}_r^g$, the gate will shift slowly if the PDE residuals are large. Also note that increasing $\gamma$ also increases the value of $g(t)$ for all collocation points, thus increasing the causally-weighted PDE loss and slowing down gate movement. Upon convergence, $\gamma$ attains a large value such that the entire time domain is revealed.

Table 1: Relative $\mathcal{L}_2$ errors (in %) of comparative methods over benchmark PDEs with $N_r = 1000$.

| | Convection ($\beta = 30$) | | Convection ($\beta = 50$) | | Allen Cahn |
|---|---|---|---|---|---|
| Epochs. | 100k | 300k | 150k | 300k | 200k |
| PINN (fixed) | $107.5 \pm 10.9\%$ | $107.5 \pm 10.7\%$ | $108.5 \pm 6.38\%$ | $108.7 \pm 6.59\%$ | $69.4 \pm 4.02\%$ |
| PINN (dynamic) | $2.81 \pm 1.45\%$ | $1.35 \pm 0.59\%$ | $24.2 \pm 23.2\%$ | $56.9 \pm 9.08\%$ | $0.77 \pm 0.06\%$ |
| Curr Reg | $63.2 \pm 9.89\%$ | $2.65 \pm 1.44\%$ | $48.9 \pm 7.44\%$ | $31.5 \pm 16.6\%$ | - |
| CPINN (fixed) | $138.8 \pm 11.0\%$ | $138.8 \pm 11.0\%$ | $106.5 \pm 10.5\%$ | $106.5 \pm 10.5\%$ | $48.7 \pm 19.6\%$ |
| CPINN (dynamic) | $52.2 \pm 43.6\%$ | $23.8 \pm 45.1\%$ | $79.0 \pm 5.11\%$ | $73.2 \pm 8.36\%$ | $1.5 \pm 0.75\%$ |
| RAR-G | $10.5 \pm 5.67\%$ | $2.66 \pm 1.41\%$ | $65.7 \pm 17.0\%$ | $43.1 \pm 28.9\%$ | $25.1 \pm 23.2\%$ |
| RAD | $3.35 \pm 2.02\%$ | $1.85 \pm 1.90\%$ | $66.0 \pm 1.55\%$ | $64.1 \pm 1.98\%$ | $0.78 \pm 0.05\%$ |
| RAR-D | $67.1 \pm 4.28\%$ | $32.0 \pm 25.8\%$ | $82.9 \pm 5.99\%$ | $75.3 \pm 9.58\%$ | $51.6 \pm 0.41\%$ |
| $L^\infty$ | $66.6 \pm 2.35\%$ | $41.2 \pm 27.9\%$ | $76.6 \pm 1.04\%$ | $75.8 \pm 1.01\%$ | $1.65 \pm 1.36\%$ |
| Evo. (ours) | $\mathbf{1.51 \pm 0.26}\%$ | $0.78 \pm 0.18\%$ | $6.03 \pm 6.99\%$ | $\mathbf{1.98 \pm 0.72}\%$ | $0.83 \pm 0.15\%$ |
| Causal Evo. (ours) | $2.12 \pm 0.67\%$ | $\mathbf{0.75 \pm 0.12}\%$ | $\mathbf{5.99 \pm 5.25}\%$ | $2.28 \pm 0.76\%$ | $\mathbf{0.71 \pm 0.007}\%$ |

## 5 RESULTS

**Experiment Setup.** We perform experiments over three benchmark PDEs that have been used in existing literature to study failure modes of PINNs. In particular, we consider two time-dependent PDEs: convection equation (with $\beta = 30$ and $\beta = 50$) and Allen Cahn equation, and one time-independent PDE: the Eikonal equation for solving signed distance fields for varying input geometries.

We consider the following baseline methods: **(1) PINN-fixed** (conventional PINN using a fixed set of uniformly sampled collocation points), **(2) PINN-dynamic** (a simple baseline where collocation points are dynamically sampled from a uniform distribution every iteration, see Appendix B for more details), **(3) Curr. Reg.** (curriculum regularization method proposed in (Krishnapriyan et al., 2021)), **(4) cPINN-fixed** (causal PINNs proposed in (Wang et al., 2022b) with fixed sampling), **(5) cPINN-dynamic** (cPINN with dynamic sampling), **(6) RAR-G** (Residual-based Adaptive Refinement strategy proposed in (Lu et al., 2021)), **(7) RAD** (Residual-based Adaptive Distribution originally proposed in (Nabian et al., 2021) and later generalized in Wu et al. (2022)), **(8) RAR-D** (RAR with a sampling Distribution proposed in Wu et al. (2022)), **(9)** $L^\infty$ (Sampling top $N_r$ collocation points at every iteration from a dense set $\mathcal{P}_{dense}$ to approximate $L^\infty$ norm). For every benchmark PDE, we use the same neural network architecture and hyper-parameter settings across all baselines and our proposed methods, wherever possible. Details about the PDEs, experiment setups, and hyper-parameter settings are provided in Appendix I. All code and datasets used in this paper are available at [1]

**Comparing Prediction Performance.** Table 1 shows the relative $\mathcal{L}_2$ errors (over 5 random seeds) of PDE solutions obtained by comparative methods w.r.t. ground-truth solutions for different time-dependent PDEs when $N_r$ is set to 1K. We particularly chose a small value of $N_r$ to study the effect of small sample size on PINN performance (note that the original formulations of baseline methods used very high $N_r$). We can see that while PINN-fixed fails to converge for convection ($\beta = 30$) and Allen Cahn equations (admitting very high errors), PINN-dynamic shows significantly lower errors. However, for complex PDEs such as convection ($\beta = 50$), PINN-dynamic is still not able to converge to low errors. We also see that cPINN-fixed shows high errors across all PDEs when $N_r = 1000$. This is likely because the small size of collocation samples are insufficient for cPINNs to converge to the correct solution. As we show later, cPINN indeed is able to converge to the correct solution when $N_r$ is large. Performing dynamic sampling with cPINN shows some reduction in errors, but it is still not sufficient for convection ($\beta = 50$) case. All other baseline methods including Curr Reg, RAR-based methods, and $L^\infty$ fail to converge on most PDEs and show worse performance than even the simple baseline of PINN-dynamic. On the other hand, our proposed approaches (Evo and Causal Evo) consistently show the lowest errors across all PDEs. Figure 4 shows that Evo and PINN-dynamic are indeed able to mitigate

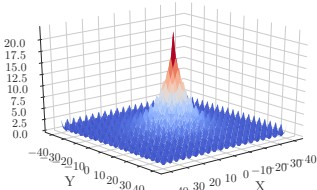

(a) Surface Plot of Auckley Function

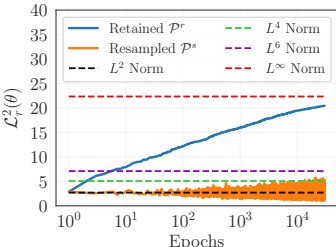

(b) Dynamics of PDE loss for Evo

Figure 5: PDE loss of $\mathcal{P}^r$ going from $L^2$ to $L^\infty$ with iterations for a sample objective function.

*propagation failures* for convection ($\beta = 50$), by maintaining low values of Skewness, Kurtosis, and Max PDE residuals across all iterations, in contrast to PINN-fixed. Additional visualizations of the evolution of samples in Evo, Caual Evo, and RAR-based methods across iterations are provided in Appendix J. Sensitivity of RAR-based methods to hyper-parameters is provided in Appendix J.3.

**Adaptive Nature of Evo PDE loss:** To demonstrate the ability of Evo to accumulate high residual points in the retained population $\mathcal{P}^r$ (or equivalently, focus on higher-order $L^p$ norms of PDE loss), we consider optimizing a fixed objective function: the Auckley function in Figure 5 (see Appendix K.1 for details of this function). We can see that at iteration 1, the expected loss over $\mathcal{P}^r$ is equal to the $L^2$ norm of PDE loss over the entire domains. As training progresses, the expected loss over $\mathcal{P}^r$ quickly reaches higher-order $L^p$ norms, and approaches $L^\infty$ at very large iterations. This confirms the gradual accumulation property of Evo as theoretically stated in Theorem 4.1. Addition visualizations of the dynamics of Evo for a number of test optimization functions are provided in Appendix K.

**Sampling Efficiency:** Figure 6 shows the scalability of Evo to smaller sample sizes of collocation points, $N_r$. Though all the baselines demonstrate similar performances when $N_r$ is large ($> 10K$), only Evo and Causal Evo manage to maintain low errors even for very small values of $N_r = 100$, showing two orders of magnitude improvement in sampling efficiency. Note that the sample size $N_r$ is directly related to the compute and memory requirements of training PINNs. We also show that Evo and Causal Evo show faster convergence speed than baseline methods for both convection

---

[1]https://www.dropbox.com/sh/45gec7qvgiutz2x/AABz4aFJ1IfMIxY11OkLbJ9Oa?dl=0.

and Allen Cahn equations (see Appendix J.2 for details). Additional results on three cases of Kuramoto-Shivashinsky (KS) Equations including chaotic behavior are provided in Appendix J.7.

**Solving Eikonal Equations.** Given the equation of a surface geometry in a 2D-space, $u(x_s, y_s) = 0$, the Eikonal equation is a time-independent PDE used to solve for the *signed distance field* (SDF), $u(x, y)$, which has negative values inside the surface and positive values outside the surface. See Appendix G for details of the Eikonal equation. The primary difficulty in solving Eikonal equation comes from determining the sign of the field (interior or exterior) in regions with rich details. We compare the performance of different baseline methods with respect to the ground-truth (GT) solution obtained from numerical methods for three complex surface geometries in Figure 7. We also plot the reconstructed geometry of the predicted solutions to demonstrate the real-world application of solving this PDE, e.g., in downstream real-time graphics rendering. The quality of reconstructed geometries are quantitatively evaluated using the mean Intersection-Over-Union (mIOU) metric. The results show that PINN-fixed shows poor performance across all three geometries, while PINN-dynamic is able to capture most of the outline of the solutions with a few details missing for difficult geometries like "sailboat" and "gear". On the other hand, Evo is able to capture even the fine details of the SDF for all three complex geometries and thus show better reconstruction quality. We can see that mIOU of Evo is significantly higher than baselines for "sailboat" and "gear". See Appendix Section J.8 for more discussion and visualizations.

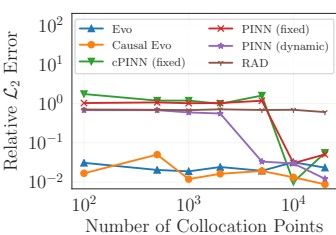

(a) Convection ($\beta = 50$, Iter: 300k)

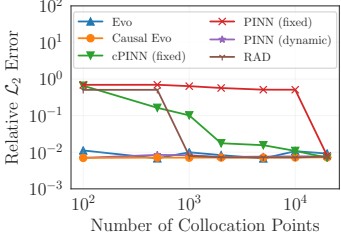

(b) Allen Cahn (Iter: 200k)

Figure 6: Sample Efficiency of Evo and Causal Evo at low $N_r$.

## 6 CONCLUSIONS AND FUTURE WORK DIRECTIONS

We present a novel perspective for identifying failure modes in PINNs named "*propagation failures.*" and develop a novel *evolutionary sampling* algorithm to mitigate propagation failures. From our experiments, we demonstrate better performance on a variety of benchmark PDEs. Future work can focus on theoretically understanding the interplay between minimizing PDE loss and sampling from high residual regions on PINN performance. Other directions of future work can include exploring more sophisticated evolutionary algorithms involving mutation and crossover techniques.

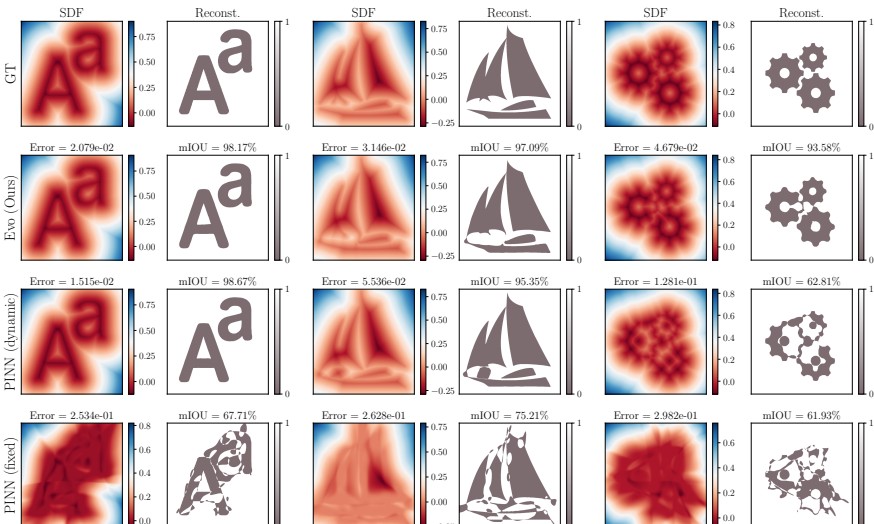

Figure 7: Solving Eikonal equation for signed distance field (SDF). The color of the heatmap represents the values of the SDF. The gray region shows the negative values of SDF that represents the interior points in the reconstructed geometry from predicted SDF.

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

# A    CONNECTIONS BETWEEN $L^p$ NORM AND SAMPLING

In this section, we provide connections between adaptively sampling collocation points from a distribution $q(\mathbf{x_r}) \propto |\mathcal{R}_\theta(\mathbf{x_r})|^k$ Wu et al. (2022) and using $L^p$ norm of the PDE loss Wang et al. (2022a).

**Theorem A.1.** *For $p \geq 2$, let $\mathcal{L}_r^p(\mathcal{U})$ denote the expected $L^p$ PDE Loss computed on collocation points sampled from a uniform distribution, $\mathcal{U}(\Omega)$. Similarly, for $k \geq 0$, let $\mathcal{L}_r^2(\mathcal{Q}^k)$ denote the expected $L^2$ PDE Loss computed on collocation points sampled from an alternate distribution $\mathcal{Q}^k(\Omega) : \mathbf{x_r} \sim q(\mathbf{x_r})$, where $q(\mathbf{x_r}) \propto |\mathcal{R}_\theta(\mathbf{x_r})|^k$. Then, $\mathcal{L}_r^2(\mathcal{Q}^k) = \frac{1}{Z^{1/2}V^{-1/2}} \left( \mathcal{L}_r^{k+2}(\mathcal{U}) \right)^{(k+2)/2}$, where $Z$ is a normalization constant, defined as $Z = \int_{\mathbf{x}_r \in \Omega} |\mathcal{R}_\theta(\mathbf{x_r})|^k d\mathbf{x_r}$ and $V$ is the volume of the domain $\Omega$.*

*Proof.* The expectation of $L^p$ PDE Loss for collocation points sampled from a uniform distribution $\mathcal{U}(\Omega) : \mathbf{x_r} \sim p(\mathbf{x_r})$ can be defined as follows [2]:

$$
\begin{aligned}
\mathcal{L}_r^p(\mathcal{U}) &= \left( \mathbb{E}_{\mathbf{x_r} \sim \mathcal{U}(\Omega)} |\mathcal{R}_\theta(\mathbf{x_r})|^p \right)^{1/p} \\
&= \left( \int p(\mathbf{x_r}) |\mathcal{R}_\theta(\mathbf{x_r})|^p d\mathbf{x_r} \right)^{1/p}
\end{aligned}
\tag{3}
$$

Note that for a uniform distribution, $p(\mathbf{x_r}) = \frac{1}{V}$, where $V$ is the volume of the domain $\Omega$, i.e., $V = \prod_{i=1}^n \left( \text{supp}(x_i) - \inf(x_i) \right)$ with supp(.) and inf(.) being the supremum and infimum operators, and $x_i$ is the $i$-th dimension of $\mathbf{x_r}$ (e.g., the space dimension $x$ or the time dimension $t$).

Now, let us consider the case where we are interested in sampling from an alternate distribution $\mathcal{Q}^k(\Omega) : \mathbf{x_r} \sim q(\mathbf{x_r})$, where $q(\mathbf{x_r}) \propto |\mathcal{R}_\theta(\mathbf{x_r})|^k$ while using the $L^2$ PDE Loss (the most standard loss formulation used in PINNs). The sampling function of $\mathcal{Q}^k(\Omega)$ can be defined as follows:

$$
q(\mathbf{x_r}) = \frac{|\mathcal{R}_\theta(\mathbf{x_r})|^k}{Z}
\tag{4}
$$

where $Z$ is the normalizing constant, i.e., $Z = \int_{\mathbf{x}_r \in \Omega} |\mathcal{R}_\theta(\mathbf{x_r})|^k d\mathbf{x_r}$.

Hence, the $L^2$ PDE Loss for collocation points sampled from $\mathcal{Q}^k(\Omega)$ can be defined as:

$$
\begin{aligned}
\mathcal{L}_r^2(\mathcal{Q}^k) &= \left( \mathbb{E}_{\mathbf{x_r} \sim \mathcal{Q}^k(\Omega)} |\mathcal{R}_\theta(\mathbf{x_r})|^2 \right)^{1/2} \\
&= \left( \int q(\mathbf{x_r}) |\mathcal{R}_\theta(\mathbf{x_r})|^2 d\mathbf{x_r} \right)^{1/2} \\
&= \left( \int \frac{|\mathcal{R}_\theta(\mathbf{x_r})|^k}{Z} |\mathcal{R}_\theta(\mathbf{x_r})|^2 d\mathbf{x_r} \right)^{1/2} && \text{(From Equation 4)} \\
&= \frac{1}{Z^{1/2}} \left( \int |\mathcal{R}_\theta(\mathbf{x_r})|^{k+2} d\mathbf{x_r} \right)^{1/2} \\
&= \frac{1}{Z^{1/2}} \left( \int \frac{p(\mathbf{x_r})}{p(\mathbf{x_r})} |\mathcal{R}_\theta(\mathbf{x_r})|^{k+2} d\mathbf{x_r} \right)^{1/2} \\
&= \frac{1}{Z^{1/2}V^{-1/2}} \left( \int p(\mathbf{x_r}) |\mathcal{R}_\theta(\mathbf{x_r})|^{k+2} d\mathbf{x_r} \right)^{1/2}, && \because \mathbf{x_r} \sim \mathcal{U}(\Omega) \implies p(\mathbf{x_r}) = \frac{1}{V} \\
&= \frac{1}{Z^{1/2}V^{-1/2}} \left( \mathbb{E}_{\mathbf{x_r} \sim \mathcal{U}(\Omega)} |\mathcal{R}_\theta(\mathbf{x_r})|^{k+2} \right)^{1/2} \\
&= \frac{1}{Z^{1/2}V^{-1/2}} \left( \mathcal{L}_r^{k+2}(\mathcal{U}) \right)^{(k+2)/2} && \text{(From Equation 3, with } p = k+2)
\end{aligned}
\tag{5}
$$

$\square$

---

[2] We assume that the batch size/number of collocation points used to compute the PDE Loss tends to infinity, i.e., $N_r \to \infty$. This allows us to analyze the behavior of the continuous PDE loss function $\mathcal{L}_r(\theta)$ as $N_r \to \infty$

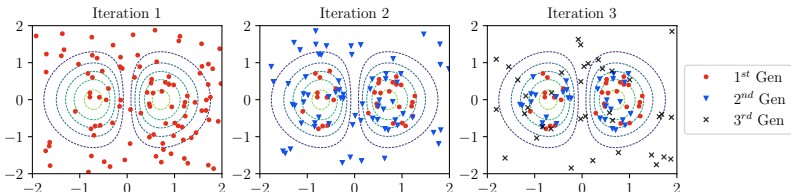

Figure 8: Schematic to describe our proposed evolutionary sampling algorithm, where collocation points are incrementally accumulated in regions with high PDE residuals (shown as contour lines).

Theorem A.1 suggests that sampling collocation points from a distribution $q(\mathbf{x_r}) \propto |\mathcal{R}_\theta(\mathbf{x_r})|^k$ and $L^p$ norm of the PDE loss are related to each other by a scaling term $\frac{1}{Z^{1/2}V^{-1/2}}$. However, since this scaling term is variable in nature as $Z$ depends on the neural network parameters $\theta$, optimizing $\mathcal{L}_r^p(\mathcal{U})$ is not directly equivalent to optimizing $\mathcal{L}_r^2(\mathcal{Q}^k)$ (or a power thereof).

## B  ANALYZING PINN-DYNAMIC: A STRONG BASELINE

A very simple baseline for mitigating propagation failures is to *dynamically* sample a random set of $N_r$ collocation points from a uniform distribution at every iteration, independently of previous iterations. To see how this simple sampling strategy can help in sampling points from high PDE residual regions, let us consider partitioning the entire input domain $\Omega$ into two subsets: regions with high PDE residuals, $\Omega_{high}$, and regions with low PDE residuals $\Omega_{low}$. If the high PDE residual regions are imbalanced because of propagation failure, we would expect that the area of $\Omega_{high}$ ($A_{high}$) is significantly smaller than the area of $\Omega_{low}$ ($A_{low}$), i.e., $A_{high} \ll A_{low}$. Let us compute the probability of sampling at least one collocation point from $\Omega_{high}$ across all iterations of PINN training. If we use a *fixed* set of collocation points sampled from a uniform distribution at every iteration, this probability will be equal to $A_{high}/(A_{high} + A_{low})$, which can be very small. Hence, we are likely to under-represent points from $\Omega_{high}$ in the training process and thus get stuck in trivial solutions with high residual regions around its boundaries, especially when $N_r$ is low.

On the other hand, if we perform *dynamic* sampling, the probability of picking at least one point from $\Omega_{high}$ across all iterations will be equal to $1 - (1 - A_{high}/(A_{high} + A_{low}))^N$, where $N$ is the number of iterations. We can see that when $N$ is large, this probability approaches 1, indicating that across all iterations of PINN training, we would have likely sampled a point from $\Omega_{high}$ in at least one iteration, and used it to minimize its PDE residual. As we empirically demonstrate later in Section 5, dynamic sampling is indeed able to control the skewness of PDE residuals compared to fixed sampling, and thus act as a strong baseline for mitigating propagation failures.

However, note that even if we use dynamic sampling, the contribution of points from $\Omega_{high}$ in the overall PDE residual loss computed at any iteration is still low. In particular, since the probability of sampling points from $\Omega_{high}$ at any iteration is equal to $A_{high}/(A_{high} + A_{low})$, the expected PDE residual loss computed over all collocation points will be equal to

$$\mathbb{E}_\Omega[\mathcal{L}_r(\theta)] = \mathbb{E}_{\Omega_{high}}[\mathcal{L}_r(\theta)] \times \frac{A_{high}}{A_{high} + A_{low}} + \mathbb{E}_{\Omega_{low}}[\mathcal{L}_r(\theta)] \times \frac{A_{low}}{A_{high} + A_{low}} \qquad (6)$$

Since $A_{high} \ll A_{low}$, the gradient update of $\theta$ at every epoch will be dominated by the low PDE residuals observed over points from $\Omega_{low}$, leading to slow propagation of information from initial/boundary points to interior points.

## C  ANALYSIS OF EVOLUTIONARY SAMPLING

In this section, we analyze the dynamic behavior (or evolution) of the collocation points for our proposed *Evolutionary Sampling* (Evo) approach over the iterations. A schematic representation of Evo is also provided in Figure 8.

### C.1  ACCUMULATION PROPERTY OF EVO.

In this section, we provide the proof of Theorem 4.1 presented in the main paper.

**Definition 1** (Objective Function). *Let $\mathcal{F}_\theta(\mathbf{x}) : \mathbb{R}^n \to \mathbb{R}^+$ be an arbitrary positive real-valued $k$-Lipschitz continuous function, where $\theta$ denotes the neural network parameters. When $\theta$ is fixed, the function $\mathcal{F}(\mathbf{x})$ does not vary with iterations, representing a fixed objective function.*

Let $\mathbf{X}^* = \{\mathbf{x}_i^* : \mathcal{F}(\mathbf{x}_i^*) = \max_\mathbf{x} \mathcal{F}(\mathbf{x}) \ \forall \ i \in [n]\}$ be the set of points where the objective function $\mathcal{F}$ is maximal.

Now, let us define an $\epsilon$-neighborhood around each of point $\mathbf{x}_i^* \in \mathbf{x}^*$ as $\mathcal{N}_\epsilon(\mathbf{x}_i)$ such that $||\mathbf{x}_i - \mathbf{x}_i^*|| \leq \epsilon$ for any arbitrarily small $\epsilon > 0$ and for all $\mathbf{x}_i \in \mathcal{N}_\epsilon(\mathbf{x}_i)$ with $i \in [n]$.

Let us also assume that the objection function $\mathcal{F}$ is $k$-Lipschitz continuous. Then the following is true:

$$|\mathcal{F}(\mathbf{x}_i^*) - \mathcal{F}(\mathbf{x}_i)| \leq k\epsilon \qquad \forall \ \mathbf{x}_i \in \mathcal{N}_\epsilon(\mathbf{x}_i) \ \& \ i \in [n] \qquad (7)$$

$$\implies \mathcal{F}(\mathbf{x}_i^*) - \mathcal{F}(\mathbf{x}_i) \leq k\epsilon \qquad \because \mathcal{F}(\mathbf{x}_i^*) = \max_\mathbf{x} \mathcal{F}(\mathbf{x}) \qquad (8)$$

**Definition 2** ($\epsilon$-maximal Neighborhood). *Let $\mathcal{F}^* = \max_\mathbf{x} \mathcal{F}(\mathbf{x})$ be the maximal value of the objective function $\mathcal{F}(\mathbf{x})$ (Definition 1). Then an $\epsilon$-maximal Neighborhood $\mathcal{N}_\epsilon^\infty$ can be defined as: $\mathcal{N}_\epsilon^\infty = \mathcal{N}_\epsilon(\mathbf{x}_0) \cup \mathcal{N}_\epsilon(\mathbf{x}_1) \cup ... \cup \mathcal{N}_\epsilon(\mathbf{x}_n)$ such that any point $\mathbf{x}$ sampled from $\mathcal{N}_\epsilon^\infty$ would have $\mathcal{F}^* - \mathcal{F}(\mathbf{x}) \leq k\epsilon \ \ \forall \ \mathbf{x} \in \mathcal{N}_\epsilon^\infty$ and for any arbitrarily small $\epsilon > 0$.*

Note that since the volume of $\mathcal{N}_\epsilon^\infty$ is greater than 0, the probability of sampling any $\mathbf{x} \in \mathcal{N}_\epsilon^\infty$ from a uniform distribution $\mathcal{U}(\mathbf{x})$ is greater than 0.

**Lemma C.1** (Population Properties). *For any population $\mathcal{P}$ generated at some iteration of Evo optimizing a given objective function $\mathcal{F}(\mathbf{x})$ (Definition 1), the following properties are always true:*

1. *The re-sampled population is always non-empty, i.e., $|\mathcal{P}^s| > 0$*

2. *The size of the retained population is always less than the total population size, i.e., $|\mathcal{P}^r| < |\mathcal{P}|$*

3. *The size of the retained population is zero, i.e., $|\mathcal{P}^r| = 0$, if and only if $\mathcal{F}(\mathbf{x}) = c, \forall \mathbf{x} \in \mathcal{P}$.*

*Proof.* The threshold $\tau$ for the Evolutionary Sampling can be computed as $\tau = \frac{1}{|\mathcal{P}|} \sum_{\mathbf{x} \in \mathcal{P}} \mathcal{F}(\mathbf{x})$.

The retained population is defined as: $\mathcal{P}^r \leftarrow \{\mathbf{x} : \mathcal{F}(\mathbf{x}) > \tau \ \ \forall \mathbf{x} \in \mathcal{P}\}$,

Similarly, the non-retained population can be defined as: $\overline{\mathcal{P}^r} \leftarrow \{\mathbf{x} : \mathcal{F}(\mathbf{x}) \leq \tau \ \ \forall \mathbf{x} \in \mathcal{P}\}$.

**Proof of Property 1:** For any arbitrary set of real numbers, there always exists some element in the set that is less than or equal to the mean. Hence, the size of the non-retained population is always non-zero as there always exists some point $\mathbf{x} \in \mathcal{P}$ such that $\mathcal{F}(\mathbf{x}) \leq \tau$. Thus, $|\overline{\mathcal{P}^r}| > 0$.

Now by definition, since the re-sampled population $\mathcal{P}^s$ replaces the non-retained population at every iteration, $|\mathcal{P}^s| = |\overline{\mathcal{P}^r}|$. Hence, $|\mathcal{P}^s| > 0$, i.e., the size of the resampled population is always non-zero.

**Proof of Property 2:** By definition, $|\overline{\mathcal{P}^r}| + |\mathcal{P}^r| = |\mathcal{P}|$ (where $|\mathcal{P}|$ is the total size of the population and is always constant). Since, $|\overline{\mathcal{P}^r}| > 0$, we can say that $|\mathcal{P}^r| < |\mathcal{P}|$, i.e., the size of the retained population can never be equal to the entire population size $|\mathcal{P}|$.

**Proof of Property 3:** Let us consider the case where $\mathcal{F}(\mathbf{x}) = c, \forall \mathbf{x} \in \mathcal{P}$ (where $c$ is some constant), i.e., the value of the function is constant at all of the points $\mathbf{x} \in \mathcal{P}$. In this case, the mean of the population $\mathcal{P}$, which is equal to the threshold, $\tau$ will be equal to $c$. This condition would lead to the entire population to be re-sampled as all element $\mathbf{x} \in \mathcal{P}$ would satisfy the condition to belong in the non-retained population. Note that the constant function $\mathcal{F}(\mathbf{x}) = c$ is the only case where all of the elements are less than or equal to the mean. Otherwise, there would always be at least one element greater than the mean, resulting in a non-zero size of the retained population. $\square$

**Lemma C.2** (Entry Condition). *If a point $\mathbf{x}_m$ is sampled from $\mathcal{N}_\epsilon^\infty$ at any arbitrary iteration $m$, then it will always enter the retained population $\mathcal{P}_m^r$ unless $\mathbb{E}_{\mathbf{x} \in \mathcal{P}_m^r}[\mathcal{F}(\mathbf{x})] > \mathcal{F}^* - k\epsilon$.*

*Proof.* The condition for any arbitrary point $\mathbf{x}_m$ to enter the retained population $\mathcal{P}_m^r$ at any arbitrary iteration $m$ is given by the following:

$$\mathcal{F}(\mathbf{x}_m) > \tau_m = \mathbb{E}_{\mathbf{x} \in \mathcal{P}_m}[\mathcal{F}(\mathbf{x})]. \qquad \text{(By definition of the threshold } \tau_m) \qquad (9)$$

Now, if the point $\mathbf{x}_m$ is sampled from $\mathcal{N}_\epsilon^\infty$, then $\mathcal{F}(\mathbf{x}_m) \geq \mathcal{F}^* - k\epsilon$ (from Definition 2). Hence, for $\mathbf{x}_m$ to enter the retained population $\mathcal{P}_m^r$, we need to ensure that $\mathcal{F}^* - k\epsilon > \tau_m$.

Let us consider the case where $\mathbf{x}_m$ is not able to enter the retained population. In such a case, we will have the following inequality:

$$\mathcal{F}^* - k\epsilon < \tau_m. \tag{10}$$

It is also easy to show from the definition of retained population that the threshold $\tau$ is always less than the expectation of the retained population:

$$\tau_m \leq \mathbb{E}_{\mathbf{x} \in \mathcal{P}_m^r}[\mathcal{F}(\mathbf{x})] \tag{11}$$

From Equations 10 and 11, we get,

$$\mathcal{F}^* - k\epsilon < \tau_m \leq \mathbb{E}_{\mathbf{x} \in \mathcal{P}_m^r}[\mathcal{F}(\mathbf{x})]$$
$$\implies \mathbb{E}_{\mathbf{x} \in \mathcal{P}_m^r}[\mathcal{F}(\mathbf{x})] > \mathcal{F}^* - k\epsilon \tag{12}$$

We have thus proved that $\mathbf{x}_m$ will not be able to enter the retained population $\mathcal{P}_m^r$ only if $\mathbb{E}_{\mathbf{x} \in \mathcal{P}_m^r}[\mathcal{F}(\mathbf{x})] > \mathcal{F}^* - k\epsilon$, which suggests that the expectation of the retained population is already close to $\mathcal{F}^*$, for any arbitrarily small $\epsilon > 0$. On the other hand, if $\mathbb{E}_{\mathbf{x} \in \mathcal{P}_m^r}[\mathcal{F}(\mathbf{x})] \leq \mathcal{F}^* - k\epsilon$, we would necessarily add $\mathbf{x}_m$ □

**Lemma C.3** (Exit Condition). *A point $\mathbf{x}_m$ sampled from $\mathcal{N}_\epsilon^\infty$ that entered the retained population at any arbitrary iteration $m$, can exit the retained population $\mathcal{P}_n^r$ at an arbitrary iteration $n$ (such that $n > m$) only if $\mathbb{E}_{\mathbf{x} \in \mathcal{P}_n^r}[\mathcal{F}(\mathbf{x})] \geq \mathcal{F}^* - k\epsilon$.*

*Proof.* The generic condition for any arbitrary point $\mathbf{x}$ to exit the retained population $\mathcal{P}_n^r$ at iteration $n$ is given by: $\mathcal{F}(\mathbf{x}) \leq \tau_n = \mathbb{E}_{\mathbf{x} \in \mathcal{P}_n}[\mathcal{F}(\mathbf{x})]$ (By definition of the threshold $\tau$).

Since a point $\mathbf{x}_m$ that was originally sampled from $\mathcal{N}_\epsilon^\infty$ will have $\mathcal{F}(\mathbf{x}_m) \geq \mathcal{F}^* - k\epsilon$ (from Definition 2), we can use this inequality in the generic exit condition shown above to get,

$$\mathcal{F}^* - k\epsilon \leq \tau_n \leq \mathbb{E}_{\mathbf{x} \in \mathcal{P}_n^r}[\mathcal{F}(\mathbf{x})] \tag{13}$$

Hence, the point $\mathbf{x}_m$ can exit the retained population at iteration $n$ only if $\mathbb{E}_{\mathbf{x} \in \mathcal{P}_n^r}[\mathcal{F}(\mathbf{x})] \geq \mathcal{F}^* - k\epsilon$. □

**Theorem C.4** (Accumulation Dynamics Theorem). *Let $\mathcal{F}_\theta(\mathbf{x}) : \mathbb{R}^n \to \mathbb{R}^+$ be a fixed real-valued $k$-Lipschitz continuous objective function optimized using the Evolutionary Sampling algorithm. Then, the expectation of the retained population $\mathbb{E}_{\mathbf{x} \in \mathcal{P}^r}[\mathcal{F}(\mathbf{x})] \geq \max_{\mathbf{x}} \mathcal{F}(\mathbf{x}) - k\epsilon$ as iteration $i \to \infty$, for any arbitrarily small $\epsilon > 0$.*

*Proof.* We prove this theorem by contradiction. For the sake of contradiction, let us assume that as iterations $i \to \infty$, the expectation of the retained population $\mathbb{E}_{\mathbf{x} \in \mathcal{P}^r}[\mathcal{F}(\mathbf{x})] < \max_{\mathbf{x}} \mathcal{F}(\mathbf{x}) - k\epsilon$, for any arbitrarily small $\epsilon > 0$. We can then make the following two remarks.

**Entry of collocation points**: Note that the probability of sampling $\mathbf{x}$ from $\mathcal{N}_\epsilon^\infty$ is non-zero because the size of the re-sampled population is non-zero, i.e., $|\mathcal{P}^s| > 0$ (proved in Lemma C.1). Also, since we have assumed $\mathbb{E}_{\mathbf{x} \in \mathcal{P}^r}[\mathcal{F}(\mathbf{x})] < \mathcal{F}^* - k\epsilon$, we can use the Entry condition proved in Lemma C.2 to arrive at the conclusion that a point from $\mathcal{N}_\epsilon^\infty$ will always be able to enter the retained population.

**Exit of collocation points**: Similarly, a point $\mathbf{x}$ that belongs in the $\epsilon$-maximal neighborhood and is part of the retained population $\mathcal{P}^r$ will not be able to escape the retained population as we have asssumed $\mathbb{E}_{\mathbf{x} \in \mathcal{P}^r}[\mathcal{F}(\mathbf{x})] < \mathcal{F}^* - k\epsilon$ (using the Exit condition proved in Lemmas C.3).

From the above two remarks, we can see that points would keep accumulating indefinitely in the retained population if our initial assumption (for the sake of contradiction) is true. However, since the total size of the population $|\mathcal{P}|$ is bounded, the size of the retained population $|\mathcal{P}^r|$ cannot grow indefinitely. We have thus arrived at a contradiction suggesting our assumption is incorrect. Hence, as iterations $i \to \infty$, the expectation of the retained population $\mathbb{E}_{\mathbf{x} \in \mathcal{P}^r}[\mathcal{F}(\mathbf{x})] \geq \max_{\mathbf{x}} \mathcal{F}(\mathbf{x}) - k\epsilon$, for any arbitrarily small $\epsilon > 0$.

□

From Theorem 4.1, we can prove a continuous accumulation of collocation points from the $\epsilon$-maximal neighborhood until the expectation of the retained population is close to the maximum point (i.e., reaches $L^\infty$), thus exhibiting the **Accumulation Property** described in Section 4. Although this theorem assumes that the objective function $\mathcal{F}(\mathbf{x})$ (or in the context of PINNs, the absolute residual values, $\mathcal{R}_\theta(\mathbf{x_r})$) is constant, this theorem is still valid when $\mathcal{R}_\theta(\mathbf{x_r})$ is gradually changing with the highest error regions (defined using our $\epsilon$-maximal neighborhood) persisting over iterations. Under such conditions, the theorem states that the retained population $\mathcal{P}^r$ would always accumulate points from the $\epsilon$-maximal neighborhood, thereby adaptively increasing their contribution to the overall PDE residual loss and eventually resulting in their minimization.

## C.2    RELEASE OF COLLOCATION POINTS FROM HIGH PDE RESIDUAL REGIONS.

Our definition of the **Sample Release Property** states that the distribution of collocation points should revert back to its original form by releasing the accumulated points in the high PDE residual regions once they are "*sufficiently minimized*". Let us define that for an arbitrary collocation point $\mathbf{x_r}^i \in \mathcal{P}$, "*sufficient minimization*" of the PDE is achieved if $\mathcal{R}_\theta(\mathbf{x_r}^i) \leq \mathbb{E}_{\mathbf{x_r} \in \mathcal{P}}[\mathcal{R}_\theta(\mathbf{x_r})] = \tau$ (where $\tau$ is the threshold used by Evo). Then, by definition, such points will belong to the "non-retained population" and will be immediately replaced by the re-sampled population. Since we generate the re-sampled population $\mathcal{P}^s$ from a uniform distribution, these "*sufficiently minimized*" collocation points are replaced with a uniform density. Thus, Evo satisfies the "*Sample Release Property*" of an "ideal" sampling algorithm.

# D    ADDITIONAL DETAILS FOR CAUSAL EVOLUTIONARY SAMPLING

Figure 9b represents a schematic describing the causally biased Evolutionary sampling described in Section 4.1 and the causal gate $g$ that is updated every iteration.

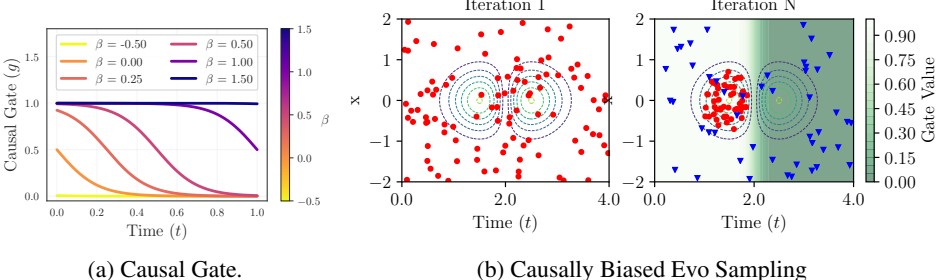

(a) Causal Gate.                          (b) Causally Biased Evo Sampling

Figure 9: Causal Evo uses a time-dependent causal gate for computing PDE loss and for sampling.

## D.1    PREVENTING ABRUPT CAUSAL GATE MOVEMENT.

The shift parameter $\gamma$ of the causal gate is updated every iteration using the following scheme: $\gamma_{i+1} = \gamma_i + \eta_g e^{-\epsilon \mathcal{L}_r^g(\theta)}$, where $\eta_g$ is the learning rate that controls how fast the gate should propagate and $\epsilon$ denotes tolerance that controls how low the PDE loss needs to be before the gate shifts to the right, and $i$ denotes the $i^{th}$ iteration. Typically, in our experiments we set the learning rate to 1e-3. Thus, for example, if the expectation of $e^{-\epsilon \mathcal{L}_r^g(\theta)}$ over 1000 iterations is 0.1, then $\gamma$ would change by a value of 0.1 after 1000 iterations (since $\gamma_{i+N} \approx \gamma_i + \eta_g * N * \mathbb{E}[e^{-\epsilon \mathcal{L}_r^g(\theta)}]$). Additionally, note that, for a typical "tanh" causal gate, the operating range of $\gamma$ values vary from $-0.5$ to $1.5$. However, if the loss is very small ($\mathcal{L}_r^g(\theta) \to 0$), the magnitude of the update $e^{-\epsilon \mathcal{L}_r^g(\theta)} \to 1$, i.e., leads to an abrupt change in the causal gate. Thus, to prevent an abrupt gate movement due to large magnitude update, we employ a magnitude clipping scheme (similar to gradient clipping in conventional ML) as follows: $\gamma_{i+1} = \gamma_i + \eta_g \min(e^{-\epsilon \mathcal{L}_r^g(\theta)}, \Delta_{max})$, where $\Delta_{max}$ is the maximum allowed magnitude of update. Typically, for our experiments we keep $\Delta_{max} = 0.1$. Note, that $\Delta_{max}$ needs to be carefully chosen depending on the gate learning rate $\eta_g$.

## D.2 Choice of Other Gate Functions.

The gate function $g$ to enforce the principle of causality is not limited to the "tanh" gate presented in Section 4.1 of the main paper. Any arbitrary function can be used for a causal gate as long as it obeys the following criteria:

1. **Continuous Time Property**: The function $g$ should be continuous in time, such that it can be evaluated at any arbitrary time $t$.

2. **Monotonic Property**: The value of gate $g$ at time $t + \Delta t$ should be less than the value of the gate at time $t$, i.e., $g(t + \Delta t) \leq g(t)$. In other words, $g$ should be a monotonically decreasing function,

3. **Shift Property**: The gate function should be parameterized using a shift parameter $\gamma$, such that $g_\gamma(t) < g_{\gamma+\delta}(t)$, where $\delta > 0$, i.e., by increasing the value of the shift parameter the gate value of any arbitrary time should increase.

An alternate choice of a causal gate is using a composition of ReLU and tanh functions: $g = ReLU(-\tanh(\alpha(t - \gamma)))$ (as shown in Figure 10. We can see that by using ReLU, this alternate gate function provides a stricter thresholding of gate values to 0 after a cutoff value of time. The effect of this strict thresholding on the incorporation of causality in training PINNs can be studied in future analyses. In our current analysis, we simply used the tanh gate function for all our experiments.

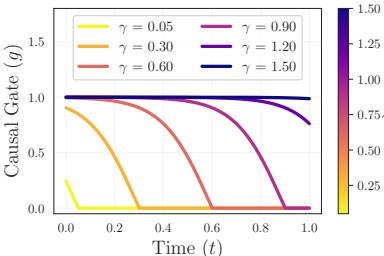

Figure 10: ReLU-tanh Causal Gate

## E Comparison of Baselines

In this Section, we compare the different baseline methods w.r.t. the motivating properties. See Table 2.

Table 2: Table comparing baseline methods in terms of their ability to comply with the motivating properties of this work.

|  | Accumulation Property | Release Property | Uniform Background Property | Computational Efficiency | Causality |
|---|---|---|---|---|---|
| RAR-G Lu et al. (2021) | ✓ |  | ✓ |  |  |
| RAD Nabian et al. (2021) | ✓ | ✓ | ✓ |  |  |
| RAR-D Wu et al. (2022) | ✓ |  | ✓ |  |  |
| $L - \infty$ Wang et al. (2022a) | ✓ | ✓ |  | ✓ |  |
| CausalPINN Wang et al. (2022b) |  |  | ✓ | ✓ | ✓ |
| Evo (ours) | ✓ | ✓ | ✓ | ✓ |  |
| Causal Evo (ours) | ✓ | ✓ | ✓ | ✓ | ✓ |

## F Computational Complexity Analysis of Evolutionary Sampling vs its baselines

In this section, we aim to provide a comprehensive comparison of the computational complexity of Evolutionary Sampling and its baselines. It is well-known that the cost of computing the PDE residuals using automatic-differentiation during training amounts is reasonably large, especially

when the PDE contains higher order gradients that require repeated backward passes through the computational graphs used by standard deep learning packages like PyTorch/Tensorflow. Thus, in this section we would mainly focus on comparing the number of PDE residual computations that each algorithm makes during training. We can quantify the effect of this difference on computational costs as follows. Let us first define the Notations that we are going to use for the analysis:

### Notations for Computational Analysis

| | |
|---|---|
| $C$ | Computational Cost to evaluate the PDE residual on a single collocation point. |
| $N$ | Total Number of Training Iterations |
| $|\mathcal{P}|$ | Total Number of Collocation Points/Population Size (Also referred to as the initial set of collocation points for RAR based methods) |
| $|\mathcal{P}_{dense}|$ | Auxiliary Set of dense points [For RAR-G,RAD, and RAR-D], where $|\mathcal{P}_{dense}| >> |\mathcal{P}|$ |
| $K$ | Resampling Period [For RAR-G,RAD, and RAR-D] |
| $M$ | Number of additional collocation points added to the initial set |

Next, we present the computational cost to run each method separately.

**PINN/Evo**: The cost of computing the PDE residual at an arbitrary epoch $i$: $C_{Evo}(i) = C|\mathcal{P}|$. Thus, the overall computational cost for $N$ iterations is $C_{Evo} = NC|\mathcal{P}|$.

**RAR-G/RAD/RAR-D**: For RAR-based methods, the initial set of collocation points $\mathcal{P}$ keeps growing as $M$ new test points are added every $K$ iterations. Thus, at an iteration $i$, the size of the collocation point set $|\mathcal{P}_i| = |\mathcal{P}_0| + m\lfloor i/K \rfloor = |\mathcal{P}| + M\lfloor i/K \rfloor$ (to simplify notations, let $|\mathcal{P}_0| = |\mathcal{P}|$, i.e., all of the methods start with the same number of initial collocation points).

Thus, the cost to compute the PDE residuals on this training set $\mathcal{P}$ is:

$$C_{RAR}^{train} = KC|\mathcal{P}| + KC(|\mathcal{P}| + M) + KC(|\mathcal{P}| + 2M) + ... + KC(|\mathcal{P}| + M\left\lfloor \frac{N}{K} \right\rfloor)$$

$$= KC|\mathcal{P}|\left( \left\lfloor \frac{N}{K} \right\rfloor + 1 + \frac{M}{2|\mathcal{P}|}\left\lfloor \frac{N}{K} \right\rfloor\left(\left\lfloor \frac{N}{K} \right\rfloor + 1\right) \right) \quad (14)$$

There is also an additional cost of evaluating the PDE residuals on the dense set $C_{RAR}^{dense}$ to select these $M$ points from high PDE residual region.

$$C_{RAR}^{dense} = C|\mathcal{P}_{dense}|\left\lfloor \frac{N}{K} \right\rfloor \quad (15)$$

Therefore, the overall cost for RAR-based methods is: $C_{RAR} = C_{RAR}^{dense} + C_{RAR}^{train}$

**Comparing the cost between Evo and RAR**: Assuming that the total number of epochs $N$ is divisible by the resampling period $K$, which is true for most practical scenarios.

$$C_{RAR} = NC|\mathcal{P}| + KC|\mathcal{P}| + \frac{MNC}{2}\left(\frac{N}{K} + 1\right) + C|\mathcal{P}_{dense}|\frac{N}{K}$$

$$C_{RAR} = C_{Evo} + KC|\mathcal{P}| + \frac{MNC}{2}\left(\frac{N}{K} + 1\right) + C|\mathcal{P}_{dense}|\frac{N}{K}$$

$$C_{RAR} - C_{Evo} = KC|\mathcal{P}| + \frac{MNC}{2}\left(\frac{N}{K} + 1\right) + C|\mathcal{P}_{dense}|\frac{N}{K} \quad (16)$$

Thus, we can see that the difference in the computational cost can quickly grow depending on the choice of $RAR$ setting. Also note that, since $|\mathcal{P}_{dense}| >> |\mathcal{P}|$, the additional cost of $C|\mathcal{P}_{dense}|\frac{N}{K}$ is significant, especially if we want to re-sample/re-evaluate the adaptive sampling frequently (i.e., for small values of $K$).

# G  DETAILS OF PARTIAL DIFFERENTIAL EQUATIONS USED IN THIS WORK

## G.1  CONVECTION EQUATION

We considered a 1D-convection equation that is commonly used to model transport phenomenon, described as follows:

$$\frac{\partial u}{\partial t} + \beta \frac{\partial u}{\partial x} = 0, \ \ x \in [0, 2\pi], t \in [0, 1] \tag{17}$$

$$u(x, 0) = h(x) \tag{18}$$

$$u(0, t) = u(2\pi, t) \tag{19}$$

where $\beta$ is the convection coefficient and h(x) is the initial condition. For our case studies, we used a constant setting of $h(x) = \sin(x)$ with periodic boundary conditions in all our experiments, while varying the value of $\beta$ in different case studies.

## G.2  ALLEN-CAHN EQUATION

We considered a 1D - Allen Cahn equation that is used to describe the process of phase-separation in multi-component alloy systems as follows:

$$\frac{\partial u}{\partial t} - 0.0001 \frac{\partial^2 u}{\partial x^2} + 5u^3 - 5u = 0, \ \ x \in [-1, 1], t \in [0, 1] \tag{20}$$

$$u(x, 0) = x^2 \cos(\pi x) \tag{21}$$

$$u(t, -1) = u(t, 1) \tag{22}$$

$$\left. \frac{\partial u}{\partial t} \right|_{x=-1} = \left. \frac{\partial u}{\partial t} \right|_{x=1} \tag{23}$$

## G.3  EIKONAL EQUATION

We formulate the Eiknonal equation for signed distance function (SDF) calculation as:

$$|\nabla u| = 1, \qquad\qquad\qquad x, t \in [-1, 1] \tag{24}$$

$$u(x_s) = 0, \qquad\qquad\qquad x_s \in \mathcal{S} \tag{25}$$

$$u(x, -1), u(x, 1), u(-1, y), u(1, y) > 0 \tag{26}$$

where $\mathcal{S}$ is zero contour set of the SDF. In training PINN, we use the zero contour constraint as initial condition loss and positive boundary constraint as boundary loss (see Table 3 for details of loss balancing).

## G.4  KURAMOTO–SIVASHINSKY EQUATION

We use 1-D Kuramoto–Sivashinsky equation from CausalPINN Wang et al. (2022b):

$$\frac{\partial u}{\partial t} + \alpha u \frac{\partial u}{\partial x} + \beta \frac{\partial^2 u}{\partial x^2} + \gamma \frac{\partial^4 u}{\partial x^4} = 0, \tag{27}$$

subject to periodic boundary conditions and an initial condition

$$u(0, x) = u_0(x) \tag{28}$$

The parameter $\alpha, \beta, \gamma$ controls the dynamical behavior of the equation. We use the same configurations as the CausalPINN: $\alpha = 5, \beta = 0.5, \gamma = 0.005$ for regular settings, and $\alpha = 100/16, \beta = 100/16^2, \gamma = 100/16^4$ for chaotic behaviors.

## H    DETAILS ON SKEWNESS AND KURTOSIS METRICS

Skewness and kurtosis are two basic metrics used in statistics to characterize the properties of a distribution of values $\{Y_i\}_{i=1}^N$. A high value of Skewness indicates lack of symmetry in the distribution, i.e., the distribution of values to the left and to the right of the center point of the distribution are not identical. On the other hand, a high value of Kurtosis indicates the presence of a heavy-tail, i.e., there are more values farther away from the center of the distribution relative to a Normal distribution. In our implementation using *scipy*, we used the adjusted Fisher-Pearson coefficient of skewness and Fisher's definition of kurtosis, as defined below.

**Skewness**: For univariate data $Y_1, Y_2, ..., Y_N$, the formula of skewness is

$$\text{skewness} = \frac{\sqrt{N(N-1)}}{N-2} \times \frac{\sum_{i=1}^N (Y_i - \bar{Y})^3 / N}{s^3}, \tag{29}$$

where $\bar{Y}$ is the sample mean of the distribution and $s$ is the standard deviation. For any symmetric distribution (e.g., Normal distribution), the skewness is equal to zero. A positive value of skewness means there are more points to the right of the center point of the distribution than there are to the left. Similarly, a negative value of skewness means there are more points to the left of the center point than there are to the right. In our use-case, a large positive value of skewness of the PDE residuals indicates that there are some asymmetrically high PDE residual values to the right.

**Kurtosis**: Kurtosis is the fourth central moment divided by the square of the variance after subtracting 3, defined as follows:

$$\text{kurtosis} = \frac{\sum_{i=1}^N (Y_i - \bar{Y})^4 / N}{s^4} - 3 \tag{30}$$

For a Normal distribution, Kurtosis is equal to 0. A positive value of Kurtosis indicates that there are more values in the tails of the distribution than what is expected from a Normal distribution. On the other hand, a negative value of Kurtosis indicates that there are lesser values in the tails of the distribution relative to a Normal distribution. In our use-case, a large positive value of Kurtosis of the PDE residuals indicates that there are some high PDE residual values occurring in very narrow regions of the spatio-temporal domain, that are being picked up as the heavy-tails of the distribution.

## I    HYPER-PARAMETER SETTINGS AND IMPLEMENTATION DETAILS

The hyper-parameter settings for the different baseline methods for every benchmark PDE are provided in Table 3. Note that we used the same network architecture and other hyper-parameter settings across all baseline method implementations for the same PDE. In this table, the column on 'r/ic/bc' represents the setting of the $\lambda_r, \lambda_{ic}, \lambda_{bc}$ hyper-parameters that are used to weight the different loss terms in the overall learning objective of PINNs. Table 3 also lists the type of Optimizer, learning rate (lr), and learning rate scheduler (lr.scheduler) used across all baselines for every PDE. For the Eikonal equation, we used the same modified multi-layer perceptron (MLP) architecture as the one proposed in (Wang et al., 2020). Additionally, for the Causal Evolutionary Sampling method, we used the following hyper-parameter settings across all PDEs: $\alpha = 5$, learning rate of the gate $\eta_g = 1e-3$, tolerance $\epsilon = 20$, initial value of $\beta = -0.5$, and $\Delta_{max} = 0.1$. The number of iterations (and the corresponding PDE coefficients for the Convection Equation) are provided in Section 5 of the main paper.

**Hardware Implementation Details**: We trained each of our models on one Nvidia Titan RTX 24GB GPU.

## J    ADDITIONAL DISCUSSION OF RESULTS

### J.1    VISUALIZING PROPAGATION FAILURE FOR DIFFERENT SETTINGS OF $\beta$

In Figure 2 of the main paper, we demonstrated the phenomenon of propagation failure for convection equation with $\beta = 50$, which was characterized by large values of Skewness and Kurtosis in the

Table 3: Hyper-parameter settings for different baseline methods for every benchmark PDE

| PDE | Method | Architecture | Periodic Encoding | r/ic/bc | Optimizer/ lr | lr. scheduler | RAR Hyperparams k/m/N/$|\mathcal{S}|$ |
|---|---|---|---|---|---|---|---|
| Convection | PINN, cPINN, Evo, CausalEvo | 50 x 4 (MLP) | No | 1/100/100 | Adam/ 1e-3 | StepLR rate=0.9 steps=5000 | N/A |
| | Curr. Reg. | 50 x 4 (MLP) | No | 1/1/1 | Adam/ 1e-4 | No | N/A |
| | RAR-G RAD RAR-D | 50 x 4 (MLP) | No | 1/100/100 | Adam/ 1e-3 | StepLR rate=0.9 steps=5000 | -/1/100/100000 1/1/100/100000 1/1/100/100000 |
| Allen Cahn | PINN, cPINN, Evo, CausalEvo | 128x4 (MLP) | Yes | 1/100/100 | Adam/ 1e-3 | StepLR rate=0.9 steps=5000 | N/A |
| | RAR-G RAD RAR-D | 128x4 (MLP) | Yes | 1/100/100 | Adam/ 1e-3 | StepLR rate=0.9 steps=5000 | -/1/100/100000 1/1/100/100000 1/1/100/100000 |
| Eikonal | PINN, Evo | 128x4 (Wang et al., 2020) (modified MLP) | No | 1/500/10 | Adam/ 1e-3 | StepLR rate=0.9 steps=5000 | N/A |

PDE residual fields for a large number of iterations (or epochs), and a simultaneous stagnation in the relative error values even though the mean PDE residual kept on decreasing. Here, in Figure 11, we show that the same phenomenon can be observed for other large values of $\beta > 10$, namely, $\beta = 30, 50, 70$. We can see that the relative errors for all these three cases remains high even though the PDE residual loss keeps on decreasing with iterations. We can also see that the absolute values of skewness and kurtosis increase as we increase $\beta$, indicating higher risks of propagation failure. In fact, for $\beta = 30$, we can even see that the epoch that marks an abrupt increase in skewness and kurtosis (around 50K iterations) also shows a sudden increase in the relative error at the same epoch, highlighting the connection between imbalanced PDE residuals and the phenomenon of propagation failure.

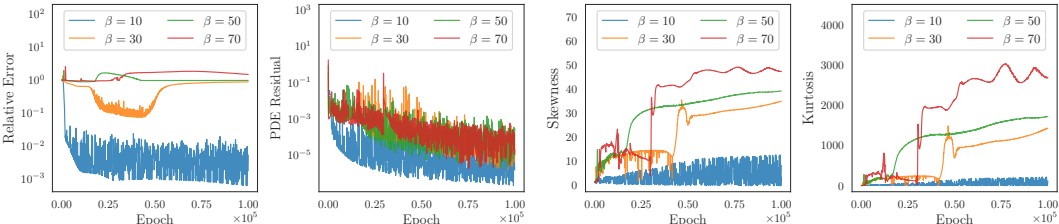

Figure 11: Demonstration of Propagation Failure for Different Settings of $\beta$ ($\beta = 10, 30, 50, 70$)

### J.2 CONVERGENCE SPEED OF EVOLUTIONARY SAMPLING

Figure 12 shows that Causal Evo is able to converge faster to low error solutions than all other baseline methods for both convection and Allen Cahn equations. This shows the importance of

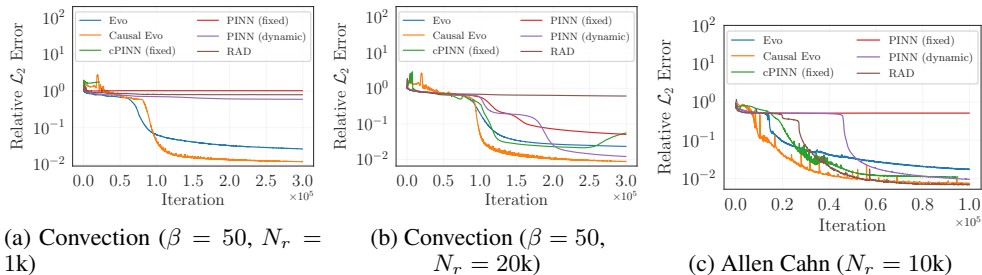

(a) Convection ($\beta = 50$, $N_r = 1k$)

(b) Convection ($\beta = 50$, $N_r = 20k$)

(c) Allen Cahn ($N_r = 10k$)

Figure 12: Comparing convergence speeds of baselines w.r.t. Evo and Causal Evo.

respecting causality along with focusing on high residual regions to ensure fast propagation of correct solution from initial/boundary points to interior points. While cPINN-fixed and PINN-dynamic do not converge for convection ($N_r = 1K$), we can see that they both converge to lower errors compared to PINN-fixed for Allen Cahn and for convection when $N_r$ is large (20$K$).

## J.3 SENSITIVITY OF RAR-BASED METHODS

Figures 13 shows the sensitivity of two RAR-based methods: RAR-G and RAR-D respectively on different values of the resampling period $K$ and the number of collocation points $m$ added from the dense set $\mathcal{P}_{dense}$. Note that although the size of the initial set of collocation points $|\mathcal{P}|$ was same for each of these experiments, the final size of the collocation points vary depending on the choice of $K$ (the final size of the collocation points increases as $k$ decreases) and $m$ (the final size of the collocation points increases with $m$). We essentially observe that adding more collocation points almost always improves the performance of these RAR-based methods.

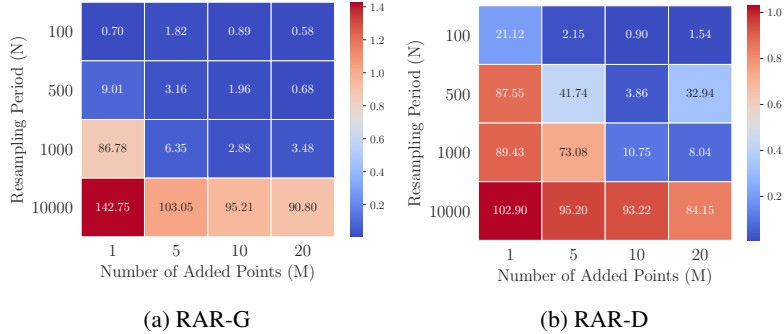

(a) RAR-G

(b) RAR-D

Figure 13: Sensitivity of RAR-based Methods on different values of $K$ and $M$ (for a fixed $|\mathcal{P}_{dense}| = 100k$ on Convection Equation with $\beta = 30$.) The numbers in the heatmap denote the % Relative $\mathcal{L}_2$ Error.

## J.4 VISUALIZING THE EVOLUTION OF COLLOCATION POINTS IN EVO

Figure 14 shows the evolution of collocation points and PDE residual maps of Evo as we progress in training iterations for the convection equation with $\beta = 50$. We can see that the retained population of Evo at every iteration (shown in red) selectively focuses on high PDE residual regions, while the re-sampled population (shown in blue) are generated from a uniform distribution. By increasing the contribution of high residual regions in the computation of the PDE loss, we can see that Evo is able to reduce the PDE loss over iterations without admitting high imbalance, thus mitigating the propagation failure mode, in contrast to conventional PINNs.

## J.5 VISUALIZING THE EVOLUTION OF COLLOCATION POINTS IN CAUSAL EVO

Figure 15 shows the evolution of collocation points and PDE residuals of Causal Evo, along with the dynamics of the Causal Gate function. We can see that the retained population at every iteration (shown in red) strictly adheres to the principle of causality such that the collocation points are sampled

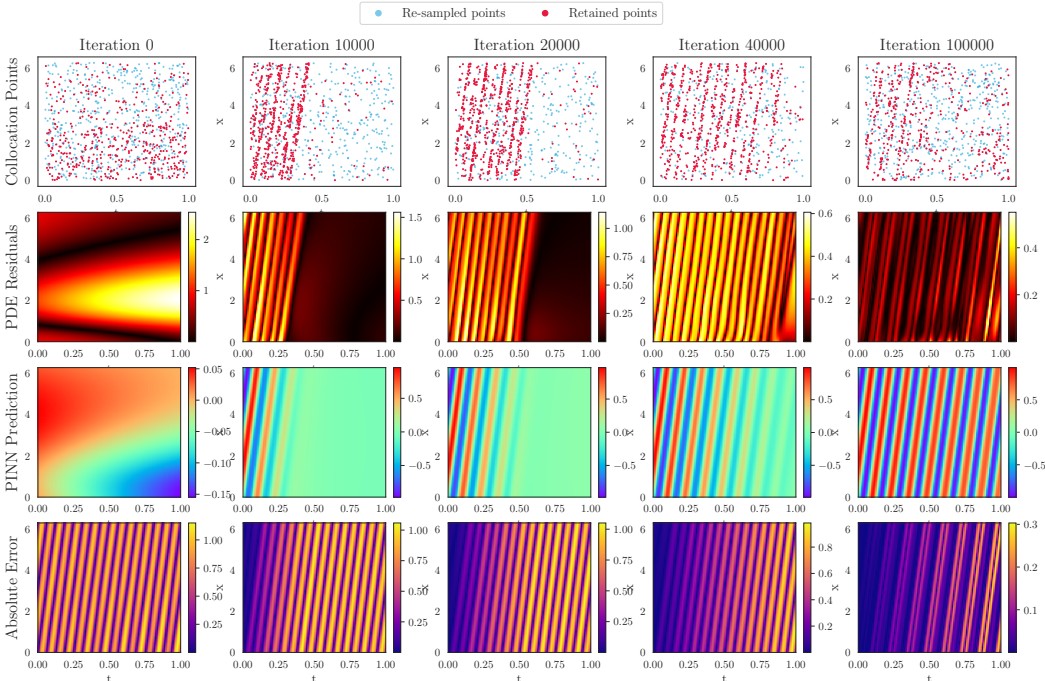

Figure 14: Demonstrating the propagation of information from the initial/boundary points to the interior points for Evo.Sample on Convection Equation($\beta = 50$)

from later times only when the PDE residuals at earlier times have been minimized. This is also reflected in the movement of the causal gate function where the gate values are close to 1 for only small portions of time domain at intermediate epochs. At 90K iterations, we can see that the causal gate values are close to 1 for all values of time, indicating that the entire time domain is now revealed for training PINN to converge to the correct solution.

### J.6 VISUALIZING THE DYNAMICS OF COLLOCATION POINTS FOR RAR-BASED METHODS

Figures 16, 17, annd 18 shows the evolution of collocation points and PDE residuals of RAR-G, RAD, and RAR-D, respectively. We can see that all three RAR-based methods are failing to converge to the correct solution even after 100K iterations, demonstrating their inabillity to mitigate propagation failures.

### J.7 KURAMOTO-SIVASHINSKY (KS) EQUATIONS

We used three additional experiments on the Kuramoto-Sivashinsky (KS) Equations (one for a regular relatively simple case and the remaining two exhibiting chaotic behavior). Please note that these equations are particularly more complex, especially the chaotic cases where a small change in the state of the solution can result in very large errors downstream in time. Thus, for chaotic domains, the successful propagation of solution from the initial and boundary conditions is critical to guarantee convergence. We would also like to highlight that the computational cost of these experiments are significantly higher. We used the exact same hyper-parameter settings as those provided in CPINN except the sample size, which was varied from 128 to 2048 in the KS-regular case, and the number of training iterations, which was kept as 300k in our proposed approaches while CPINN was allowed to use about 1 M maximum number of iterations with early stopping. Our method on average takes 50-60% less time than CPINN because of the significantly smaller number of iterations.

Figure 19 compares the performance of CPINN, Evo, and Causal Evo on the KS equation (regular case) as a function of the number of collocation points used in PINN training. We can see that both Evo and Causal Evo show improvements over CPINN when the number of collocation points is small ($N_r = 128$). As the number of collocation points is increased, Causal Evo shows better performance

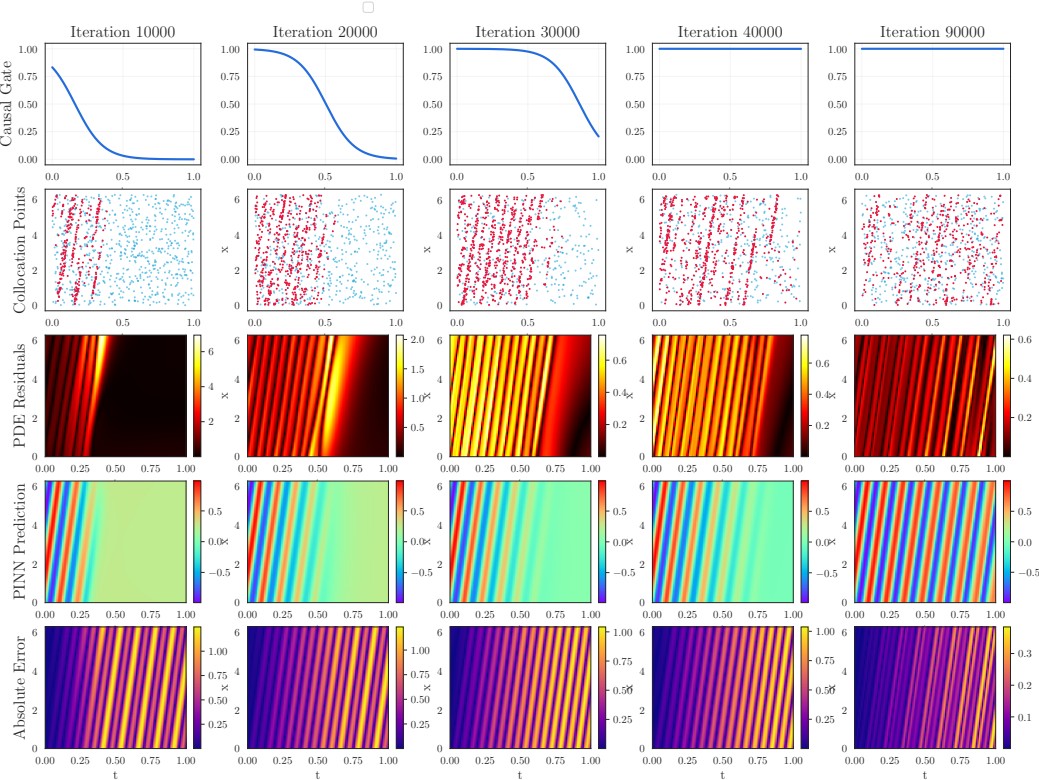

Figure 15: Demonstrating the propagation of information from the initial/boundary points to the interior points for Causal Evo. on Convection Equation($\beta = 50$)

than Evo, as it incorporates an additional prior of causality along with satisfying the four motivating properties of Evo. Overall, Causal Evo mostly performs better than both CPINN and Evo across different training set sizes. Note that these curves have been obtained using a single run of every method due to the computational cost of training for the KS equations, and having multiple runs for every method will help to quantify the variance in these results.

Table 4: Relative $\mathcal{L}_2$ errors (in %) of CausalPINN, Evo and CausalEvo over the three different KS-Equation Benchmarks Wang et al. (2022b).

|  | CausalPINN | Evo (ours) | CausalEvo (ours) |
|---|---|---|---|
| Regular | 2.120% | 3.740% | **0.761**% |
| Chaotic | **3.272**% | 6.924% | 7.630% |
| Chaotic - Extended | 52.66% | **29.26**% | 33.50% |

Table 4 compares the performance of CPINN, Evo, and Causal Evo on the three KS-Equations cases as was used in the original CPINN paper. Please note that for these experiments, we used the exact same hyper-parameter settings as the original CPINN, thus a large number of collocation points were used (2048 for the regular case and 8192 for the two chaotic regimes for each time-window). We can observe that on the regular case, Evo performs similarly to CPINN, while Causal Evo is significantly better than both. However, in the first chaotic case, CPINN is slightly better than both Evo and Causal Evo. Finally, in the much more chaotic regime for the KS-Equation (extended case), we find that all of the methods struggle to obtain a high fidelity solution of the field. However, Evo and Causal Evo are somewhat better than CPINNs. Hence, we can comment that Evo and CausalEvo have comparable performance to CPINN on their benchmark settings. Additional visualizations of

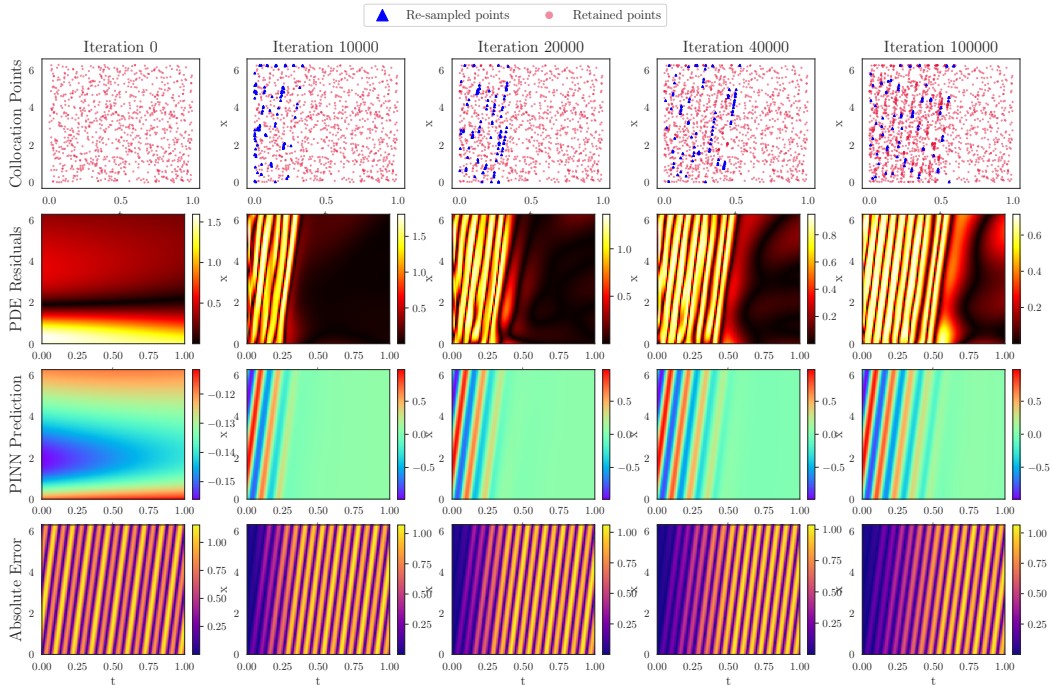

Figure 16: Demonstrating the dynamic changes in the collocation points for RAR-G on Convection Equation($\beta = 50$)

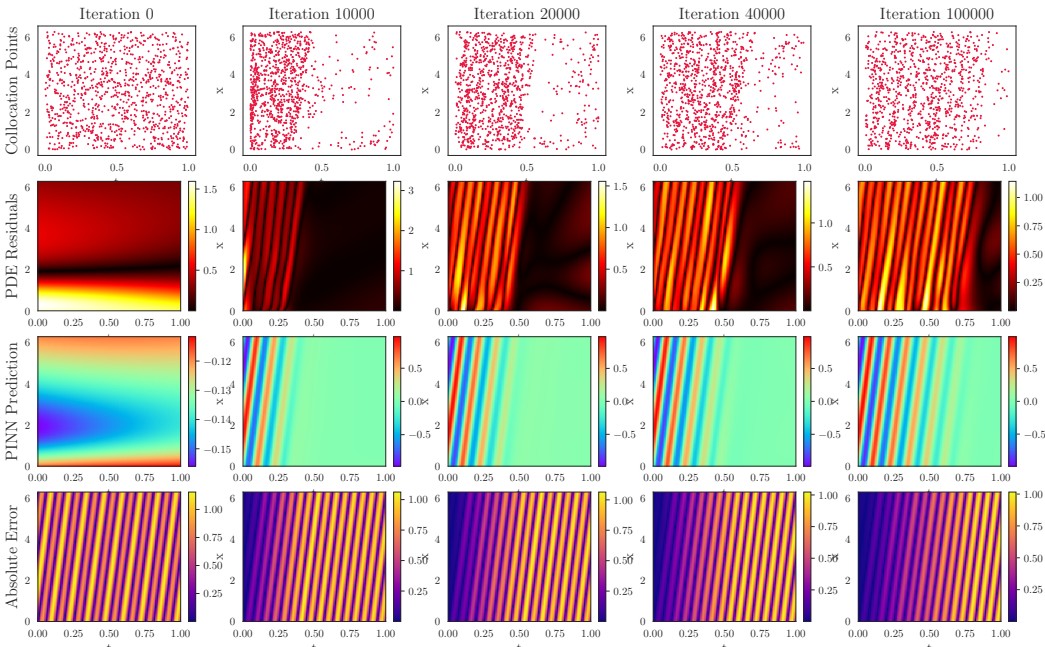

Figure 17: Demonstrating the dynamic changes in the collocation points for RAD on Convection Equation($\beta = 50$)

the solutions of comparative methods on the three KS equations cases are provided in Figures 20, 21, and 22.

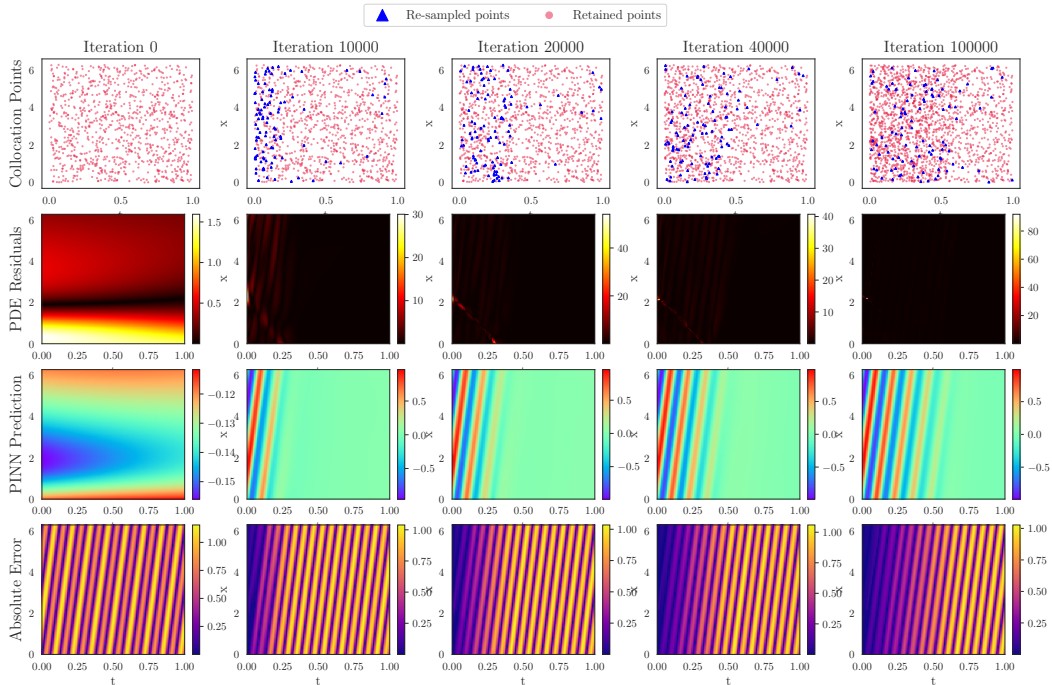

Figure 18: Demonstrating the dynamic changes in the collocation points for RAR-D on Convection Equation($\beta = 50$)

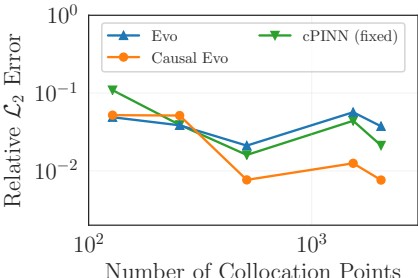

Figure 19: Comparison of Evo, CausalEvo and CausalPINNs on Simple KS Equation with varying number of collocation points.

### J.8 ADDITIONAL DISCUSSION AND VISUALIZATION FOR THE EIKONAL EQUATION

We chose to solve 2D Eikonal Equations for complex arbitrary surface geometries as they represent particularly hard PDE problems that are susceptible to PINN failure modes. In these problems, we are given the zero contours of the equation on the boundaries (representing the outline of the 2D object), which can take arbitrary shapes. The goal is to correctly propagate the boundary conditions to obtain the unique target solution where the interior is negative and the exterior is positive. Here, any small error in propagation from the boundaries can lead to cascading errors such that a large segment of the predicted field can have opposite signs compared to the ground-truth, even though their PDE residuals are close to 0. Since Evo is explicitly designed to break propagation barriers and thus enable easy transmission of the solution from the boundary to the interior/exterior points, we can see that it shows significantly better performance. On the other hand, PINN (fixed) and PINN (dynamic) struggle to converge to the correct solution especially for complex geometries (e.g., the 'gear') because of the inherent challenge in sampling an adequate number of points from arbitrary shaped object boundaries exhibiting highly imbalanced residuals.

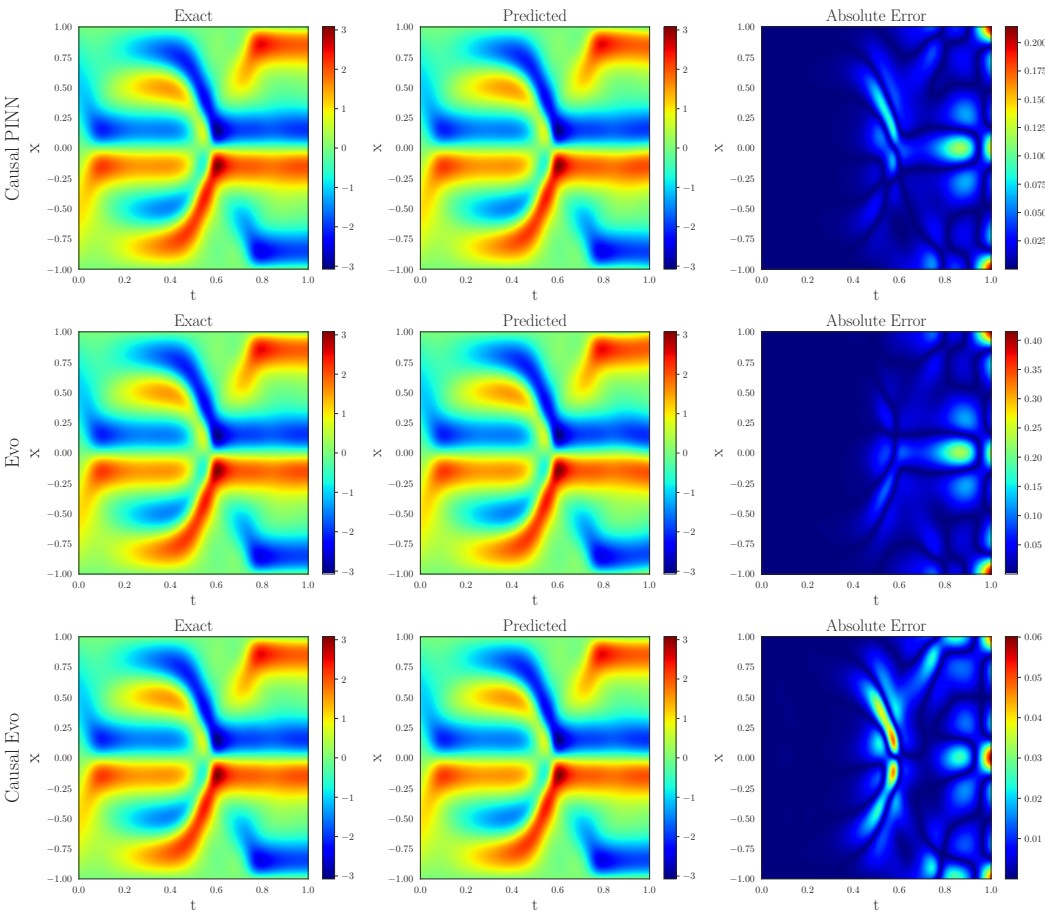

Figure 20: Visualization of the Evo, Causal Evo and CausalPINN on KS-Equation (Regular Case).

In Figure 23, we show the evolution of the solutions of comparative methods for the 'gear' case over iterations. We can see that Evo is able to resolve the high residual regions better than the baselines, and thus encounter less incorrect "sign flips" compared to the ground-truth, even in the early iterations of PINN training.

## K  OPTIMIZATION CHARACTERISTICS OF EVOLUTIONARY SAMPLING ON TEST OPTIMIZATION FUNCTIONS

In this section, we demonstrate the ability of our proposed Evolutionary Algorithm to find global minimas on various test optimization functions. We will also provide other characterizations of our proposed Evolutionary Sampling algorithm.

### K.1  AUCKLEY FUNCTION

The two-dimensional form of the Auckley function has multiple local maximas in the near-flat region of the function and one large peak at the center.

$$f(x) = a\exp\left(-b\sqrt{\frac{1}{d}\sum_{i=1}^{d}x_i^2}\right) + \exp\left(\frac{1}{d}\sum_{i=1}^{d}\cos(cx_i)\right) + a + \exp(1) \tag{31}$$

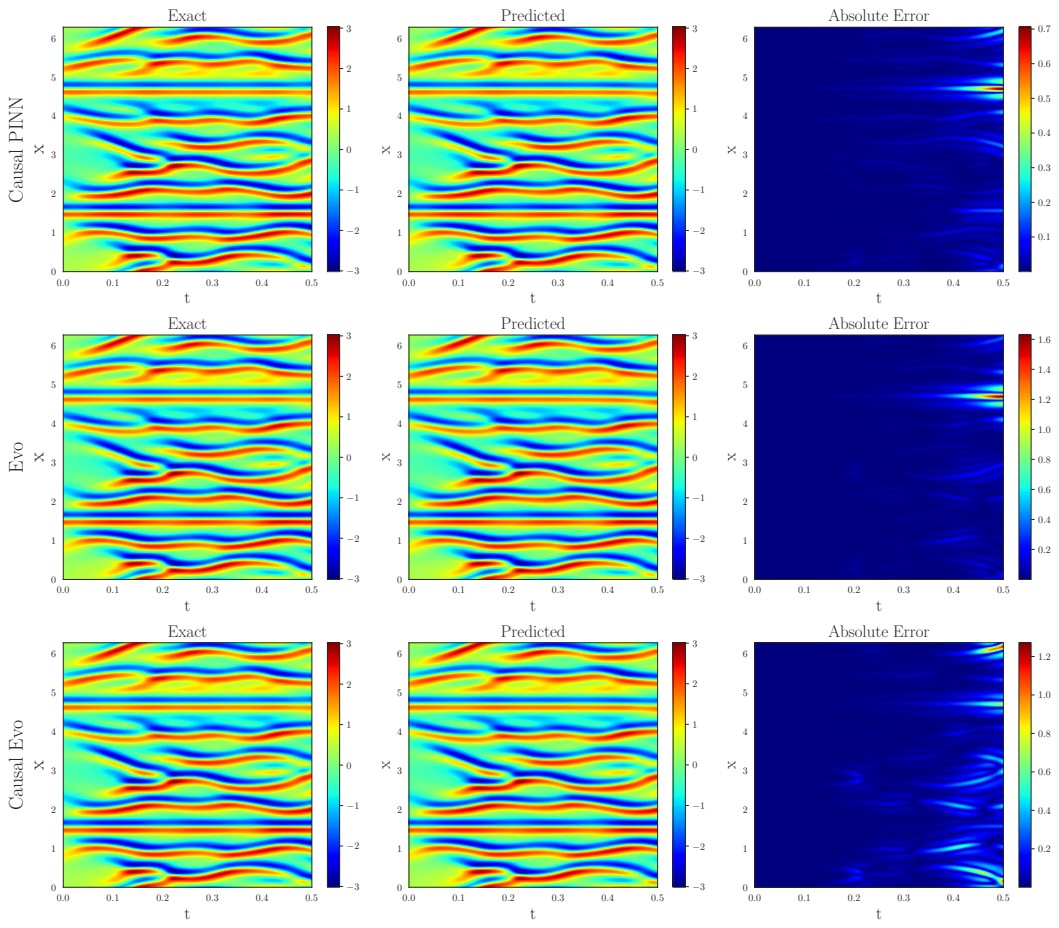

Figure 21: Visualization of the Evo, Causal Evo and CausalPINN on KS-Equation (Chaotic Case).

## K.2 BOHACHEVSKY FUNCTION

The two-dimensional form of the Bohachevsky function is a bowl shaped function having one global maxima.

$$f(x, y) = -x^2 - 2y^2 + 0.3\cos(3\pi x) + 0.4cos(4\pi y) - 0.7 \tag{32}$$

## K.3 DROP-WAVE FUNCTION

The two-dimensional form of the Drop-Wave function which is multimodal and highly complex.

$$f(x, y) = \frac{1 + \cos(12\sqrt{x^2 + y^2})}{0.5(x^2 + y^2) + 2} \tag{33}$$

## K.4 EGG-HOLDER FUNCTION

The two-dimensional form of the Egg-Holder function is highly complex function that is difficult to optimize because of the presence of multiple local maximas.

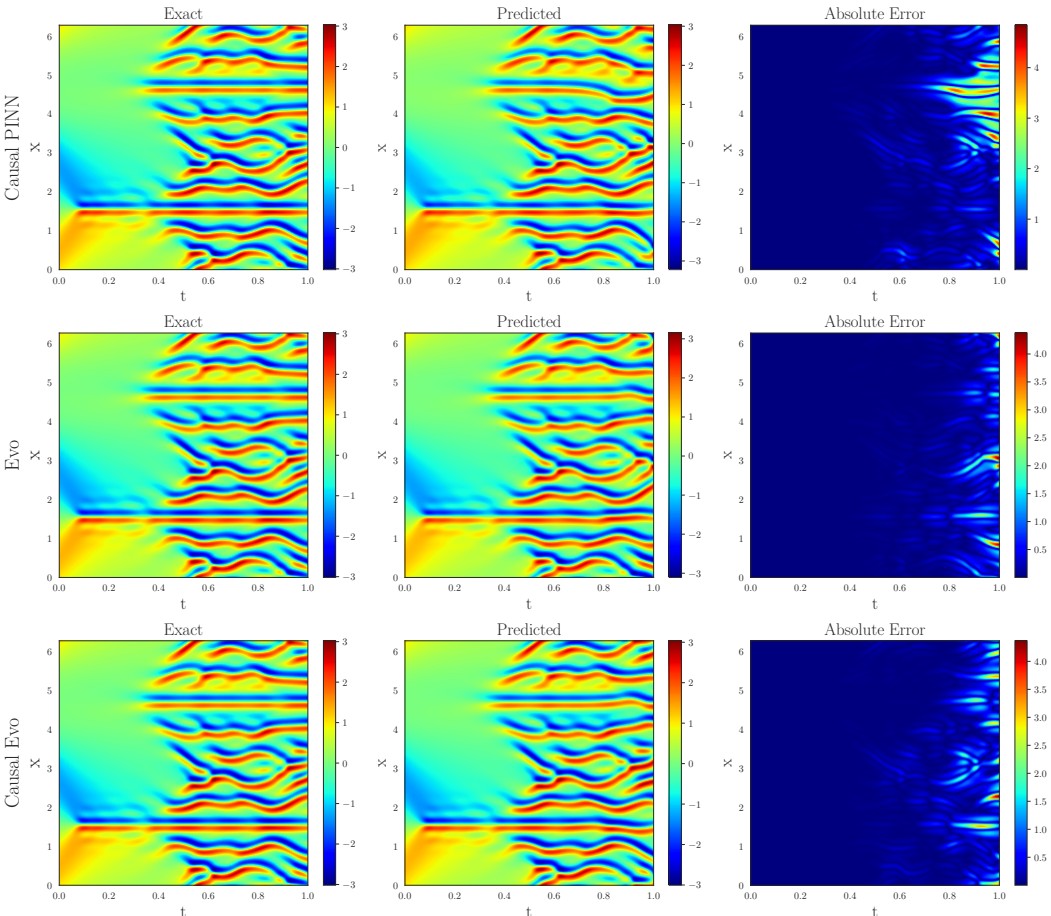

Figure 22: Visualization of the Evo, Causal Evo and CausalPINN on KS-Equation (Extended Chaotic Case).

$$f(x, y) = (y + 47)\sin(\sqrt{|y + \frac{x}{2} + 47|}) + x \sin(\sqrt{|x - (y + 47)|}) \qquad (34)$$

### K.5 HOLDER-TABLE FUNCTION

The two-dimensional form of the Holder-Table function has many local maximas, but has 4 global maximas at the four corners.

$$f(x, y) = |\sin(x)\cos(y)\exp\left(|1 - \frac{\sqrt{x^2 + y^2}}{\pi}|\right)| \qquad (35)$$

### K.6 BUKIN FUNCTION

The two-dimensional form of the Bukin function has many local maximas, all of which lie on a ridge.

$$f(x, y) = -100\sqrt{|y - 0.01x^2|} - 0.01|x + 10| \qquad (36)$$

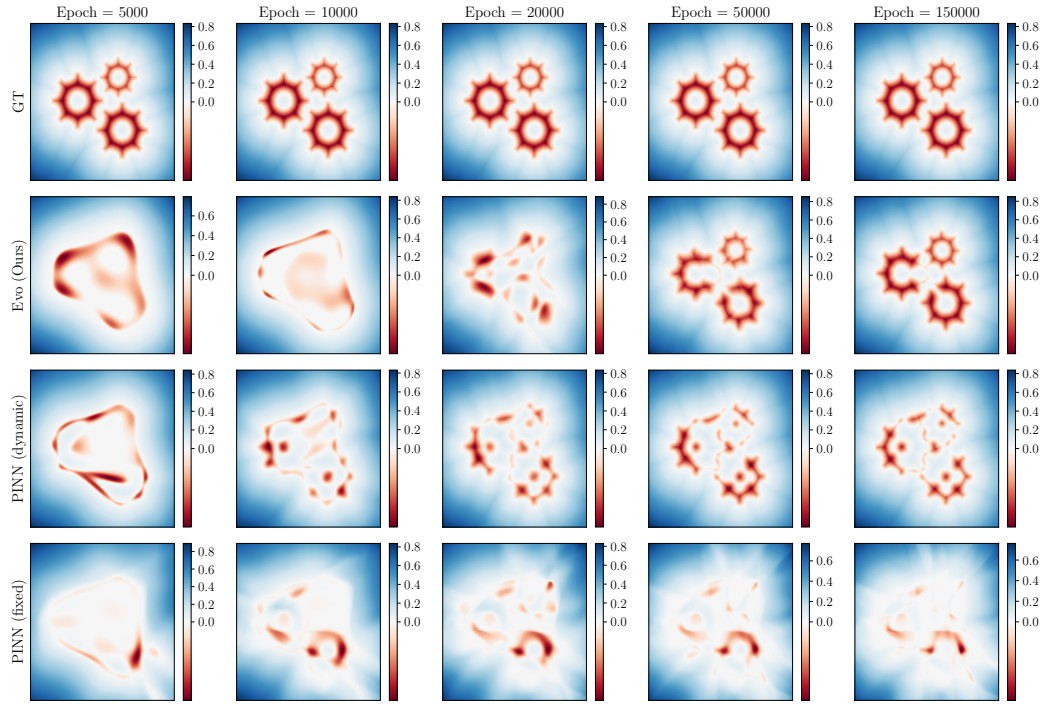

Figure 23: The predicted solutions of Eikonal equation (Figure 7) at different iterations during training.

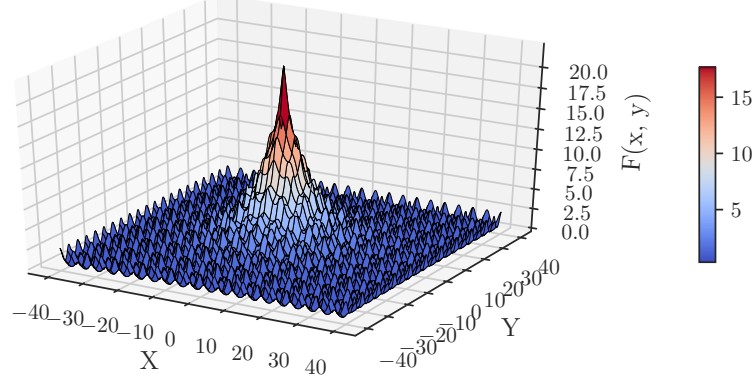

Figure 24: Surface Plot of the 2-D Auckley Function

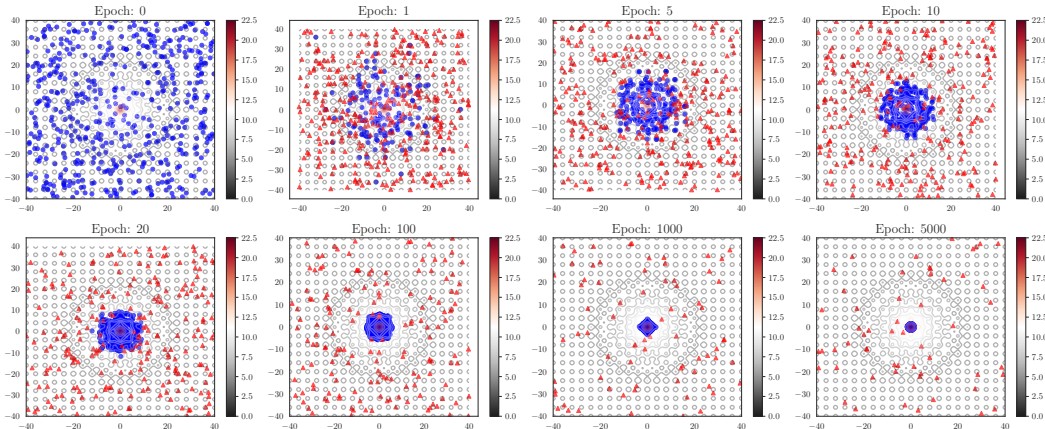

Figure 25: Demonstrating the evolution of the randomly initialized points while optimizing the Auckley function. The red triangles represent the re-sampled population at that epoch, and the blue dots represent the retained population at that epoch. The contour function of the objective function is shown in the background.

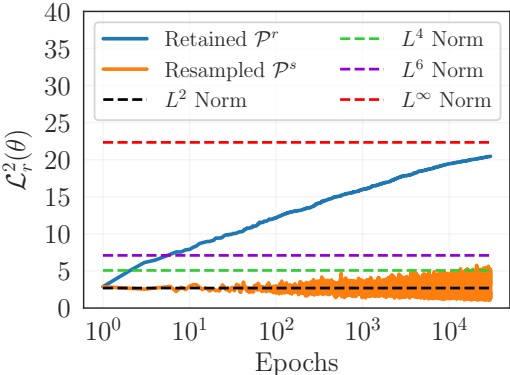

Figure 26: Illustrating the dynamic behavior of the Evolutionary Sampling algorithm on Auckley Function using the $L^2$ Physics-informed Loss computed on the retained and re-sampled populations. The horizontal lines represent the $L^p$ Physics-informed Loss on a dense set of uniformly sampled collocation points (where $p = 2, 4, 6, \infty$).

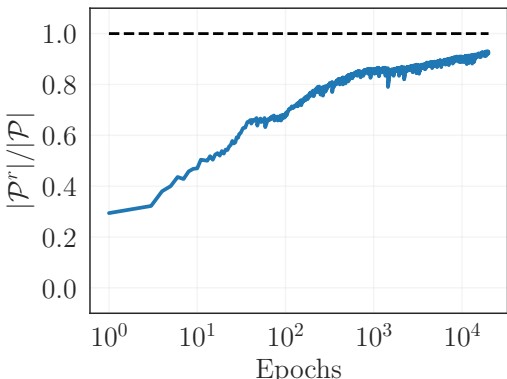

Figure 27: Demonstrating the dynamic evolution of the retained population size over epochs on the Auckley Function.

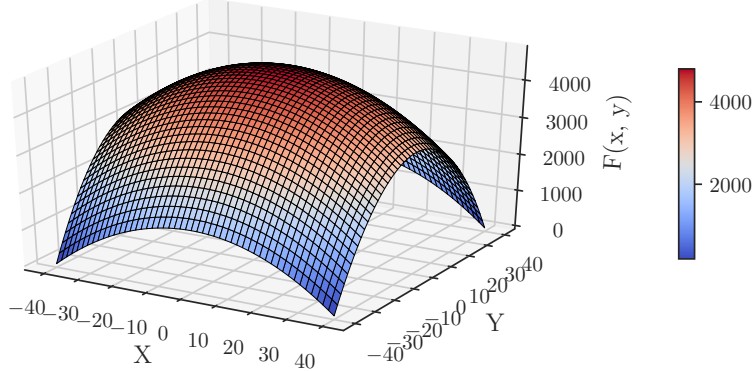

Figure 28: Surface Plot of the 2-D Bohachevsky Function

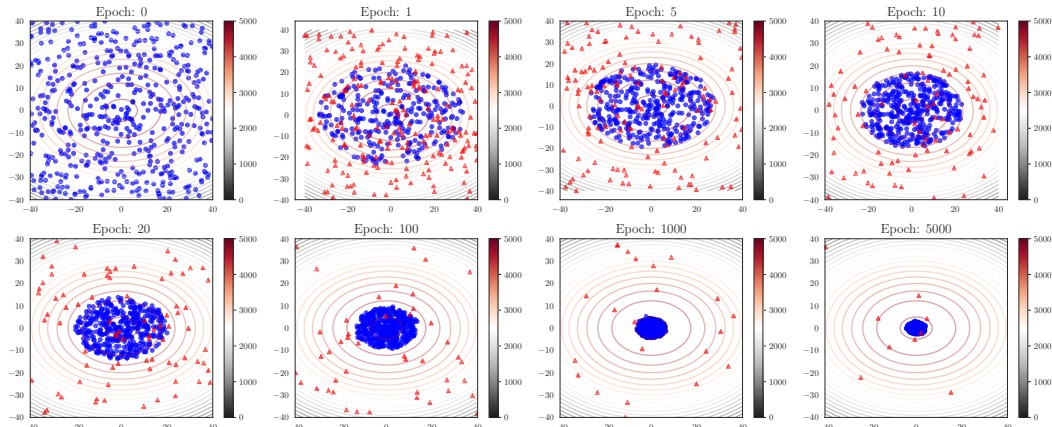

Figure 29: Demonstrating the evolution of the randomly initialized points while optimizing the Bohachevsky function. The red triangles represent the re-sampled population at that epoch, and the blue dots represent the retained population at that epoch. The contour function of the objective function is shown in the background.

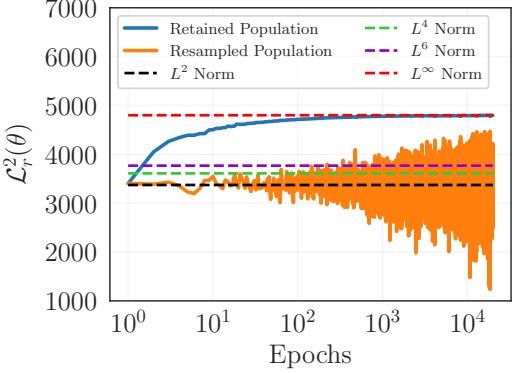

Figure 30: Illustrating the dynamic behavior of the Evolutionary Sampling algorithm on Bohachevsky Function using the $L^2$ Physics-informed Loss computed on the retained and re-sampled populations. The horizontal lines represent the $L^p$ Physics-informed Loss on a dense set of uniformly sampled collocation points (where $p = 2, 4, 6, \infty$).

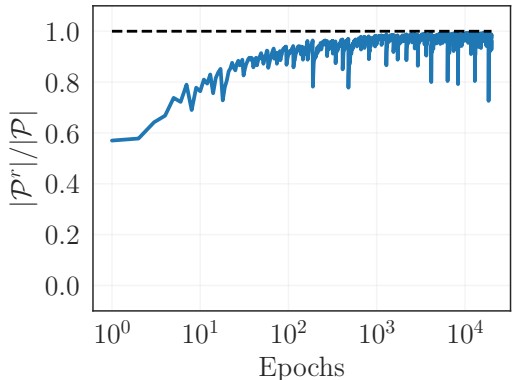

Figure 31: Demonstrating the dynamic evolution of the retained population size over epochs on the Bohachevsky Function.

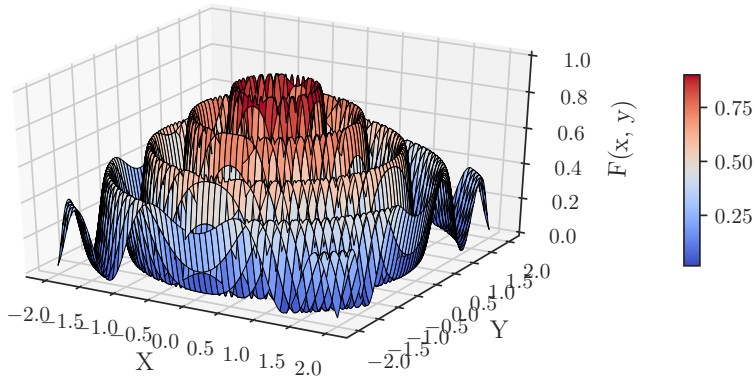

Figure 32: Surface Plot of the 2-D Drop-Wave Function

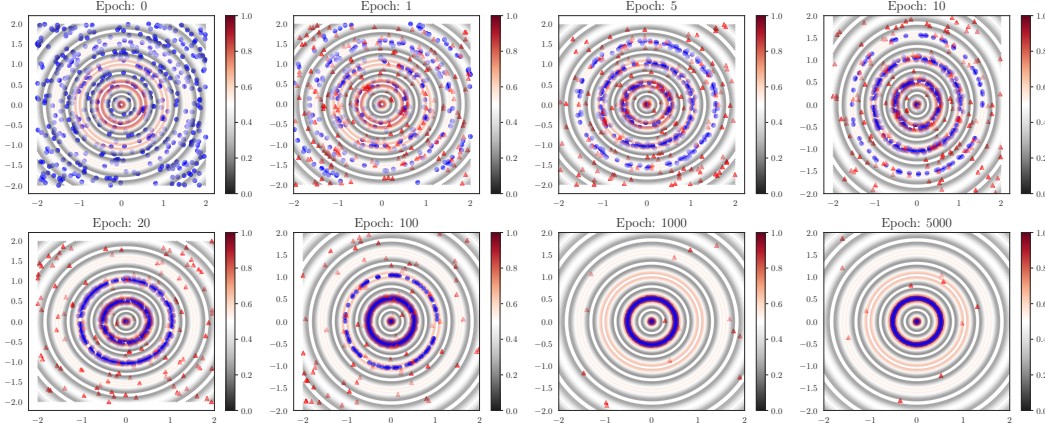

Figure 33: Demonstrating the evolution of the randomly initialized points while optimizing the Drop-Wave function. The red triangles represent the re-sampled population at that epoch, and the blue dots represent the retained population at that epoch. The contour function of the objective function is shown in the background.

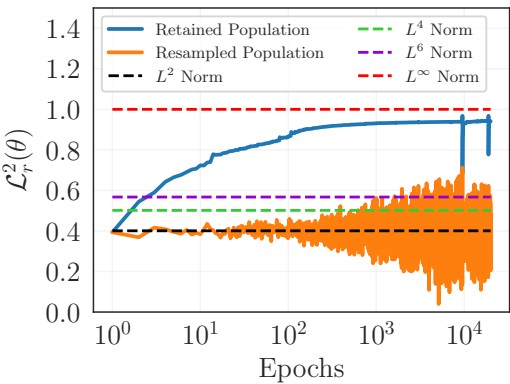

Figure 34: Illustrating the dynamic behavior of the Evolutionary Sampling algorithm on Drop-Wave Function using the $L^2$ Physics-informed Loss computed on the retained and re-sampled populations. The horizontal lines represent the $L^p$ Physics-informed Loss on a dense set of uniformly sampled collocation points (where $p = 2, 4, 6, \infty$).

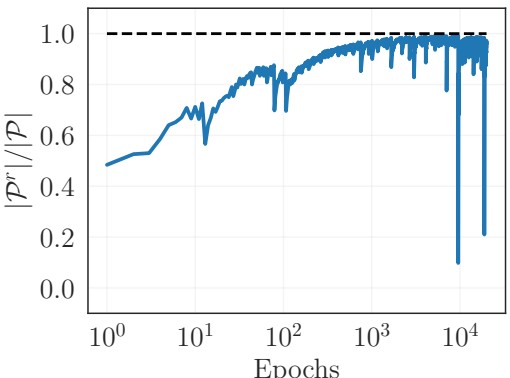

Figure 35: Demonstrating the dynamic evolution of the retained population size over epochs on the Drop-Wave Function.

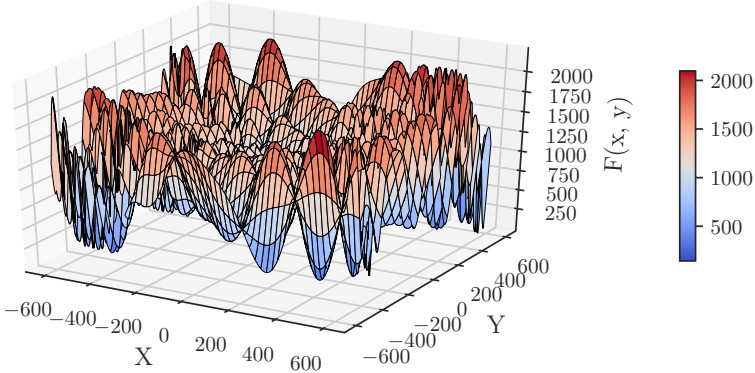

Figure 36: Surface Plot of the 2-D Egg-Holder Function

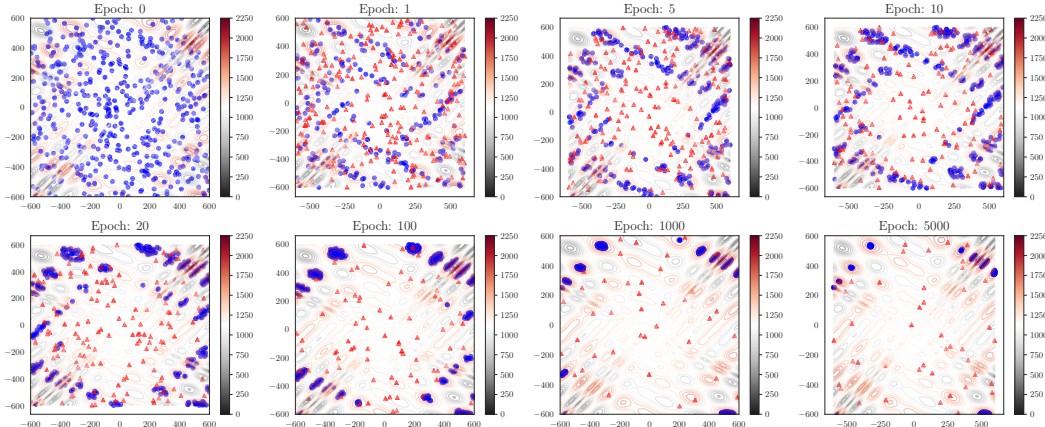

Figure 37: Demonstrating the evolution of the randomly initialized points while optimizing the Egg-Holder function. The red triangles represent the re-sampled population at that epoch, and the blue dots represent the retained population at that epoch. The contour function of the objective function is shown in the background.

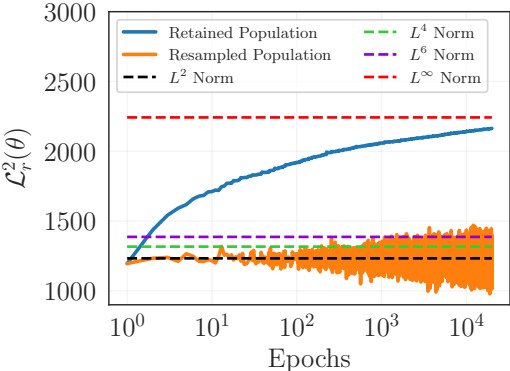

Figure 38: Illustrating the dynamic behavior of the Evolutionary Sampling algorithm on Egg-Holder Function using the $L^2$ Physics-informed Loss computed on the retained and re-sampled populations. The horizontal lines represent the $L^p$ Physics-informed Loss on a dense set of uniformly sampled collocation points (where $p = 2, 4, 6, \infty$).

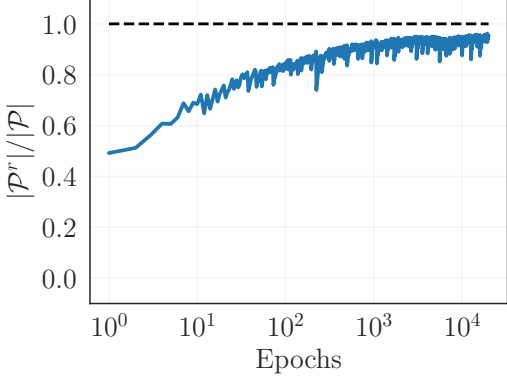

Figure 39: Demonstrating the dynamic evolution of the retained population size over epochs on the Egg-Holder Function.

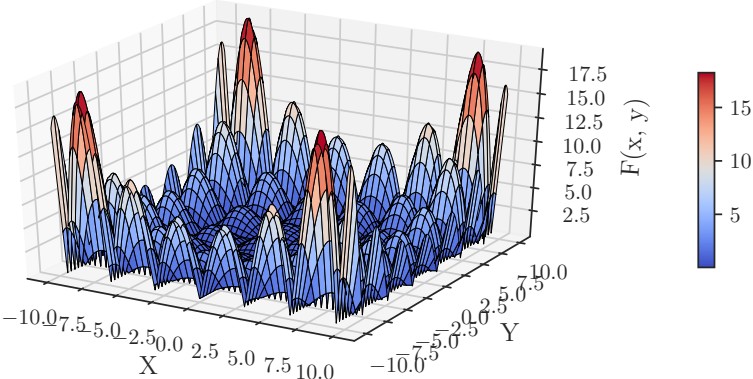

Figure 40: Surface Plot of the 2-D Holder-Table Function

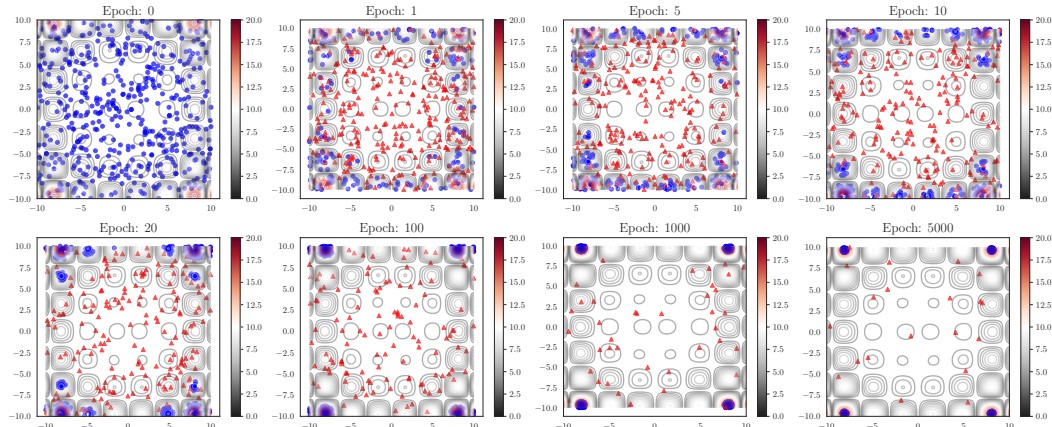

Figure 41: Demonstrating the evolution of the randomly initialized points while optimizing the Holder-Table function. The red triangles represent the re-sampled population at that epoch, and the blue dots represent the retained population at that epoch. The contour function of the objective function is shown in the background.

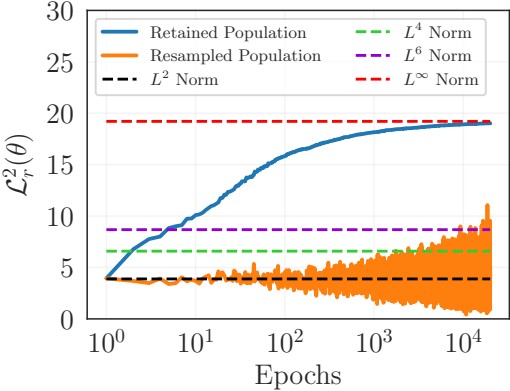

Figure 42: Illustrating the dynamic behavior of the Evolutionary Sampling algorithm on Holder-Table Function using the $L^2$ Physics-informed Loss computed on the retained and re-sampled populations. The horizontal lines represent the $L^p$ Physics-informed Loss on a dense set of uniformly sampled collocation points (where $p = 2, 4, 6, \infty$).

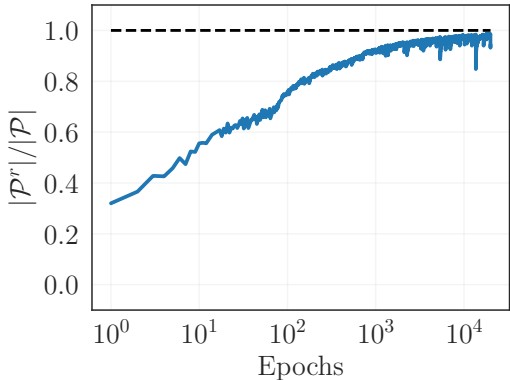

Figure 43: Demonstrating the dynamic evolution of the retained population size over epochs on the Holder-Table Function.

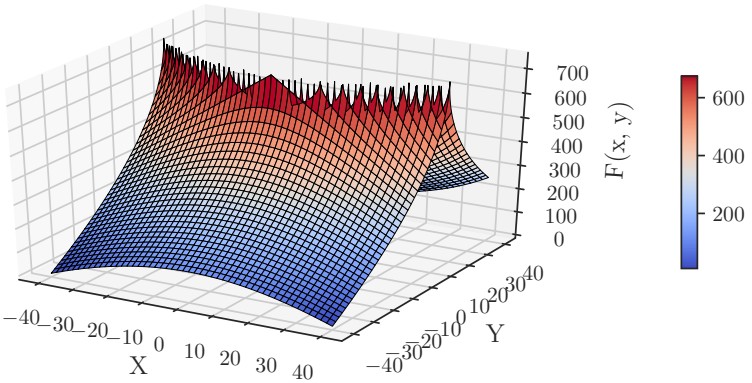

Figure 44: Surface Plot of the 2-D Bukin Function

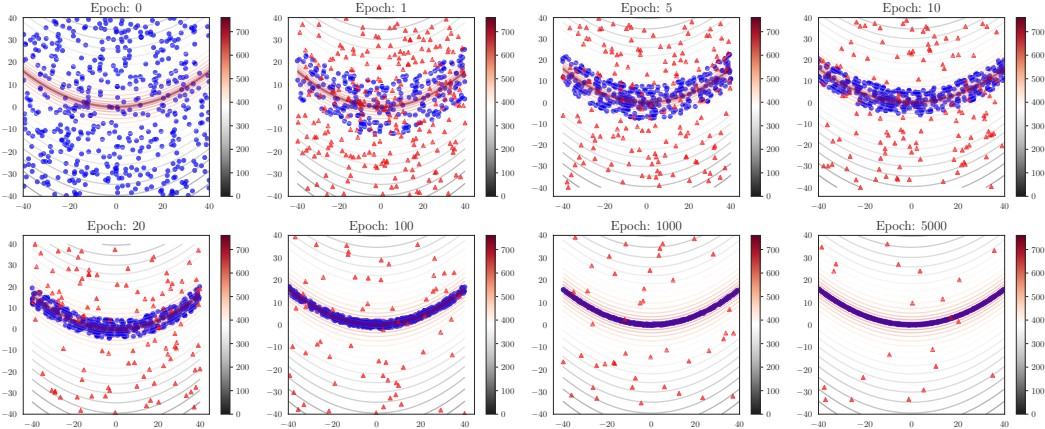

Figure 45: Demonstrating the evolution of the randomly initialized points while optimizing the Bukin function. The red triangles represent the re-sampled population at that epoch, and the blue dots represent the retained population at that epoch. The contour function of the objective function is shown in the background.

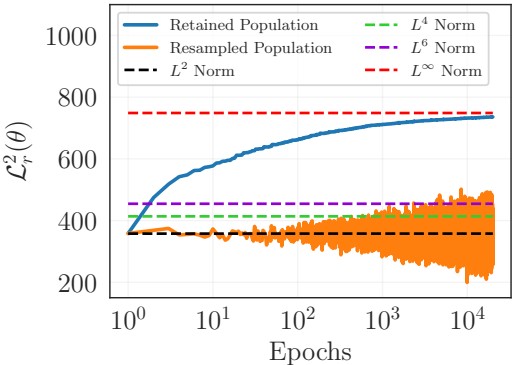

Figure 46: Illustrating the dynamic behavior of the Evolutionary Sampling algorithm on Bukin Function using the $L^2$ Physics-informed Loss computed on the retained and re-sampled populations. The horizontal lines represent the $L^p$ Physics-informed Loss on a dense set of uniformly sampled collocation points (where $p = 2, 4, 6, \infty$).

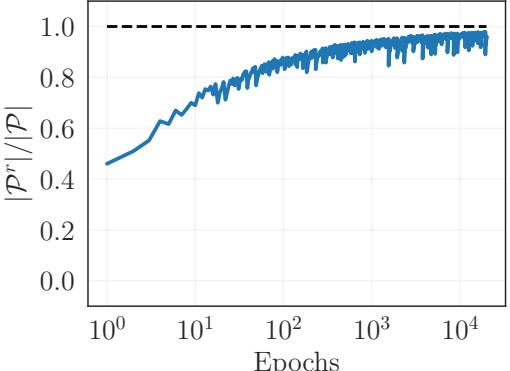

Figure 47: Demonstrating the dynamic evolution of the retained population size over epochs on the Bukin Function.

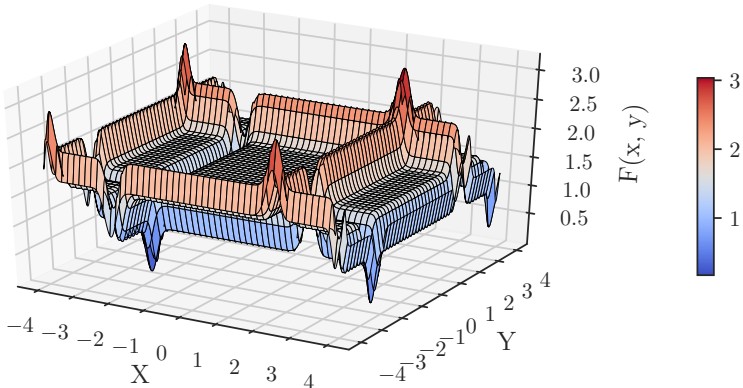

Figure 48: Surface Plot of the 2-D Michalewicz Function

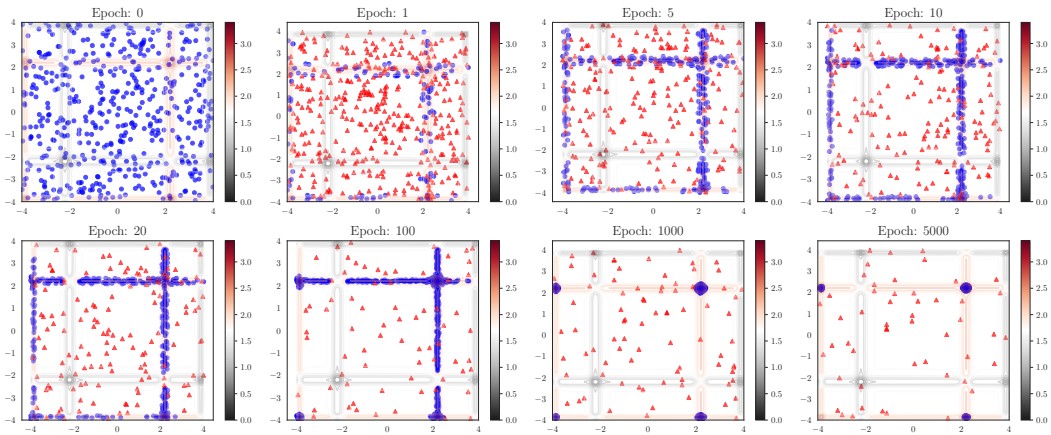

Figure 49: Demonstrating the evolution of the randomly initialized points while optimizing the Michalewicz function. The red triangles represent the re-sampled population at that epoch, and the blue dots represent the retained population at that epoch. The contour function of the objective function is shown in the background.

### K.7 MICHALEWICZ FUNCTION

The two-dimensional form of the Michalewicz function has multiple ridges and valleys which are very steep.

$$f(x) = \sum_{i=1}^{d} \sin(x_i) \sin^{2m}\left(\frac{ix_i^2}{\pi}\right) \tag{37}$$

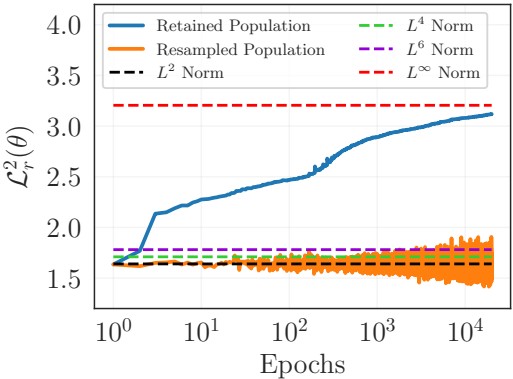

Figure 50: Illustrating the dynamic behavior of the Evolutionary Sampling algorithm on Michalewicz Function using the $L^2$ Physics-informed Loss computed on the retained and re-sampled populations. The horizontal lines represent the $L^p$ Physics-informed Loss on a dense set of uniformly sampled collocation points (where $p = 2, 4, 6, \infty$).

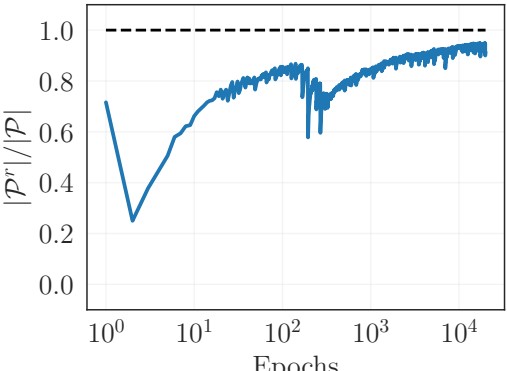

Figure 51: Demonstrating the dynamic evolution of the retained population size over epochs on the Michalewicz Function.

