# OpenReview forum: "Mitigating Propagation Failures in PINNs using Evolutionary Sampling"
_ICLR.cc/2023/Conference — Submitted to ICLR 2023_

### Official Review · Reviewer_rJ3m · 2022-10-17

**Confidence:** 4
**Correctness:** 3
**Technical Novelty And Significance:** 3
**Empirical Novelty And Significance:** 3
**Recommendation:** 8

**Clarity, Quality, Novelty And Reproducibility:**

The paper is clearly written and of high quality. The novelty is somewhat limited by the fact that several sampling schemes have been proposed so far, and that the experimental evaluation is limited to only two systems (one with two different parameterizations). Code and datasets are available, which makes the paper reproducible.

**Strength And Weaknesses:**

The paper is excellently written, and the arguments for the Evo strategy are clear. The approach is simple, intuitive, and yet effective. The experimental evidence is good, although I would appreciate results with more benchmarks.

The two weaknesses I see are that 1) the benefit of Causal Evo is not shown or discussed in the experiments, and 2) that comparisons with some parts of the literature are missing entirely or difficult to evaluate.

## Ad 1)
I understand Evo as a method that mitigated training failures, rather than one that improves performance for a successfully trained PINN. Thus, from that perspective, Evo and Causal Evo perform equivalently in the shown experiments. It would be interesting to see an example where Evo fails because causality is not respected, but where Causal Evo successfully trains the PINN.

## Ad 2)
I am missing other approaches that have been proposed to mitigate training failures. These approaches include domain decomposition, loss weighting schemes, the architecture from arXiv:2109.09338, or the regularization term proposed in arXiv:2112.05620. It would further be interesting to see if and how the sampling scheme can be combined with the former methods. Further, the comparison is difficult to evaluate because the chosen benchmarks show little overlap with what has been proposed in the literature. For example, CPINN was evaluated in the original paper on a set of benchmarks that does not intersect with the benchmarks chosen in this paper. I would be interested to see how Evo performs on these benchmarks.

## Minor
- In the paragraph "Propagation Failure", the authors argue that trivial solutions can become attractive for PINNs. This has been made precise theoretically in arXiv:2109.09338 and experimentally in arXiv:2203.13648.
- References to the appendix are broken.
- The computational overhead of the comparison methods is not clear ("Challenges with Potential Remedies"). Similarly, for Evo it is not clear how $\mathcal{P}_i^s$ changes throughout training (i.e., is it of fixed size, or is rather $\mathcal{P}_i$ of fixed size?).
- The four salient properties seem somewhat artificial and post-hoc. For example, the gradual accumulation property is not well justified. As an alternative property, it seems equally intuitive that initially the collocation points should concentrate around the initial and boundary conditions (ensuring successful learning locally, from which propagation can continue), while at later stages the collocation points should be uniform (ensuring uniform accuracy across the computational domain). Thus, a priori, the CPINN sampling strategy seems equally powerful to mitigate propagation failures as Evo. It is even more interesting to see that the CPINN fail for the systems for which Evo succeeds, and I would appreciate an evaluation of Evo on the benchmarks that were chosen by the original authors of CPINN.

**Summary Of The Paper:**

The paper proposes a novel sampling strategy for collocation points to improve the training performance of PINNs. The strategy is based on the propagation hypothesis, which suggests that information from the boundary of the computational domain has to "propagate" to its interior, and that this propagation sometimes fails during PINN training. The authors show that the approach indeed leads to more stable training.

**Summary Of The Review:**

A clear paper with an intuitively appealing, simple, and yet effective method to mitigate training failures in PINNs. To judge if Evo is a major contribution towards PINN training requires more evidence.

*EDIT:* Raised my score according to the discussion with the authors.

---

> ### Author Response · Authors · 2022-11-19
> **Author Response for Reviewer rJ3m - Part 1**
>
> **Comment 1 [Missing References]:** I am missing other approaches that have been proposed to mitigate training failures. These approaches include domain decomposition, loss weighting schemes, the architecture from arXiv:2109.09338, or the regularization term proposed in arXiv:2112.05620. It would further be interesting to see if and how the sampling scheme can be combined with the former methods.
>
> **Response:** We thank the reviewer for pointing us to these papers that are relevant to our work, which we have now included in the paper. Note that failure modes in PINNs have been studied and characterized from a wide variety of complementary perspectives as described in the paragraph “Prior Work on Characterizing Failure Modes of PINNs” in Section 2. This includes studies focusing on the effect of imbalanced gradients, poor initializations, and adaptive weighting coefficients of loss terms on the occurrence of PINN failure modes. In this work, we specifically focus on the effect of sampling collocation points on PINN failure modes from the novel perspective of the propagation hypothesis. We agree with the reviewer that combining these different perspectives of PINN failure modes will be very interesting to explore. However, such an exploration is outside the scope of this paper, which can be focussed on in future works.
>
> Regarding the Arxiv paper arXiv:2109.09338 suggested by the reviewer, it is interesting to see that this work added a regularization loss term to penalize sharp changes in PDE residual fields, which is somewhat related to the idea of avoiding highly imbalanced PDE residuals motivating our work. However, note that without performing adaptive sampling, it is difficult to adequately represent regions with sharp PDE changes in the set of collocation points, especially when the sample size is small.
>
>
> **Comment 2 [Choice of PDEs and Comparison with CPINN]:** Further, the comparison is difficult to evaluate because the chosen benchmarks show little overlap with what has been proposed in the literature. For example, CPINN was evaluated in the original paper on a set of benchmarks that does not intersect with the benchmarks chosen in this paper. I would be interested to see how Evo performs on these benchmarks.”
>
> **Response:**
>
> Our choice of PDEs (Convection equation and Allen Cahn) was primarily motivated to have the same set of benchmark PDEs as those used in previous literature, to allow for easy comparison. For example, we borrowed the convection equation from the Curriculum Regularization paper [1] (NeurIPS 2021) with the exact same PDE parameters, PINN hyper-parameters, and experiment settings (except the number of collocation points). It was shown in [1] that large values of $\beta$ could be difficult to optimize for PINNs. Hence, we used $\beta=30$ and $\beta=50$. On the other hand, we borrowed the Allen Cahn Equation from the CausalPINN paper [2], which is currently only available as an Arxiv version. In [2], it was shown that Allen Cahn can fail to converge if fewer collocation points were used. The comprehensive study on RAR-based methods [3,4] also evaluated their method on the Allen Cahn Equation.
>
> Note that due to the diversity of physical phenomena represented by PDEs, experiments in the domain of PINNs are very diverse and there is a lack of standard benchmarks as is common in other AI/ML fields such as computer vision. Most of the papers involving PINNs use a different set of PDEs with varying set of hyper-parameters, depending on the specific target problem they are trying to solve. To consolidate the diversity of PDEs explored in related works, we have prepared a table of PDE datasets used in 10 representative papers in the area of PINNs that we have included in our related work (including the ones suggested by the reviewer). In our work, we used the same training and hyper-parameters setups as those reported in previous baselines to perform direct comparisons. For example, for both Convection Equation and the Allen Cahn Equation, we have used the same setting of neural network architecture, initial conditions, PDE coefficients, and number of training iterations as the existing methods, without fine-tuning our hyper-parameters on any problem. Hence, our method can be thought of as a “plug-in” that was added to existing evaluation frameworks.

---

> > ### Author Response · Authors · 2022-11-19
> > **Author Response for Reviewer rJ3m - Part 2**
> >
> > **Table to compare the Benchmark PDEs used for each representative paper:**
> >
> > |                              | Curr.  Reg. [1] | CPINN [2] | DeepXDE [3]  | RAR-D [4] | Sine  Arch. [5] | Phy. Loss Grad [6] | RAD/ Imp. Sample [7] | L-p [8] | A-PINN [9] | NTK [10] | Evo/ CEvo (ours) |   |   |
> > |------------------------------|-----------------|-----------|--------------|-----------|-----------------|--------------------|----------------------|---------|------------|----------|------------------|---|---|
> > | Convection Equation          | ✓               |           |              |           |                 |                    |                      |         |            |          | ✓                |   |   |
> > | Reaction-Diffusion Equation  | ✓               |           |              | ✓         |                 |                    |                      |         |            |          |                  |   |   |
> > | Ekkonal Equation             |                 |           |              |           |                 |                    |                      |         |            |          | ✓                |   |   |
> > | Allen Cahn                   |                 | ✓         |              | ✓         |                 |                    |                      |         |            |          | ✓                |   |   |
> > | Lorentz System               |                 | ✓         |              |           |                 |                    |                      |         |            |          |                  |   |   |
> > | KS-Equation (simple)         |                 | ✓         |              |           |                 |                    |                      |         |            |          | ✓                |   |   |
> > | KS-Equation (chaotic)        |                 | ✓         |              |           |                 |                    |                      |         |            |          | ✓                |   |   |
> > | Navier Stokes                |                 | ✓         |              |           |                 |                    |                      |         |            |          |                  |   |   |
> > | Burgers Equation             |                 |           | ✓            | ✓         |                 | ✓                  |                      |         |            |          |                  |   |   |
> > | Harmonic Equation            |                 |           |              |           |                 | ✓                  |                      |         |            |          |                  |   |   |
> > | 1D-Possion Equation          |                 |           | ✓            |           |                 |                    |                      |         |            | ✓        |                  |   |   |
> > | Wave Equation                |                 |           |              | ✓         |                 |                    |                      |         |            | ✓        |                  |   |   |
> > | 1-D Transient Wave Eqn       |                 |           |              |           | ✓               |                    |                      |         |            |          |                  |   |   |
> > | KdV Equation                 |                 |           |              |           | ✓               |                    |                      |         |            |          |                  |   |   |
> > | Helmoltz Equation            |                 |           |              |           | ✓               |                    |                      |         | ✓          |          |                  |   |   |
> > | Lid-driven Cavity            |                 |           |              |           | ✓               |                    |                      |         | ✓          |          |                  |   |   |
> > | Isotropic-Elasticity Problem |                 |           |              |           |                 |                    | ✓                    |         |            |          |                  |   |   |
> > | Plane Stress Problem         |                 |           |              |           |                 |                    | ✓                    |         |            |          |                  |   |   |
> > | HJB Equation                 |                 |           |              |           |                 |                    |                      | ✓       |            |          |                  |   |   |
> > | Klein Gordon Equation        |                 |           |              |           |                 |                    |                      |         | ✓          |          |                  |   |   |

---

> > > ### Author Response · Authors · 2022-11-19
> > > **Author Response for Reviewer rJ3m - Part 3**
> > >
> > > **References for the papers used in the Table:**
> > >
> > > [1] Krishnapriyan, Aditi, Amir Gholami, Shandian Zhe, Robert Kirby, and Michael W. Mahoney. "Characterizing possible failure modes in physics-informed neural networks." Advances in Neural Information Processing Systems 34 (2021): 26548-26560.
> > >
> > > [2] Wang, Sifan, Shyam Sankaran, and Paris Perdikaris. "Respecting causality is all you need for training physics-informed neural networks." arXiv preprint arXiv:2203.07404 (2022).
> > >
> > > [3] Lu, Lu, Xuhui Meng, Zhiping Mao, and George Em Karniadakis. "DeepXDE: A deep learning library for solving differential equations." SIAM Review 63, no. 1 (2021): 208-228.
> > >
> > > [4] Wu, Chenxi, Min Zhu, Qinyang Tan, Yadhu Kartha, and Lu Lu. "A comprehensive study of non-adaptive and residual-based adaptive sampling for physics-informed neural networks." Computer Methods in Applied Mechanics and Engineering 403 (2023): 115671.
> > >
> > > [5] Wong, Jian Cheng, Chinchun Ooi, Abhishek Gupta, and Yew-Soon Ong. "Learning in sinusoidal spaces with physics-informed neural networks." IEEE Transactions on Artificial Intelligence (2022).
> > >
> > > [6] Leiteritz, Raphael, and Dirk Pflüger. "How to Avoid Trivial Solutions in Physics-Informed Neural Networks." arXiv preprint arXiv:2112.05620 (2021).
> > >
> > > [7] Nabian, Mohammad Amin, Rini Jasmine Gladstone, and Hadi Meidani. "Efficient training of physics-informed neural networks via importance sampling." Computer‐Aided Civil and Infrastructure Engineering 36, no. 8 (2021): 962-977.
> > >
> > > [8] Wang, Chuwei, Shanda Li, Di He, and Liwei Wang. "Is $ L^ 2$ Physics-Informed Loss Always Suitable for Training Physics-Informed Neural Network?." arXiv preprint arXiv:2206.02016 (2022).
> > >
> > > [9] Wang, Sifan, Yujun Teng, and Paris Perdikaris. "Understanding and mitigating gradient flow pathologies in physics-informed neural networks." SIAM Journal on Scientific Computing 43, no. 5 (2021): A3055-A3081.
> > >
> > > [10] Wang, Sifan, Xinling Yu, and Paris Perdikaris. "When and why PINNs fail to train: A neural tangent kernel perspective." Journal of Computational Physics 449 (2022): 110768.
> > >
> > >
> > > **Additional Results on KS-Equations:**
> > > As per the reviewer’s suggestion to compare on the PDEs that were used in the CPINN paper, we have added three new experiments on KS-Equations (one for a regular relatively simple case and the remaining two exhibiting chaotic behavior). Please note that these equations are particularly more complex, especially the chaotic cases where a small change in the state of the solution can result in very large errors downstream in time. Thus, for chaotic domains, the successful propagation of solution from the initial and boundary conditions is critical to guarantee convergence. We would also like to highlight that the computational cost of these experiments are significantly higher (please refer to the original CPINN for training time comparisons). We used the exact same hyper-parameter settings as those provided in CPINN except the sample size, which was varied from 128 to 2048 in the KS-Equation (Regular Case), and the number of training iterations, which was kept as 300k in our proposed approaches while CPINN was allowed to use about 1 M maximum number of iterations with early stopping. Our method on average takes 50-60% less time than CPINN because of the significantly smaller number of iterations.
> > >
> > > **Comparing Evo, Causal Evo, and CPINN:**
> > > Figure 19 compares the performance of CPINN, Evo, and Causal Evo on the KS equation (regular case) as a function of the number of collocation points used in PINN training. We can see that both Evo and Causal Evo show improvements over CPINN when the number of collocation points is small ($N_r = 128$). As the number of collocation points is increased, Causal Evo shows better performance than Evo, as it incorporates an additional prior of causality along with satisfying the four motivating properties of Evo. Overall, Causal Evo mostly performs better than both CPINN and Evo across different training set sizes. Note that these curves have been obtained using a single run of every method due to the computational cost of training for the KS equations, and having multiple runs for every method will help to quantify the variance in these results.

---

> > > > ### Author Response · Authors · 2022-11-19
> > > > **Author Response for Reviewer rJ3m - Part 4**
> > > >
> > > > Table 4 compares the performance of CPINN, Evo, and Causal Evo on the three KS-Equations cases as was used in the original CPINN paper. Please note that for these experiments, we used the exact same hyper-parameter settings as the original CPINN, thus a large number of collocation points were used (2048 for the regular case and 8192 for the two chaotic regimes for each time-window). We can observe that on the regular case, Evo performs similarly to CPINN, while Causal Evo is significantly better than both. However, in the first chaotic case, CPINN is slightly better than both Evo and Causal Evo. Finally, in the much more chaotic regime for the KS-Equation  (extended case), we find that all of the methods struggle to obtain a high fidelity solution of the field. However, Evo and Causal Evo are somewhat better than CPINNs. Hence, we can comment that Evo and CausalEvo have comparable performance to CPINN on their benchmark settings (additional visualizations are shown in Appendix Section J.7).
> > > >
> > > > |                    | CausalPINN | Evo (ours) | CausalEvo (ours) |
> > > > |:------------------:|:----------:|:----------:|:----------------:|
> > > > |       Regular      |   2.120%   |   3.740%   |      0.761%      |
> > > > |       Chaotic      |   3.272%   |   6.924%   |      7.630%      |
> > > > | Chaotic - Extended |   52.66%   |   29.26%   |      33.50%      |
> > > >
> > > > **Comment 3 [Discussing Benefits of Causal Evo]:** the benefit of Causal Evo is not shown or discussed in the experiments. I understand Evo as a method that mitigated training failures, rather than one that improves performance for a successfully trained PINN. Thus, from that perspective, Evo and Causal Evo perform equivalently in the shown experiments. It would be interesting to see an example where Evo fails because causality is not respected, but where Causal Evo successfully trains the PINN.
> > > >
> > > > **Response:** We thank the reviewer for asking this question. Note that while Evo satisfies all of the four motivating properties behind our work to mitigate propagation failures, Causal Evo also additionally respects the causality principle for time-dependent PDEs. As suggested by the reviewer, Causal Evo shows similar performance than Evo on the Convection equation and Allen Cahn equation.
> > > >
> > > > To show examples where Evo and Causal Evo show different performances, we have now included new experimental results on three cases of the KS equation, as detailed in the response to Comment 2 above. We would like to specifically refer the reviewer to see Figure 19 in the revised paper that compares Evo, Causal Evo, and CPINN over varying numbers of collocation points, where we can see that Evo and Causal Evo perform better at different ranges of sample sizes.. We are copying the discussion of this Figure below (same as what is mentioned in the response to comment 2).
> > > >
> > > > Figure 19 compares the performance of CPINN, Evo, and Causal Evo on the KS equation (regular case) as a function of the number of collocation points used in PINN training. We can see that both Evo and Causal Evo show improvements over CPINN when the number of collocation points is small ($N_r = 128$). As the number of collocation points is increased, Causal Evo shows better performance than Evo, as it incorporates an additional prior of causality along with satisfying the four motivating properties of Evo. Overall, Causal Evo mostly performs better than both CPINN and Evo across different training set sizes. Note that these curves have been obtained using a single run of every method due to the computational cost of training for the KS equations, and having multiple runs for every method will help to quantify the variance in these results.
> > > >
> > > >
> > > >
> > > > **Comment 4 [Missing References]:** In the paragraph "Propagation Failure", the authors argue that trivial solutions can become attractive for PINNs. This has been made precise theoretically in arXiv:2109.09338 and experimentally in arXiv:2203.13648.
> > > >
> > > > **Response:** We thank the reviewer for the pointers to these papers, which is very helpful in motivating the problem of trivial solutions. We have added them now to the related works section.
> > > >
> > > > **Comment 5 [Appendix References Error]:** References to the appendix are broken.
> > > >
> > > > **Response:** We have fixed this now and the appendix is now available at the end of the main paper (which was earlier provided as a separate PDF in the supplementary materials). Thanks for pointing this out.

---

> > > > > ### Author Response · Authors · 2022-11-19
> > > > > **Author Response for Reviewer rJ3m - Part 5**
> > > > >
> > > > > **Comment 6 [Clarity on Computational Overhead]:** The computational overhead of the comparison methods is not clear ("Challenges with Potential Remedies").
> > > > >
> > > > > **Response:** To clarify how existing methods such as RAR suffer from high computational cost, we have revised the text in our discussion of “Challenges with Potential Remedies” as follows:
> > > > >
> > > > > “Sampling methods such as RAR and its variants require using a dense set of collocation points $\mathcal{P}_{dense}$ (typically with $100\text{K} \sim 1\text{M}$ points spread uniformly across the entire domain) to locate high residual regions, such that points from high residual regions can added to the training set every $K$ iterations. This increases the computational cost in two ways. First, computing the PDE residuals on the entire dense set is very expensive. Second, the size of the training set keeps growing every $K$ iterations, further increasing the training costs at later iterations. See Appendix F for a detailed analysis of the computational complexity of RAR based methods.”
> > > > >
> > > > >
> > > > > **Comment 7 [Clarity on Size of Retained/Resampled Population]:**  For Evo it is not clear how p_i^s changes throughout training (i.e., is it of fixed size, or is rather  of fixed size?).
> > > > >
> > > > > **Response:** The total size of the population $\mathcal{P}_i$ is kept constant (equal to $N_r$) across all iterations in Evo. Hence, for any iteration $i$, $|\mathcal{P}_i| = N_r = |\mathcal{P}^r_i| + |\mathcal{P}^s_i|$, where $|\mathcal{P}|$ denotes the size of $\mathcal{P}$. Since the size of $ \mathcal{P}^r_i$ is variable and depends on the fitness of collocation points in $\mathcal{P}_i$, the size of $\mathcal{P}^s_i$ is also variable such that they both add up to $N_r$. We have made necessary modifications to Algorithm 1 in line 6 to clarify this point. We refer the reviewers to see Appendix Section K for detailed visualizations on how the size and distribution of the retained and re-sampled populations changes over iterations for a variety of “fixed” objective functions.

---

> > > > > > ### Author Response · Authors · 2022-11-19
> > > > > > **Author Response for Reviewer rJ3m - Part 6**
> > > > > >
> > > > > > **Comment 8 [Sufficiency of Motivating Properties and Comparison with CPINN]:** The four salient properties seem somewhat artificial and post-hoc. For example, the gradual accumulation property is not well justified. As an alternative property, it seems equally intuitive that initially the collocation points should concentrate around the initial and boundary conditions (ensuring successful learning locally, from which propagation can continue), while at later stages the collocation points should be uniform (ensuring uniform accuracy across the computational domain). Thus, a priori, the CPINN sampling strategy seems equally powerful to mitigate propagation failures as Evo. It is even more interesting to see that the CPINN fail for the systems for which Evo succeeds, and I would appreciate an evaluation of Evo on the benchmarks that were chosen by the original authors of CPINN.
> > > > > >
> > > > > > **Response:** We agree with the reviewer that the four properties used to motivate Evo are not exhaustive and there are other properties that can motivate alternative sampling algorithms, e.g., respecting the principle of causality. However, note that respecting causality alone is not sufficient to mitigate propagation failure modes of PINNs, which are characterized by highly imbalanced residuals appearing in very narrow regions of the domain. This is because not all points at the same time instance are equally important to be sampled to mitigate the imbalance in the PDE residuals. Instead, we need to specifically target the high residual regions acting as propagation barriers at every time instance. We have used the principle of causality as an additional prior in Causal Evo along with respecting the four motivating properties of Evo. We have revised our description of the four properties to clarify that they do not constitute an exhaustive list describing any ideal sampling algorithm but rather are the ones that we focus on in this work to motivate Evo.
> > > > > >
> > > > > > As mentioned in a previous response above, we chose to compare our results on the Allen Cahn equation as a common benchmark PDE that was also used in the Causal PINN (CPINN) paper (available currently as an Arxiv version). To further expand our evaluation set, we have now added new results comparing the performance of CPINN and our proposed methods (Evo and Causal Evo) on three cases of the complex Kuramoto-Sivashinsky (KS) equation that were used in the CPINN paper, with the same setting of hyper-parameters and experiment design as CPINN. We can see that both Evo and Causal Evo show improvements over CPINN when the number of collocation points is small. In fact, on the Allen Cahn equations, CPINN fails to converge with low errors when the number of collocation points is 128.  As the number of collocation points is increased, Causal Evo shows better performance than Evo on the KS equation (regular case), as it incorporates an additional prior of causality along with satisfying the four motivating properties of Evo.

---

> ### Comment · Reviewer_rJ3m · 2022-11-21
> **Thank you very much!**
>
> I thank the authors for the extensive reply. This has indeed clarified a lot, and I appreciate the effort the authors have taken to produce additional results. This in itself puts the experimental analysis on more solid grounds, and I am tending to increase my score accordingly. The specific differences between Eva and CausalEvo are still not very clear, but I accept that this may be the subject of future work.

---

> > ### Author Response · Authors · 2022-12-12
> > **Thank you and follow up**
> >
> > Dear Reviewer rJ3m,
> >
> > Thank you again for acknowledging our response and appreciating our efforts in addressing the comments/questions of the reviewers. It is great to hear that you are tending to increase your score accordingly, now that the experimental analysis is on more solid grounds. We would really appreciate if you can update your score before the meta review is out so that it is reflected in the overall review process.
> >
> > Thank you!

---

### Official Review · Reviewer_9MDS · 2022-10-18

**Confidence:** 4
**Correctness:** 3
**Technical Novelty And Significance:** 2
**Empirical Novelty And Significance:** 3
**Recommendation:** 3

**Clarity, Quality, Novelty And Reproducibility:**

Clarity: Overall, the paper is well written.

Quality: I have some concerns regarding the theory (listed above).

Novelty: Somehow,  proposals are mostly reappraisals of previous techniques for PINNs (i.e., weighting based on error and enforcing "physical causality").

Reproducibility: The authors also provide code and experiments seem reproducible.

**Strength And Weaknesses:**

## Strengths
* Authors formulate an intuitive concept to identify failure modes of PINNS;
* Authors propose a simple strategy that improves significantly the sample efficiency of PINNS;
* Ideas are clearly presented and the work is easy to follow.

## Weaknesses
* The idea of focusing on regions of high error is not exactly novel in PINNs;
* While the "propagation hypothesis" is intuitive, there is no formal characterization for it in the work;
* The experimental campaign is shorter than I would expect from a paper with a  very simple idea and no theoretical guarantees;
* The theory is inconsistent (and perhaps wrong) --- see details below.
* The experiments lack a more thorough discussion. For instance, do you have an intuition on why Evo is capable of solving the Eikonal equations in some cases (Figure 8) and other methods not?

###More about the theory

For **theorem 3.1**, there are two issues:

1. The step from lines 5 to 6 on page 11 of the appendix multiplies by $V^{-1/2}$ but should multiply by $V^{1/2}$ since $p(\mathbf{x}_r)^{-1} = V$;

 2. The paragraph before theorem 3.1 is misleading. It says that sampling proportional to a power of $\mathcal{R}$ is equivalent to changing the $L^p$ loss in the norm. However, minimizing $\mathcal{L}^2_r(\mathcal{Q}^k)$ is not the same as minimizing $\mathcal{L}^2_r(\mathcal{Q}^k)$  (or a power thereof) os not the same since $Z= \int_{\mathbf{x}_r \in \Omega} |\mathcal{R}(\mathbf{x}_r) |^k d\, \mathbf{x}_r$ depends on network parameters (and therefore is not a constant).

For **theorem 4.1**, states that, as the number of iterations of Algorithm one approach $\infty$, their loss reverts to an $L^\infty$ loss. However, on page 5, the authors state that training PINNS with $L^\infty$ is undesirable and can lead to $oscillatory behavior between different peaks of the PDE residual landscape". Does this mean we should always use a small number of iterations in Algorithm 1? This aspect deserves further discussion.


**Summary Of The Paper:**

The paper proposes "the propagation hypothesis" as an explanation of why some (collocation) sampling schemes might cause PINNs to fail.
To mitigate the issue, the authors propose an _evolutionary_ sampling strategy, that keeps points with high error in a pool and resamples others. They also incorporate a time-weighting scheme to propagate the importance of samples from $t=0$ onwards. The authors show that their method outperforms prior art in 3 datasets when the number of samples used is small.

**Summary Of The Review:**

Given that the novelty is somehow limited and my concerns with the implications of the theory & the short experimental campaign, I recommend rejection.

---

> ### Author Response · Authors · 2022-11-19
> **Author Response for Reviewer 9MDS - Part 1**
>
> **Comment 1 [Novelty]:** The idea of focusing on regions of high error is not exactly novel in PINNs;
>
> **Response:** We agree with the reviewer that the problem of focusing on high error regions is not novel, which is reflected from our discussion of existing approaches in the paragraph “Potential Remedies for Mitigating Propagation Failures” in Section 3. However, all existing approaches for this problem suffer from one or more of the following two key limitations limiting their practical usability: (1) **high computational cost:** they require an additional dense set of collocation points to perform adaptive sampling, which can be computationally prohibitive, and (2) **poor prediction performance:** they require knowing the optimal settings of hyper-parameters in the sampling algorithm (e.g., the value of $k$  in sampling-based methods or the value of $p$ in $L^p$ norm-based methods), which is domain-specific and may change dynamically over iterations. Both these limitations are described in detail in the paragraph “Challenges with Potential Remedies” in Section 3.
>
> The novelty of our work lies in providing the first solution to this problem that overcomes both these limitations simultaneously: (1) Evo does not require a dense set of collocation points and hence is sample efficient, (2) Evo shows superior prediction performance over a wide selection of benchmark PDEs as it can dynamically accumulate points from high residual regions and release points when their residuals are resolved, while maintaining non-zero representation of points from other regions. We also introduce a novel perspective of the motivation behind focusing on high residual regions by postulating the propagation hypothesis, which has roots in physics-based numerical methods. We have added the following paragraph to the Introduction section to explicitly state the novel contributions of our work.
>
> “The novel contributions of our work are as follows. (1) We provide a novel perspective for characterizing failure modes in PINNs by postulating the ``Propagation Hypothesis.’’ (2) We propose a novel evolutionary algorithm Evo to adaptively sample collocation points in PINNs that shows superior prediction performance empirically with little to no computational overhead compared to existing methods for adaptive sampling. (3) We theoretically show that Evo can accumulate points from high residual regions if they persist over iterations and release points if they have been resolved by PINN training, while maintaining non-zero representation of points from other regions.”
>
> To further highlight the novelty of our work, we have also added a new table in Appendix E comparing baseline methods with Evo and Causal Evo in terms of their ability to comply with the motivating properties of this work. We have copied the table below for the ease of reviewers.
>
> |            | Accumulation Property | Release Property | Uniform Background Property | Computational Efficiency | Causality |   |
> |------------|:---------------------:|:----------------:|:---------------------------:|:------------------------:|:---------:|---|
> | RAR-G      |           ✓           |                  |              ✓              |                          |           |   |
> | RAD        |           ✓           |         ✓        |              ✓              |                          |           |   |
> | RAR-D      |           ✓           |                  |              ✓              |                          |           |   |
> | L-infinity |           ✓           |         ✓        |                             |             ✓            |           |   |
> | CausalPINN |                       |                  |              ✓              |             ✓            |     ✓     |   |
> | Evo        |           ✓           |         ✓        |              ✓              |             ✓            |           |   |
> | Causal Evo |           ✓           |         ✓        |              ✓              |             ✓            |     ✓     |   |

---

> > ### Author Response · Authors · 2022-11-19
> > **Author Response for Reviewer 9MDS - Part 2**
> >
> > **Comment 2 [Theoretical Proof of Propagation Hypothesis]:** While the "propagation hypothesis" is intuitive, there is no formal characterization for it in the work
> >
> > **Response:** We thank the reviewer for the comment and agree that our work does not provide a theoretical characterization of the “propagation hypothesis,” even though it is intuitive to understand and can be demonstrated empirically by visualizing the skewness and kurtosis of PDE residual fields. Given that the field of research in PINNs is still in a nascent stage, devising a general tool for theoretically characterizing the failure modes of PINNs for any arbitrary PDE is a hard problem with no existing solutions. As a result, existing work on characterizing PINN failure modes that have been published at high-impact venues have mostly relied on empirical observations. For example, the seminal work on characterizing failure modes of PINNs by Krishnapriyan et al. [1] published at NeurIPS 2021 solely relied on empirical evidence on a set of simple benchmark PDE problems (e.g.,  the 1D-Convection equation and Reaction-Diffusion equations).
> >
> > We have named our new perspective of PINN failure modes as a “hypothesis” for the simple reason that we only have empirical evidence to support the propagation hypothesis. It is worth mentioning that hypothesis-driven papers have made major impacts in other fields of machine learning such as the best paper award-winning work in ICLR 2018 on the Lottery Ticket Hypothesis by Frankle et al. [2], which inspired a number of subsequent works published at ICLR. We expect our work on the propagation hypothesis to similarly inspire future research in the field of PINNs that explore its formal characterization.
> >
> > Finally, please note that while we do not provide a formal characterization of the propagation hypothesis, we theoretically analyze our proposed Evo method to show that it satisfies the four motivating properties that we focus on in our work, derived from empirical observations of the propagation hypothesis.
> >
> > [1] Krishnapriyan, Aditi, et al. "Characterizing possible failure modes in physics-informed neural networks." Advances in Neural Information Processing Systems 34 (2021)
> >
> > [2] Frankle, Jonathan, and Michael Carbin. "The Lottery Ticket Hypothesis: Finding Sparse, Trainable Neural Networks." (Best Paper Award, ICLR 2018)
> >
> >
> > **Comment 3 [More Experimental Results]:** The experimental campaign is shorter than I would expect from a paper with a very simple idea and no theoretical guarantees;
> >
> > **Response:** Our choice of PDEs (Convection equation and Allen Cahn) was primarily motivated to have the same set of benchmark PDEs as those used in previous literature, to allow for easy comparison. For example, we borrowed the convection equation from the Curriculum Regularization paper [1] (NeurIPS 2021) with the exact same PDE parameters, PINN hyper-parameters, and experiment settings (except the number of collocation points). It was shown in [1] that large values of $\beta$ could be difficult to optimize for PINNs. Hence, we used $\beta=30$ and $\beta=50$. On the other hand, we borrowed the Allen Cahn Equation from the CausalPINN paper [2], which is currently only available as an Arxiv version. In [2], it was shown that Allen Cahn can fail to converge if fewer collocation points were used. The comprehensive study on RAR-based methods [3,4] also evaluated their method on the Allen Cahn Equation.
> >
> > Note that due to the diversity of physical phenomena represented by PDEs, experiments in the domain of PINNs are very diverse and there is a lack of standard benchmarks as is common in other AI/ML fields such as computer vision. Most of the papers involving PINNs use a different set of PDEs with varying set of hyper-parameters, depending on the specific target problem they are trying to solve. To consolidate the diversity of PDEs explored in related works, we have prepared a table of PDE datasets used in 10 representative papers in the area of PINNs that we have included in our related work (including the ones suggested by the reviewer). In our work, we used the same training and hyper-parameters setups as those reported in previous baselines to perform direct comparisons. For example, for both Convection Equation and the Allen Cahn Equation, we have used the same setting of neural network architecture, initial conditions, PDE coefficients, and number of training iterations as the existing methods, without fine-tuning our hyper-parameters on any problem. Hence, our method can be thought of as a “plug-in” that was added to existing evaluation frameworks.

---

> > > ### Author Response · Authors · 2022-11-19
> > > **Author Response for Reviewer 9MDS - Part 3**
> > >
> > > **Table to compare the Benchmark PDE's used for each representative paper:**
> > >
> > > |                              | Curr.  Reg. [1] | CPINN [2] | DeepXDE [3]  | RAR-D [4] | Sine  Arch. [5] | Phy. Loss Grad [6] | RAD/ Imp. Sample [7] | L-p [8] | A-PINN [9] | NTK [10] | Evo/ CEvo (ours) |   |   |
> > > |------------------------------|-----------------|-----------|--------------|-----------|-----------------|--------------------|----------------------|---------|------------|----------|------------------|---|---|
> > > | Convection Equation          | ✓               |           |              |           |                 |                    |                      |         |            |          | ✓                |   |   |
> > > | Reaction-Diffusion Equation  | ✓               |           |              | ✓         |                 |                    |                      |         |            |          |                  |   |   |
> > > | Ekkonal Equation             |                 |           |              |           |                 |                    |                      |         |            |          | ✓                |   |   |
> > > | Allen Cahn                   |                 | ✓         |              | ✓         |                 |                    |                      |         |            |          | ✓                |   |   |
> > > | Lorentz System               |                 | ✓         |              |           |                 |                    |                      |         |            |          |                  |   |   |
> > > | KS-Equation (simple)         |                 | ✓         |              |           |                 |                    |                      |         |            |          | ✓                |   |   |
> > > | KS-Equation (chaotic)        |                 | ✓         |              |           |                 |                    |                      |         |            |          | ✓                |   |   |
> > > | Navier Stokes                |                 | ✓         |              |           |                 |                    |                      |         |            |          |                  |   |   |
> > > | Burgers Equation             |                 |           | ✓            | ✓         |                 | ✓                  |                      |         |            |          |                  |   |   |
> > > | Harmonic Equation            |                 |           |              |           |                 | ✓                  |                      |         |            |          |                  |   |   |
> > > | 1D-Possion Equation          |                 |           | ✓            |           |                 |                    |                      |         |            | ✓        |                  |   |   |
> > > | Wave Equation                |                 |           |              | ✓         |                 |                    |                      |         |            | ✓        |                  |   |   |
> > > | 1-D Transient Wave Eqn       |                 |           |              |           | ✓               |                    |                      |         |            |          |                  |   |   |
> > > | KdV Equation                 |                 |           |              |           | ✓               |                    |                      |         |            |          |                  |   |   |
> > > | Helmoltz Equation            |                 |           |              |           | ✓               |                    |                      |         | ✓          |          |                  |   |   |
> > > | Lid-driven Cavity            |                 |           |              |           | ✓               |                    |                      |         | ✓          |          |                  |   |   |
> > > | Isotropic-Elasticity Problem |                 |           |              |           |                 |                    | ✓                    |         |            |          |                  |   |   |
> > > | Plane Stress Problem         |                 |           |              |           |                 |                    | ✓                    |         |            |          |                  |   |   |
> > > | HJB Equation                 |                 |           |              |           |                 |                    |                      | ✓       |            |          |                  |   |   |
> > > | Klein Gordon Equation        |                 |           |              |           |                 |                    |                      |         | ✓          |          |                  |   |   |

---

> > > > ### Author Response · Authors · 2022-11-19
> > > > **Author Response for Reviewer 9MDS - Part 4**
> > > >
> > > > **References for the papers used in the Table:**
> > > >
> > > > [1] Krishnapriyan, Aditi, Amir Gholami, Shandian Zhe, Robert Kirby, and Michael W. Mahoney. "Characterizing possible failure modes in physics-informed neural networks." Advances in Neural Information Processing Systems 34 (2021): 26548-26560.
> > > >
> > > > [2] Wang, Sifan, Shyam Sankaran, and Paris Perdikaris. "Respecting causality is all you need for training physics-informed neural networks." arXiv preprint arXiv:2203.07404 (2022).
> > > >
> > > > [3] Lu, Lu, Xuhui Meng, Zhiping Mao, and George Em Karniadakis. "DeepXDE: A deep learning library for solving differential equations." SIAM Review 63, no. 1 (2021): 208-228.
> > > >
> > > > [4] Wu, Chenxi, Min Zhu, Qinyang Tan, Yadhu Kartha, and Lu Lu. "A comprehensive study of non-adaptive and residual-based adaptive sampling for physics-informed neural networks." Computer Methods in Applied Mechanics and Engineering 403 (2023): 115671.
> > > >
> > > > [5] Wong, Jian Cheng, Chinchun Ooi, Abhishek Gupta, and Yew-Soon Ong. "Learning in sinusoidal spaces with physics-informed neural networks." IEEE Transactions on Artificial Intelligence (2022).
> > > >
> > > > [6] Leiteritz, Raphael, and Dirk Pflüger. "How to Avoid Trivial Solutions in Physics-Informed Neural Networks." arXiv preprint arXiv:2112.05620 (2021).
> > > >
> > > > [7] Nabian, Mohammad Amin, Rini Jasmine Gladstone, and Hadi Meidani. "Efficient training of physics-informed neural networks via importance sampling." Computer‐Aided Civil and Infrastructure Engineering 36, no. 8 (2021): 962-977.
> > > >
> > > > [8] Wang, Chuwei, Shanda Li, Di He, and Liwei Wang. "Is $ L^ 2$ Physics-Informed Loss Always Suitable for Training Physics-Informed Neural Network?." arXiv preprint arXiv:2206.02016 (2022).
> > > >
> > > > [9] Wang, Sifan, Yujun Teng, and Paris Perdikaris. "Understanding and mitigating gradient flow pathologies in physics-informed neural networks." SIAM Journal on Scientific Computing 43, no. 5 (2021): A3055-A3081.
> > > >
> > > > [10] Wang, Sifan, Xinling Yu, and Paris Perdikaris. "When and why PINNs fail to train: A neural tangent kernel perspective." Journal of Computational Physics 449 (2022): 110768.
> > > >
> > > > **Additional Results on KS-Equations:**
> > > > To expand our set of experimental results as suggested by the reviewer, we have added three new experiments on KS-Equations (one for a regular relatively simple case and the remaining two exhibiting chaotic behavior). Please note that these equations are particularly more complex, especially the chaotic cases where a small change in the state of the solution can result in very large errors downstream in time. Thus, for chaotic domains, the successful propagation of solution from the initial and boundary conditions is critical to guarantee convergence. We would also like to highlight that the computational cost of these experiments are significantly higher (please refer to the original CPINN for training time comparisons). We used the exact same hyper-parameter settings as those provided in CPINN except the sample size, which was varied from 128 to 2048 in the KS-Equation (Regular Case), and the number of training iterations, which was kept as 300k in our proposed approaches while CPINN was allowed to use about 1 M maximum number of iterations with early stopping. Our method on average takes 50-60% less time than CPINN because of the significantly smaller number of iterations.
> > > >
> > > > **Comparing Evo, Causal Evo, and CPINN:**
> > > > Figure 19 compares the performance of CPINN, Evo, and Causal Evo on the KS equation (regular case) as a function of the number of collocation points used in PINN training. We can see that both Evo and Causal Evo show improvements over CPINN when the number of collocation points is small ($N_r = 128$). As the number of collocation points is increased, Causal Evo shows better performance than Evo, as it incorporates an additional prior of causality along with satisfying the four motivating properties of Evo. Overall, Causal Evo mostly performs better than both CPINN and Evo across different training set sizes. Note that these curves have been obtained using a single run of every method due to the computational cost of training for the KS equations, and having multiple runs for every method will help to quantify the variance in these results.

---

> > > > > ### Author Response · Authors · 2022-11-19
> > > > > **Author Response for Reviewer 9MDS - Part 5**
> > > > >
> > > > > Table 4 compares the performance of CPINN, Evo, and Causal Evo on the three KS-Equations cases as was used in the original CPINN paper. Please note that for these experiments, we used the exact same hyper-parameter settings as the original CPINN, thus a large number of collocation points were used (2048 for the regular case and 8192 for the two chaotic regimes for each time-window). We can observe that on the regular case, Evo performs similarly to CPINN, while Causal Evo is significantly better than both. However, in the first chaotic case, CPINN is slightly better than both Evo and Causal Evo. Finally, in the much more chaotic regime for the KS-Equation  (extended case), we find that all of the methods struggle to obtain a high fidelity solution of the field. However, Evo and Causal Evo are somewhat better than CPINNs. Hence, we can comment that Evo and CausalEvo have comparable performance to CPINN on their benchmark settings (additional visualizations are shown in Appendix Section J.7).
> > > > >
> > > > > |                    | CausalPINN | Evo (ours) | CausalEvo (ours) |
> > > > > |:------------------:|:----------:|:----------:|:----------------:|
> > > > > |       Regular      |   2.120%   |   3.740%   |      0.761%      |
> > > > > |       Chaotic      |   3.272%   |   6.924%   |      7.630%      |
> > > > > | Chaotic - Extended |   52.66%   |   29.26%   |      33.50%      |
> > > > >
> > > > > **Comment 4 [Clarity on Theorem 3.1]:** For theorem 3.1, there are two issues:
> > > > > 1. The steps in last three lines of Equation 5 in the Appendix multiplies by $V^{1/2}$ in the denominator but should multiply by $V^{-1/2}$ since $p(\mathbf{x}_r)^{-1}=V$.
> > > > > 2. The paragraph before theorem 3.1 is misleading. It says that sampling proportional to a power of k is equivalent to changing the $L^{p}$ loss in the norm. However, minimizing $\mathcal{L}^2_r(\mathcal{Q}^k)$ is not the same as minimizing $\mathcal{L}^p_r(\mathcal{U})$ (or a power thereof) since $Z = \int_{\mathbf{x}_r \in \Omega} |\mathcal{R}_\theta(\mathbf{x_r})|^k d\mathbf{x_r}$ depends on network parameters (and therefore is not a constant).
> > > > >
> > > > > **Response:**
> > > > > 1. We thank the reviewer for pointing this out. This was indeed a typo, which we have now corrected by using $V^{-1/2}$ instead of $V^{1/2}$ in the denominator.
> > > > > 2. We thank the reviewer for this comment and agree that the normalization constant Z indeed is a function of the network parameters. Note that the steps used to derive Theorem 3.1 are still correct and there is indeed a relationship between sampling from a distribution $q(x)$ and using the $L^p$ norm formulation of the loss. In particular, the two loss formulations are related to each other by a scaling term $\frac{1}{Z^{1/2}V^{-1/2}}$, which is variable in nature as $Z$ depends on the neural network parameters $\theta$. However, optimizing $\mathcal{L}^p_r(\mathcal{U})$ is not directly equivalent to optimizing $\mathcal{L}^2_r(\mathcal{Q}^k)$ (or a power thereof), as correctly pointed out by the reviewer. To remove ambiguity, we have added the following text clarifying the interpretation of Theorem 3.1. “Theorem 3.1 suggests that sampling collocation points from a distribution $q(\mathbf{x_r}) \propto |\mathcal{R}_\theta(\mathbf{x_r})|^k$ and $L^p$ norm of the PDE loss are related to each other by a scaling term $\frac{1}{Z^{1/2}V^{-1/2}}$. However, since this scaling term is variable in nature as $Z$ depends on the neural network parameters $\theta$, optimizing $\mathcal{L}^p_r(\mathcal{U})$ is not directly equivalent to optimizing $\mathcal{L}^2_r(\mathcal{Q}^k)$ (or a power thereof).”
> > > > >
> > > > > Note that this theorem is not directly related to our proposed approach and was originally used to demonstrate the connection between existing methods. In particular, it does not affect the correctness of the theoretical analysis of our proposed approach (Theorem 4.1). In the interest of space, we have now moved this theorem to the Appendix to make room for adding new clarifications and discussions in the main paper..
> > > > >
> > > > > **Comment 5 [Clarity on Theorem 4.1]:** Theorem 4.1 states that as the number of iterations of Algorithm 1 approach $\infty$, their loss reverts to an $L^\infty$ loss. However, the authors also state that training PINNs with $L^\infty$ is undesirable and can lead to oscillatory behavior between different peaks of the PDE residual landscape. Does this mean we should always use a small number of iterations in Algorithm 1? This aspect deserves further discussion.
> > > > >
> > > > > **Response:** We would like to clarify that Theorem 4.1 only applies when the fitness function is kept fixed, e.g., when the neural network parameters are kept constant. However, in our setup, the PINN optimizer is modifying the fitness function by minimizing the residuals at every iteration. As a result, we would not expect the expectation of the fitness for the retained population to reach $L^\infty$. We have added the following revised text in the updated paper to clarify this point.

---

> > > > > > ### Author Response · Authors · 2022-11-19
> > > > > > **Author Response for Reviewer 9MDS - Part 6**
> > > > > >
> > > > > > “Note that at every iteration of PINN training, Evo attempts to retain the set of collocation points in  $\mathcal{P}^r$ with the highest fitness (corresponding to high residual regions). At the same time, the PINN optimizer is attempting to minimize the residuals  by updating $\theta$ and thus in turn affecting the fitness function. We first show that when $\mathcal{F}(\mathbf{x})$ is fixed (e.g., when $\theta$ is kept constant), Evo maximizes $\mathcal{F}(\mathbf{x})$ in the retained population and thus accumulates points from high residual regions. In particular, Theorem 4.1 shows that for a fixed $\mathcal{F}(\mathbf{x})$, the expectation of the retained population in Evo becomes maximum (equal to $L^\infty$) when the number of iterations approaches $\infty$.
> > > > > >
> > > > > > This demonstrates the *accumulation property* of Evo as points from high residual regions would keep accumulating in the retained population and make its expectation maximal if the fitness function is kept fixed. However, since the PINN optimizer is also minimizing the residuals at every iteration, we would not expect the fitness function to be fixed unless a high residual region persists over a long number of iterations. In fact, points from a high residual region would keep on accumulating until they are resolved by the PINN optimizer and thus eventually released from $\mathcal{P}^r$. Also note that Evo always maintains some collocation points from a uniform distribution, i.e., the re-sampled population $\mathcal{P}^s$ is always non-empty. Theoretical proofs of the *sample release property* and the *uniform background property* of Evo are provided in Appendix C.2 and Lemma C.1.1, respectively.  We also provide details of the computational complexity of Evo in comparison with baseline methods in Appendix  F, showing that Evo is *computationally efficient*. Table 2 in the Appendix summarizes our ability to satisfy the motivating properties of our work in comparison with baselines.”
> > > > > >
> > > > > > Hence, to summarize and answer the reviewer’s questions, Algorithm 1 does not have any constraint on the number of iterations to train, as the PINN’s gradient based optimizer (Adam optimizer) would minimize the PDE residual on the set of collocation points at every iteration. Thus, it is highly unlikely that a high residual region would persist for a large number of iterations. However, if such a case arises during the training, the Evo sample would gradually increase the representation of the high residual region in the loss, and would increasingly focus on this region until it is eventually resolved and thus released.

---

> > > > > > > ### Author Response · Authors · 2022-11-19
> > > > > > > **Author Response for Reviewer 9MDS - Part 7**
> > > > > > >
> > > > > > > **Comment 6 [More Discussion of Results]:** The experiments lack a more thorough discussion. For instance, do you have an intuition on why Evo is capable of solving the Eikonal equations in some cases (Figure 8) and other methods not?
> > > > > > >
> > > > > > > **Response:**
> > > > > > > We thank the reviewer for this comment, which helped us to further expand our discussion of results in Appendix J. In particular, we have included the following:
> > > > > > > 1. We have added visualizations of the dynamics of collocation points and discussed its effect on the PDE residuals and absolute errors for Evo, Causal Evo, and RAR-based variants on the Convection equation ($\beta=50$) in Sections J.4, J.5, and J.6.
> > > > > > > 2. We have added visualizations of the dynamics of the solutions of Evo, PINN (fixed), and PINN (dynamic) for the Eikonal equation in Section J.8 for the particularly difficult geometry of `gear.’ We have also added a discussion of the results describing why this problem is hard and why Evo is able to solve it while others are not.
> > > > > > > 3. We have added new experimental results on the computationally demanding problem of solving the Kuramoto-Sivashinsky (KS) equations in Section J.7 for three cases, including two chaotic cases. We have added discussions comparing the performance of  Evo, Causal Evo, and CPINNs on the KS-equations with varying numbers of collocation points.
> > > > > > >
> > > > > > > Regarding the Eikonal equation, we have added the following discussion of results in Section J.8.
> > > > > > >
> > > > > > > “We chose to solve 2D Eikonal Equations for complex arbitrary surface geometries as they represent particularly hard PDE problems that are susceptible to PINN failure modes. In these problems, we are given the zero contours of the equation on the boundaries (representing the outline of the 2D object), which can take arbitrary shapes. The goal is to correctly propagate the boundary conditions to obtain the unique target solution where the interior is negative and the exterior is positive. Here, any small error in propagation from the boundaries can lead to cascading errors such that a large segment of the predicted field can have opposite signs compared to the ground-truth, even though their PDE residuals are close to 0. Since Evo is explicitly designed to break propagation barriers and thus enable easy transmission of the solution from the boundary to the interior/exterior points, we can see that it shows significantly better performance. On the other hand, PINN (fixed) and PINN (dynamic) struggle to converge to the correct solution especially for complex geometries (e.g., the `gear’) because of the inherent challenge in sampling an adequate number of points from arbitrary shaped object boundaries exhibiting highly imbalanced residuals.
> > > > > > >
> > > > > > > In Figure 23, we show the evolution of the solutions of comparative methods for the `gear’ case over iterations. We can see that Evo is able to resolve the high residual regions better than the baselines, and thus encounter less incorrect ``sign flips’’ compared to the ground-truth, even in the early iterations of PINN training.”

---

### Official Review · Reviewer_3fsB · 2022-10-25

**Confidence:** 3
**Correctness:** 3
**Technical Novelty And Significance:** 3
**Empirical Novelty And Significance:** 3
**Recommendation:** 6

**Clarity, Quality, Novelty And Reproducibility:**

- The exposition in the paper is very clear. I enjoyed reading the paper.
- The quality of the paper in terms of ideas, explanations, analysis, experiments and writing are very good.
- Code is available for the paper. Although I did not test it, the code along with the experimental details in the paper should be sufficient to reproduce the main results of the paper.

**Strength And Weaknesses:**

Strengths:
1. Well written, proposed method is sound for the most part.
2. Experimental validation is conducted on multiple problems that are challenging for standard PINN methods and the proposed method is effective in practice.

Weaknesses:
1. I am concerned about the practical viability of the idea of adaptive sampling. In real-world scenarios how realistic is to expect the PDE solution to be available at arbitrary space-time coordinates on demand? I imagine these are obtained through sensor measurements, which may be too constrained to obtain at arbitrary points or too expensive to obtain.
2. While the proposed method is based on a population of samples, there are no mutation or crossover (interaction) operators as you would expect in a typical evolutionary algorithm.

Other Comments:
- Not a weakness per se, but all the references to the appendix in the main paper were missing. This should be an easy fix.

**Summary Of The Paper:**

The paper proposes to use evolutionary sampling to mitigate failures in training PINNs. The main idea is based on the premise that solution of a PINN propagates from boundaries towards the interior regions. In contrast to existing work, Evolutionary sampling is able to adapt to the changing PDE residual error landscape while being computationally efficient. Experiments are conducted on multiple challenging PDEs, and the results demonstrate the effectiveness of the proposed approach.

**Summary Of The Review:**

I am positive about the paper, except for the probable impracticality of real-world adoption. The paper is very well written, and the experiments and analysis are thorough. Most importantly, the method seems to be effective across a range of PDEs.

Post Rebuttal Update:
I read the other reviews and the author's responses. I will maintain the initial rating for the paper.

---

> ### Author Response · Authors · 2022-11-19
> **Author Response for Reviewer 3fsB**
>
> **Comment 1 [Use of Sensor Measurements in Evo]:** I am concerned about the practical viability of the idea of adaptive sampling. In real-world scenarios how realistic is it to expect the PDE solution to be available at arbitrary space-time coordinates on demand? I imagine these are obtained through sensor measurements, which may be too constrained to obtain at arbitrary points or too expensive to obtain.
>
> **Response:** We thank the reviewer for the comment on the practical viability of adaptive sampling. Please note that in the field of PINNs for forward modeling, we do not need to know the PDE “solution” at arbitrary collocation points during training; we only need to compute the “residuals” of the PDE at any collocation point given a neural prediction of its solution. As a result, PINN-based methods for forward modeling are inherently label-free, i.e., they do not require any sensor measurements and only require knowledge of the PDE and initial/boundary conditions, while for the rest of the domain the PDE residuals are computed over a set of sampled collocation points to guide PINN training. In our proposed Evolutionary sampling method, we adaptively sample collocation points at every iteration based on their current PDE residual values, which can be computed from the PDE using automatic differentiation without using additional labels.
>
>
> **Comment 2 [Connections with Evolutionary Algorithms]:** While the proposed method is based on a population of samples, there are no mutation or crossover (interaction) operators as you would expect in a typical evolutionary algorithm.
>
> **Response:** We thank the reviewer for the comment on connections with evolutionary algorithms. We would like to point out that even though our algorithm borrows inspiration from evolutionary algorithms to expedite the sampling process, our method still uses a gradient-based optimizer (Adam optimizer) to minimize the PDE residuals of PINN on collocation points sampled using the proposed evolutionary sampling scheme. Following the dynamics of typical evolutionary algorithms where the population/samples evolve towards the “fittest” population, our algorithm gradually accumulates collocation points in regions where the PDE residuals are high (i.e., the value of the fitness function is large). Although more complex evolutionary algorithms can be formulated to achieve adaptive sampling, we found that our simple EvoSample algorithm satisfies the four motivating properties of our work, namely accumulation property, sample release property, uniform background property, and computational efficiency. We would also like to highlight that our evolutionary sampling is a step towards building a bridge between evolutionary algorithms and sampling techniques. Future work can explore more sophisticated evolutionary algorithms involving mutation and crossover techniques. For example, one can design a mutation scheme such that samples with higher fitness are mutated the least while samples with lower fitness values would undergo significant mutation. In EvoSample, we simply retain points with high fitness and resample the remaining population. We have updated our conclusion section to reflect this future work discussion.
>
>
>
> **Comment 3 [Appendix References Error]:** Not a weakness per se, but all the references to the appendix in the main paper were missing. This should be an easy fix.
>
> **Response:** We have fixed this now and the appendix is now available at the end of the main paper (which was earlier provided as a separate PDF in the supplementary materials). Thanks for pointing this out.

---

### Author Response · Authors · 2022-11-19
**Response to all Reviewers**

We sincerely thank all the reviewers for their feedback. We are encouraged that the reviewers found our work:
1. Well-written and easy to follow
2. Provides intuitive explanations for the methods and propagation hypothesis
3. Proposes a simple strategy that significantly improves the sample efficiency of PINNs

We have provided our response to each reviewers' comments separately below. Here is a summary of the main comments raised by the reviewers and the revisions we have made to address them:
1. Comment: **More experimental results**. Response: We have added new experimental results on three complex and computationally expensive cases of the Kuramoto-Sivashinsky (KS) equations, including two involving chaotic behavior. This enables us to perform one-on-one comparison with CausalPINNs (CPINNs) over a wide range of benchmark PDEs as suggested by the reviewers.
2. Comment: **More discussion of results**. Response: We have included additional visualizations of the dynamics of collocation points and evolution of solutions over iterations for our proposed method and baseline methods over a variety of PDEs. We have also added a detailed discussion of our results on the particularly complex Eikonal equations describing why Evo is able to converge while other methods fail.
3. Comment: **Describing Novelty**. Response: We have added a paragraph at the end of the Introduction section of the paper explicitly stating the novel contributions of this work.
4. Comment: **Clarity on Theorems**. Response: We have revised Theorem 3.1 based on feedback from the reviewers and added more clarifying text in the paragraphs before and after Theorem 4.1 to better describe the theoretical properties of Evo.
5. Comment: **Missing References**. Response: We are grateful to the reviewers for pointing these papers and we have added the suggested references in the related work discussions of the paper.
6. Comment: **Broken Appendix References**. Response: Fixed.

To make it easier for the reviewers to see the revisions in our paper, we have highlighted the major changes that we have made in response to the reviewer’s comments in our revised version in red. Based on our revisions and responses, if the reviewers feel that we have adequately addressed their concerns and clarified their questions, we kindly request the reviewers to appropriately modify their scores.

---

> ### Author Response · Authors · 2022-12-05
> **Thank you and looking forward to your feedback**
>
> Thank you very much reviewer rJ3m for acknowledging our response! We feel very encouraged to be appreciated by the reviewer and to know that they are tending towards increasing their score. We are eagerly waiting to get feedback from the other reviewers on our response and to see if we have adequately addressed their concerns/comments to the point that they are willing to appropriately modify their scores. We will be happy to clarify any outstanding questions. We would again like to thank all the reviewers for their valuable suggestions towards improving our paper.

---

### Decision · Program_Chairs · 2023-01-20

**Decision:**

Reject

**Justification For Why Not Higher Score:**

There are too many open questions regarding the validity of the formal concept proposed.

**Justification For Why Not Lower Score:**

N/A

**Metareview: Summary, Strengths And Weaknesses:**

For this paper, we have two "borderline" (though slightly positive) reviews and one clearly negative review. On the positive side, all reviewers agreed that the main idea is well-motivated and that the paper is well-written. All other aspects, however, have been discussed in a more controversial way.
Severe concerns include the somewhat missing conceptual novelty, not fully convincing comparison studies and  problems with the main concept of the "propagation hypothesis". In the end, none of these two reviewers finally wanted to assign a clearly positive score, and none of them was willing to champion this paper.

A third reviewer was highly critical from the beginning, mentioning  issues with the theoretical foundation (Theorems 3.1 and 4.1). It seems that none of these fundamental points of criticism could be addressed convincingly in the rebuttal, and this reviewer clearly expressed that the paper was "not good enough". Actually, I agree with most of the critical points raised, and I think that this paper generally lacks a clear and transparent theoretical basis. Therefore, I vote for rejection.